# Performative Control for Linear Dynamical Systems

**Songfu Cai**\*, **Fei Han**\*, **Xuanyu Cao**†
Department of Electronic and Computer Engineering
The Hong Kong University of Science and Technology
eesfcai@ust.hk, fhanac@connect.ust.hk, eexcao@ust.hk

## Abstract

We introduce the framework of performative control, where the policy chosen by the controller affects the underlying dynamics of the control system. This results in a sequence of policy-dependent system state data with policy-dependent temporal correlations. Following the recent literature on performative prediction [21], we introduce the concept of a performatively stable control (PSC) solution. We first propose a sufficient condition for the performative control problem to admit a unique PSC solution with a problem-specific structure of distributional sensitivity propagation and aggregation. We further analyze the impacts of system stability on the existence of the PSC solution. Specifically, for almost surely strongly stable policy-dependent dynamics, the PSC solution exists if the sum of the distributional sensitivities is small enough. However, for almost surely unstable policy-dependent dynamics, the existence of the PSC solution will necessitate a temporally backward decaying of the distributional sensitivities. We finally provide a repeated stochastic gradient descent scheme that converges to the PSC solution and analyze its non-asymptotic convergence rate. Numerical results validate our theoretical analysis.

## 1 Introduction

Control theory is a fundamental field of study in engineering and mathematics that centers on coordinating the behaviors of dynamical systems. It has extensive applications in aerospace, robotics, manufacturing, economics, natural sciences, etc. It provides a framework for designing control policies that regulate the system states, enabling them to evolve in a desired manner over time dynamically. The linear state space model is a powerful tool for representing dynamical systems, which employs a set of difference equations to describe the Markovian system state process with a linear state transition model. Many of the existing works on control of linear dynamical systems (LDSs) [26, 9, 31, 7, 17, 28, 31, 32] are developed based on the key assumption that the system state transition model is static. However, in many real world applications, such a static dynamics assumption usually does not hold because system state transition model can be changed by control policies. For instance, in the stock market, the investment policy of well-known investors can impact the actions of the general public investors, resulting in famed investment strategy-dependent changes to the dynamics of stock prices. Another example is autonomous vehicle (AV). A deployed AV might change how the pedestrians and other neighboring cars behave, and the resulting traffic environment might be quite different from what the designers of the AV had in mind [20]. Such interplay between decision-making and decision-dependent system dynamics is a pervasive phenomenon in a multitude of domains, including finance, transportation, and public policy, among others.

Most of the existing works on the control of linear systems with non-static state transition model are primarily focused on the additive random perturbations to the system state transition matrix,

---

\*Equal contribution.
†Corresponding author.

38th Conference on Neural Information Processing Systems (NeurIPS 2024).

where policy optimization methods are employed to find the optimal control policy that minimizes quadratic costs [10, 1, 11, 8]. This type of problem also includes the control of linear systems with additive state-dependent noise [15, 6] or action-dependent noise [30, 4], which is equivalent to having additive perturbations on the state transition matrix or the control input gain matrix, respectively. Some other works on jump/switched linear systems formulate the variation of system model as stochastic model jumps among multiple linear modes, where the jumping law is governed by a finite time-homogeneous Markov process. However, in all these works, the changes of the system model are unrelated to the control policy.

*Performative prediction* provides a systematic way to model the interaction between decision-making and data via decision-dependent data distribution maps [23]. The pioneering work by [21] has led to a growing body of research dedicated to performative prediction problems. Most of these studies are focused on establishing the conditions for the existence and uniqueness of the performative stable point and designing learning algorithms with provable convergence to such a unique performative stable point [13, 14, 5, 3, 18, 29, 25], or algorithms to find a stationary solution of the performative risk [12, 19, 25].

An open question follows: Can the idea of performative prediction, which tackles decision-dependent data in the learning/prediction domain, be employed to address the decision-dependent dynamics in the control domain? Notice that in all the aforementioned performative prediction works [21, 13, 14, 5, 3, 18, 29, 25, 12, 19, 25], the data input to the learning algorithms are independently generated based on the decision-dependent distributions. No temporal correlation is considered or exploited between two consecutive sets of input data. However, in the context of control, the situation is different as the changing system dynamics introduce various additional complexities. This is because the data of the system state at each time the step will depend on the decision-dependent state transition matrices in all previous time steps. The expected total cost function to be optimized will depend on all the decision-dependent state transition matrices across the entire control time horizon. This implies that we need a framework of *performative control* that is more general than the framework of performative prediction, and that can accommodate a sequence of decision-dependent data with temporal correlations, where the temporal correlations are again decision-dependent.

In this work, we provide an affirmative answer to the above question. Our idea hinges on adopting the concept of performative stable solution [21], which is a fixed point solution for the interplay between the controller and the system dynamics that react to the controller's decisions. The condition for the existence of such a performative stable solution is obtained by analyzing the propagation and aggregation of sensitivities associated with the distributions of policy-dependent system dynamics over the entire control time horizon. In particular, we follow existing studies [1, 11, 8] and focus on the disturbance-action policy, which allows the consideration of general convex control costs instead of only quadratic control costs.

To the best of our knowledge, this paper provides the first study and analysis of performative control of linear systems where the system dynamics are constantly changing in a control-policy-dependent manner. We highlight the following key contributions:

- We introduce the notion of performative control, where the deployed control policy affects the underlying dynamics of the control system. We provide sufficient conditions for the existence and uniqueness of the performative stable control (PSC) solution. The sufficient condition is expressed in terms of a weighted sum of all the distributional sensitivities associated with the policy-dependent system state transition matrices over the entire control horizon. An interesting finding is that the sufficient condition exhibits a structure of sensitivity propagation and aggregation, implying that it is preferable for sensitivities to be relatively small in the early stages of the system state evolution.

- We analyze the impacts of system stability on the existence and uniqueness of the PSC solution. We show that when the policy-dependent dynamics are almost surely strongly stable, the PSC solution exists if the sum of all distributional sensitivities is below a certain threshold. On the other hand, when the policy-dependent dynamics are almost surely unstable, the proposed sufficient condition for the existence of the PSC solution will place a necessary requirement on a temporally backwards decaying of the distributional sensitivities.

- We propose a repeated stochastic gradient descent (RSGD) scheme and analyze its convergence towards the PSC solution. We show that the scheme is convergent under the same

sufficient condition for the existence of the PSC solution. With an appropriate step size rule, the expected squared distance between the PSC solution and the iterates decays as $\mathcal{O}\left(\frac{1}{N}\right)$, where $N$ is the iteration number.

Finally, we conduct experiments on policy-dependent stock investment risk minimization problem. The numerical results validate the effectiveness of our algorithm and theoretical analysis.

### 1.1 Related Work

**Performative Prediction.** The notion of performative prediction is initiated by [21], where performative stability is first introduced, and a sufficient and necessary condition is provided so that the performative stable point can be reached via iterative risk minimization algorithms. Since then, there has been a growing literature analyzing the performative prediction problem and studying the convergence of learning algorithms to the performative stable point [18, 5, 3, 13, 29].There are also some other papers that find algorithms that converge to a stationary solution of the performative risk [12, 19, 25]. Our problem poses entirely new challenges because the decision-dependent data of system state and control costs also have decision-dependent temporal correlations. Such a two-layer decision-dependent structure cannot be incorporated into the existing performative prediction framework.

There are some very recent works on *performative reinforcement learning*, where a Markov decision process (MDP) is considered and the deployed policy not only changes the costs but also the underlying state transition kernel [16, 24]. However, these works only considered the tabular MDP, where the state and action space are finite. The LDS considered in our work may also be viewed as a Markov decision process with a decision-dependent linear transition kernel and decision-dependent costs. But both the state and action space are continuous and there are infinitely many admissible state-action pairs, which are beyond the scopes of [16, 24].

**Stateful Performative Prediction.** Note that most of the existing performative prediction works [21, 18, 18, 5, 13, 29, 5, 3, 13, 29] are non-stateful in the sense that, for any deployed policy $\theta$, the data sample $Z$ follows a static distribution $\mathcal{D}\left(\theta\right)$, i.e., $Z \sim \mathcal{D}\left(\theta\right)$. The seminal work [3] generalizes the stateless framework of [21] and proposes a more general framework of stateful performative prediction via a stateful distribution transition map $f\left(\cdot\right)$. Specifically, the observed data distribution in [3] is time-varying and depends on the history of previously deployed polices, i.e., $\mathcal{D}_{t+1} = f\left(\mathcal{D}_t, \theta_t\right)$. Consequently, in comparison to the conventional non-stateful performative prediction works [21, 18, 18, 5, 13, 29, 5, 3, 13, 29], this framework is capable of encapsulating the phenomenon of strategic decision-making with outdated information, and serves as a foundation for investigations of the disparate effects of performativity (please refer to Examples 1-3 in [3] for more details). The interconnections and generalizations between our work and the stateful performative prediction will be substantiated in Section 2.

**Nonstochastic Control.** Another line of relevant works pertains to non-stochastic control, which is initiated by [1]. Various applications of non-stochastic control can be found in [10, 1, 11, 8]. At the core of non-stochastic control is the disturbance-action control policy, which chooses the action as a linear map of the past disturbances [11]. Such a disturbance-action policy facilitates efficient algorithms for control problems with arbitrary additive disturbances in the dynamics and arbitrary convex control costs instead of quadratic costs only. In this paper, we adopt the disturbance-action control policy for analysis. However, in contrast to [1, 10, 1, 11, 8], where the system dynamics are static, our analysis involves policy-dependent nonstationary dynamics.

## 2 Motivations

In this section, we discuss the key motivations for our performative linear control framework. We also outline the key connections and generalizations of our proposed framework to the stateful performative prediction framework of [3].

**Connections between Static Stateful Distribution Transition Maps and LDS.** The LDS modeling via control theory in our work is closely aligned with the static stateful distribution maps within the state performative prediction framework [3]. Let us first consider a static stateful distribution transition map $f\left(\cdot\right)$ with $\mathcal{D}_{t+1} = f\left(\mathcal{D}_t, \theta_t\right), \forall t \geq 0$. To facilitate the analysis, we use a performative data sample $Z_t \sim \mathcal{D}_t$ to equivalently characterize the performative distribution $\mathcal{D}_t$. If the

properties of $f(\cdot)$ are nice enough, then from the perspective of random variables, there will exist a corresponding mapping $\widetilde{f}(\cdot)$ such that

$$Z_{t+1} \stackrel{\mathrm{d}}{=} \widetilde{f}(Z_t, \theta_t), Z_{t+1} \sim \mathcal{D}_{t+1}, Z_t \sim \mathcal{D}_t, \forall t \geq 0, \tag{1}$$

where $\stackrel{\mathrm{d}}{=}$ denotes equal in distribution. Linearizing $\widetilde{f}(\cdot)$ at some equilibrium point $(\overline{Z}, \overline{\theta})$ with $\overline{Z} \sim \overline{\mathcal{D}}$ and ignoring the higher order terms will lead to

$$Z_{t+1} \stackrel{\mathrm{d}}{=} \mathbf{A}Z_t + \mathbf{B}\theta_t + \mathbf{w}, \tag{2}$$

where $\mathbf{A} = \left[\frac{\partial \widetilde{f}}{\partial d}\right]_{(\overline{d}, \overline{\theta})}$ and $\mathbf{B} = \left[\frac{\partial \widetilde{f}}{\partial \theta}\right]_{(\overline{d}, \overline{\theta})}$ are the static Jacobin matrices at the equilibrium point $(\overline{d}, \overline{\theta})$ and $\mathbf{w} = \widetilde{f}(\overline{Z}, \overline{\theta}) - \mathbf{A}\overline{Z} - \mathbf{B}\overline{\theta}$. Consider the performative data sample $Z_t$, the deployed policy $\theta_t$ and the residual $\mathbf{w}$ in (2) as the state variable $\mathbf{x}_t$, the control input action $\mathbf{u}_t$ and the additive system noise $\mathbf{w}_t \sim \mathbf{w}$ of an LDS, respectively. The linearized static stateful distribution map model in (2) is thus precisely connected to a LDS with system dynamics given by

$$\mathbf{x}_{t+1} = \mathbf{A}\mathbf{x}_t + \mathbf{B}\mathbf{u}_t + \mathbf{w}_t. \tag{3}$$

**Extension to Performative Stateful Distribution Transition Maps via Performative LDS.** Further consider a more complicated case of *performative distribution transition maps* $f_{\theta_t}(\cdot)$ with $\mathcal{D}_{t+1} = f_{\theta_t}(\mathcal{D}_t, \theta_t), \forall t \geq 0$, where the specific form of the distribution transition map $f_{\theta_t}$ is time-varying and depends on the delployed policy $\theta_t$. Employing a similar linearization for $\widetilde{f}_{\theta_t}(\cdot)$ and neglecting the higher order terms, we obtain

$$Z_{t+1} \stackrel{\mathrm{d}}{=} \mathbf{A}_{\theta_t} Z_t + \mathbf{B}_{\theta_t}\theta_t + \mathbf{w}_{\theta_t}, \tag{4}$$

where $\mathbf{A}_{\theta_t} = \left[\frac{\partial \widetilde{f}_{\theta_t}}{\partial d}\right]_{(\overline{d}, \overline{\theta})}$ and $\mathbf{B}_{\theta_t} = \left[\frac{\partial \widetilde{f}_{\theta_t}}{\partial \theta}\right]_{(\overline{d}, \overline{\theta})}$ are the performative Jacobin matrices, and $\mathbf{w}_t = \widetilde{f}_{\theta_t}(\overline{Z}, \overline{\theta}) - \mathbf{A}_{\theta_t}\overline{Z} - \mathbf{B}_{\theta_t}\overline{\theta}$. This results in an equivalent performative LDS given by

$$\mathbf{x}_{t+1} = \mathbf{A}_{\mathbf{u}_t}\mathbf{x}_t + \mathbf{B}_{\mathbf{u}_t}\mathbf{u}_t + \mathbf{w}_{\mathbf{u}_t}. \tag{5}$$

In contrast to (3), where the system dyanmics are static, the state transition matrix $\mathbf{A}_{\mathbf{u}_t}$, control input gain matrix $\mathbf{B}_{\mathbf{u}_t}$ and additive noise $\mathbf{w}_{\mathbf{u}_t}$ in (5) are all performative and depend on the deployed control policy $\mathbf{u}_t$.

In conclusion, the LDS modeling presented in our paper allows for the *performative transition maps of performative distributions*, thereby extending the technical results previously obtained for fixed performative distribution transition maps in the stateful performative prediction work [3]. As a result, our proposed performative LDS framework has the potential to enhance the understanding of general performative prediction.

## 3  Problem Setup

We consider the control of a linear dynamic system with per stage cost $c_t(\mathbf{x}_t, \mathbf{u}_t)$. A control policy $\pi$ is a mapping $\pi : \mathbb{R}^{d_x \times 1} \to \mathbb{R}^{d_u \times 1}$, which maps the system state $\mathbf{x}_t$ to the control action $\mathbf{u}_t$, i.e., $\mathbf{u}_t = \pi(\mathbf{x}_t)$. For each control policy $\pi$, we attribute a finite time horizon expected cost defined as

$$C_T^\pi = \mathbb{E}_{\mathbf{x}_0, \{\mathbf{A}_t\}, \{\mathbf{w}_t\}} \left[\sum_{t=0}^{T} c_t(\mathbf{x}_t, \mathbf{u}_t)\right], \tag{6}$$

where $\mathbf{x}_{t+1} = \mathbf{A}_t\mathbf{x}_t + \mathbf{B}\mathbf{u}_t + \mathbf{w}_t$, the initial system state $\mathbf{x}_0$ follows a general distribution of $\mathcal{D}_{x_0}$ with bounded support $\|\mathbf{x}_0\| \leq x_0$, and $\mathbb{E}_{\mathbf{x}_0, \{\mathbf{A}_t\}, \{\mathbf{w}_t\}}$ represents the expectation over $\mathbf{x}_0$, the entire policy-dependent state transition matrix sequence $\{\mathbf{A}_t, 1 \leq t \leq T\}$ and the entire disturbance sequence $\{\mathbf{w}_t, 1 \leq t < T\}$.

To the best of our knowledge, this paper is the very first work to investigate the performative LDS, and we intend to provide a thorough theoretical investigation and aim at establishing various new theoretical results. Therefore, we opt to construct our performative LDS theoretical framework upon

the performative state transition matrix $\mathbf{A}_t$, while maintaining the control input gain matrix $\mathbf{B}$ and additive disturbance $\mathbf{w}_t$ as non-performative. This facilitates us to streamline the theoretical analysis and consolidates the system design insights.

We have the following assumption on the additive disturbances $\{\mathbf{w}_t, 1 \leq t \leq T\}$.

**A1** *The additive disturbance per time step $\mathbf{w}_t$ is bounded, i.i.d, and zero-mean with a lower bounded covariance i.e., $\mathbf{w}_t \sim \mathcal{D}_\mathbf{w}, \mathbb{E}[\mathbf{w}_t] = \mathbf{0}, \mathbb{E}[\mathbf{w}_t \mathbf{w}_t^\top] \succeq \sigma^2 \mathbf{I}, \|\mathbf{w}_t\| \leq W, \forall 0 \leq t < T$, and $\mathbb{E}[\mathbf{w}_{t_1} \mathbf{w}_{t_2}^\top] = \mathbf{0}, \forall 0 \leq t_1 \neq t_2 < T$.*

**Disturbance-Action Control Policy.** We work with the following class of disturbance-action control policy throughout this paper, which is commonly used in nonstochastic control [10, 1, 11, 8] to address general convex control cost functions.

**Definition 1** *(Disturbance-Action Policy). For a disturbance-action control policy, the mapping $\pi$, $\forall 0 \leq t < T$, is uniquely characterized by a set of matrices $\{\mathbf{M}^{(1)}, \cdots, \mathbf{M}^{(H)}\}$. At every time step $t$, such a disturbance-action control policy assigns a control action $\mathbf{u}_t^{(\mathbf{M})}$ in the form of*

$$\mathbf{u}_t^{(\mathbf{M})} = \pi\left(\mathbf{x}_t\right) = -\mathbf{K}\mathbf{x}_t + \sum_{i=1}^{H} \mathbf{M}^{(i)} \mathbf{w}_{t-i} = -\mathbf{K}\mathbf{x}_t + \mathbf{M}\left[\mathbf{w}\right]_{t-1}^H, \tag{7}$$

*where $\mathbf{M} = \left[\mathbf{M}^{(1)}, \mathbf{M}^{(2)}, \cdots, \mathbf{M}^{(H)}\right]$ belongs to a convex set $\mathbb{M}$ with bounded support $\|\mathbf{M}\|_F \leq M$, $[\mathbf{w}]_{t-1}^H$ is short for $\left[\mathbf{w}_{t-1}^\top, \cdots \mathbf{w}_{t-1-H}^\top\right]^\top$, $H < T$ is a constant and $\mathbf{w}_i = \mathbf{0}$ for all $i < 0$.*

In the disturbance-action policy (7), we adopt a class of linear controller $\mathbf{K}$ defined as follows.

**Definition 2** *(Strongly Stabilizing Linear Controller [1, 10, 11, 8]). Given $\mathbf{A}$ and $\mathbf{B}$, a linear controller $\mathbf{K}$ is $(\kappa, \gamma)$ almost surely strongly stable for real numbers $\kappa \geq 1, \gamma < 1$, if $\|\mathbf{K}\| \leq \kappa$, and there exists matrices $\mathbf{Q}$ and $\mathbf{L}$ such that $\widetilde{\mathbf{A}} := \mathbf{A} - \mathbf{B}\mathbf{K} := \mathbf{Q}\mathbf{L}\mathbf{Q}^{-1}$, with $\|\mathbf{L}\| \leq 1 - \gamma$ and $\|\mathbf{Q}\|, \|\mathbf{Q}^{-1}\| \leq \kappa$.*

It is worth noting that the disturbance-action policy is only parameterized by the matrix $\mathbf{M}$. Whereas the state feedback gain $\mathbf{K}$, which is a fixed matrix, is not part of the parameterization of the policy. As pointed out in [2], a typical choice of the parameter $H$ is $H = \gamma^{-1} \log(T\kappa^2)$. With an appropriate choice of the policy $\mathbf{M}$, the control action $\mathbf{u}_t^{(\mathbf{M})}$ in (7) is capable of approximating any linear state feedback control policy in terms of the total cost suffered with a finite time horizon of $H$ [10, 1, 11, 8].

**Policy-dependent Dynamics.** Without loss of generality, at any time step $t$, the impact of the disturbance-action control policy $\mathbf{u}_t^{(\mathbf{M})}$ to the dynamics of the linear system is modeled as a policy-dependent additive perturbation $\boldsymbol{\Delta}_t$ to a common state transition matrix $\mathbf{A}$.

**A2** *(Policy-dependent State Transition Matrix). The disturbance-action policy-dependent state transition matrix $\mathbf{A}_t$ takes the form of*

$$\mathbf{A}_t = \mathbf{A} + \boldsymbol{\Delta}_t, \boldsymbol{\Delta}_t \sim \mathcal{D}_t\left(\mathbf{M}\right), \forall 0 \leq t < T, \tag{8}$$

*where $\mathbf{A}$ is the mean value of $\mathbf{A}_t$, and $\boldsymbol{\Delta}_t$ is the policy-dependent state transition perturbation with zero mean and bounded support, i.e., $\mathbb{E}\left[\boldsymbol{\Delta}_t\right]=\mathbf{0}$ and there exists a bounded constant $\xi_t$ such that $\|\boldsymbol{\Delta}_t\| \leq \xi_t, \forall 0 \leq t < T$. For different time steps $t_1$ and $t_2$, $\boldsymbol{\Delta}_{t_1}$ and $\boldsymbol{\Delta}_{t_2}$ are mutually independent. Besides, $\{\boldsymbol{\Delta}_t, 0 \leq t < T\}$ and $\{\mathbf{w}_t, 0 \leq t < T\}$ are mutually independent.*

**Remark 1** *(Non-zero Mean $\boldsymbol{\Delta}_t$). If the mean of the disturbance is non-zero and time-varying, i.e., $\mathbb{E}\left[\boldsymbol{\Delta}_t\right] = \Theta_t$, we will have a new equivalent $\mathbf{A}_t' = \mathbf{A} + \Theta_t$ with zero mean disturbances. We only need to choose a new linear controller $\mathbf{K}_t'$ such that $\mathbf{A}_t' - \mathbf{B}\mathbf{K}_t'$ is $(\kappa_t, \gamma_t)$-strongly stabilizing. As a result, without loss of generality, we assume $\boldsymbol{\Delta}_t$ is zero mean throughout this paper.*

**Remark 2** *(Policy-dependent Control Input Gain Matrix $\mathbf{B}$). Note that under the DAP (7), the disturbance-action is given by $\mathbf{BM}[\mathbf{w}]_{t-1}^{H}$, which is linear in policy $\mathbf{M}$. Such kind of linear disturbance-action control policy has various nice theoretical performance guarantees as substantiated in [10, 1, 11, 8]. On the other hand, if $\mathbf{B}$ is also performative, we will have a generalized disturbance-action $\mathbf{B}_t(\mathbf{M})\mathbf{M}[\mathbf{w}]_{t-1}^{H}$, which can be possibly nonlinear in policy $\mathbf{M}$. Such kind of generalized nonlinear disturbance-action control policy has received very few research attention, and it can serve as a very interesting future research direction.*

We make the following sensitivity assumption on the distributions $\{\mathcal{D}_t(\mathbf{M}), 0 \leq t < T\}$.

**A3** *($\varepsilon$-Sensitivity). For any $t = 0, 1, \cdots, T-1$, there exists a constant $\varepsilon_t > 0$ such that*

$$\mathcal{W}^1\left(\mathcal{D}_t(\mathbf{M}), \mathcal{D}_t(\mathbf{M}')\right) \leq \varepsilon_t \left\|\mathbf{M} - \mathbf{M}'\right\|_F, \ \forall \mathbf{M}, \mathbf{M}' \in \mathbb{M}, \tag{9}$$

*where $\mathcal{W}^1(\mathcal{D}, \mathcal{D}')$ denotes the Wasserstein-1 distance between the distributions $\mathcal{D}$ and $\mathcal{D}'$.*

Assumption A3 imposes a regularity requirement on the distributions $\{\mathcal{D}_t(\mathbf{M}), 0 \leq t < T\}$. Intuitively, if the disturbance-action control policies are made according to similar policy parameterizations $\mathbf{M}$, then the resulting distributions of the policy-dependent state transition perturbations should also be similar.

**System State Evolutions.** Under the disturbance-action policy (7), the following lemma shows that the system state $\mathbf{x}_t, \forall 1 \leq t \leq T$, can be uniquely determined by $\mathbf{x}_0$, $\widetilde{\mathbf{A}}$, $\mathbf{M}$, $\{\boldsymbol{\Delta}_t, 0 \leq t < T\}$ and $\{\mathbf{w}_t, -H \leq t < T\}$.

**Lemma 1** *Given a disturbance-action policy $\mathbf{M}$, the system state $\mathbf{x}_t$, $\forall 1 \leq t \leq T$, can be represented as*

$$\mathbf{x}_t = \mathbf{x}_t^{(\mathbf{M})} = \prod_{i=0}^{t-1}\left(\widetilde{\mathbf{A}} + \boldsymbol{\Delta}_i\right)\mathbf{x}_0 + \sum_{i=0}^{t-1}\prod_{j=i+1}^{t-1}\left(\mathbf{1}_{j<t}\left(\widetilde{\mathbf{A}} + \boldsymbol{\Delta}_j\right) + \mathbf{1}_{j=t}\mathbf{I}\right)\mathbf{BM}\left[\mathbf{w}\right]_{i-1}^{H} \tag{10}$$

$$+ \sum_{i=0}^{t-1}\prod_{j=i+1}^{t-1}\left(\mathbf{1}_{j<t}\left(\widetilde{\mathbf{A}} + \boldsymbol{\Delta}_j\right) + \mathbf{1}_{j=t}\mathbf{I}\right)\mathbf{w}_i.$$

*Besides, let the $\mathbf{K}$ in DAP (7) be a strongly stabilizing linear controller, the norm of $\mathbf{x}_t$ is upper bounded as $\|\mathbf{x}_t\| \leq x_0\kappa^2\alpha_t + \kappa^2 W\left(\|\mathbf{B}\|HM + 1\right)\beta_t$, where $\alpha_t = \prod_{i=0}^{t-1}\left(1 - \gamma + \kappa^2\xi_i\right)$ and $\beta_t = \sum_{i=0}^{t-1}\prod_{j=i+1}^{t-1}\left(\mathbf{1}_{j<t}\left(1 - \gamma + \kappa^2\xi_j\right) + \mathbf{1}_{j=t}\right)$.*

The *performative optimal control (POC)* problem can therefore be formulated as:

$$\min_{\mathbf{M}\in\mathbb{M}} \quad C_T^{\mathbf{M}} = \mathbb{E}_{\mathbf{x}_0, \{\boldsymbol{\Delta}_t \sim \mathcal{D}_t(\mathbf{M})\}, \{\mathbf{w}_t\}}\left[\sum_{t=0}^{T}c_t\left(\mathbf{x}_t^{(\mathbf{M})}, \mathbf{u}_t^{(\mathbf{M})}\right)\right]. \tag{11}$$

The *POC* problem (11) comprises of a stochastic objective function with policy-dependent distributions. Due to non-convexity, the performative optimal solution $\mathbf{M}^{PO}$ to (11) is usually difficult to obtain. Alternatively, in this paper, we are interested in the *performative stable control (PSC)* solution:

$$\mathbf{M}^{PS} = \Phi(\mathbf{M}^{PS}) := \arg\min_{\mathbf{M}\in\mathbb{M}}\mathbb{E}_{\mathbf{x}_0, \{\boldsymbol{\Delta}_t \sim \mathcal{D}_t(\mathbf{M}^{PS})\}, \{\mathbf{w}_t\}}\left[\sum_{t=0}^{T}c_t\left(\mathbf{x}_t^{(\mathbf{M})}, \mathbf{u}_t^{(\mathbf{M})}\right)\right]. \tag{12}$$

Notice that $\mathbf{M}^{PS}$ is defined to be a fixed point of the map $\Phi$. Compared to the *POC* (11), the distribution of $\boldsymbol{\Delta}_t$ in *PSC* (12) changes from $\boldsymbol{\Delta}_t \sim \mathcal{D}_t(\mathbf{M})$ to $\boldsymbol{\Delta}_t \sim \mathcal{D}_t(\mathbf{M}^{PS}), \forall 0 \leq t < T$. The existence and uniqueness of $\mathbf{M}^{PS}$ will be dicsussed in Lemma 4.

**Comparison to Existing Works.** Most of the existing performative prediction works [21, 13, 14, 5, 3, 18, 29, 25, 12, 19, 25] consider the cost in the form of $l(\theta; Z)$, where $\theta$ is the decision variable. Then the relationship between data samples $Z$ and decision $\theta$ is parameterized

**Algorithm 1** Repeated Stochastic Gradient Descent (RSGD)

---

**Input:** Step sizes $\{\eta_n, 0 \leq n \leq N\}$, parameters $\mathbf{K}, H$. Define $\mathbb{M} = \{\mathbf{M} : \|\mathbf{M}\| \leq M\}$. Initialize $\mathbf{M}_0 \in \mathbb{M}$ arbitrarily.

1: **for** $n = 0, \cdots, N$, **do**
2:     Initialize $\nabla J_T = \mathbf{0}$.
3:     **for** $t = 0, \cdots, T-1$, **do**
4:         Use control $\mathbf{u}_t = -\mathbf{K}\mathbf{x}_t + \mathbf{M}_n [\mathbf{w}]_{t-1}^H$.
5:         Observe $\mathbf{A}_t, \mathbf{x}_{t+1}$; compute noise $\mathbf{w}_t = \mathbf{x}_{t+1} - \mathbf{A}_t\mathbf{x}_t - \mathbf{B}\mathbf{u}_t$.
6:         Compute the gradient $\nabla_{\mathbf{M}_n} c_t (\mathbf{x}_t, \mathbf{u}_t)$ and update $\nabla J_T \leftarrow \nabla J_T + \nabla_{\mathbf{M}_n} c_t (\mathbf{x}_t, \mathbf{u}_t)$.
7:     **end for**
8:     Update $\mathbf{M}_{n+1} \leftarrow \text{Proj}_{\mathbb{M}}\{\mathbf{M}_n - \eta_n \nabla J_T\}$.
9: **end for**

---

by a fixed distribution $Z \sim \mathcal{D}(\theta)$ with fixed sensitivity $\varepsilon$. However, in our work, the relationship between system state data samples $\mathbf{x}_t$ and policy $\mathbf{M}$ is characterized by a time-varying distribution $\mathbf{x}_t \sim f_t (\mathcal{D}_0(\mathbf{M}), \cdots, \mathcal{D}_{t-1}(\mathbf{M}))$ with a time-varying sequence of joint sensitivities $\{\varepsilon_0, \cdots, \varepsilon_t\}, \forall 0 \leq t < T$. The total cost $\sum_{t=0}^{T} c_t$ depends on all the policy-dependent distributions $\{\mathcal{D}_0(\mathbf{M}), \cdots, \mathcal{D}_{T-1}(\mathbf{M})\}$ with a collection of sensitivities $\{\varepsilon_0, \cdots, \varepsilon_{T-1}\}$. These key differences lead to a more complicated analysis in our work.

**RSGD Scheme.** We propose a repeated stochastic gradient descent (RSGD) scheme in Algorithm 1 to find a PSC solution to (12). The metric of the projection in line 9 of Algorithm 1 is the matrix Frobenius norm. Specifically, $\text{Proj}_{\mathbb{M}}\{\mathbf{M}\} = \arg\min_{\mathbf{M}' \in \mathbb{M}} \|\mathbf{M} - \mathbf{M}'\|_F^2$. Note that such a projection is computationally tractable because the Frobenius norm square minimization is a convex optimization problem. The RSGD scheme first computes the stochastic gradient of the total cost w.r.t. policy $\sum_{t=0}^{T} \nabla_{\mathbf{M}_n} c_t (\mathbf{x}_t, \mathbf{u}_t)$ and then perform stochastic gradient descent on $\mathbf{M}$. The detailed steps for computation of the stochastic gradient $\nabla_{\mathbf{M}_n} c_t (\mathbf{x}_t, \mathbf{u}_t)$ are provided in Appendix B.

## 4 Main Results

This section investigates the existence of a PSC solution $\mathbf{M}^{PS}$ to (12) and the convergence of the RSGD scheme to $\mathbf{M}^{PS}$. We require the following assumptions on the per stage cost $c_t$ [1, 10, 1, 11, 8].

**A4** *(Strongly Convex). The per stage cost function $c_t (\mathbf{x}, \mathbf{u})$ is $\mu$-strongly convex such that*

$$c_t (\mathbf{x}_1, \mathbf{u}_1) \geq c_t (\mathbf{x}_2, \mathbf{u}_2) + \nabla_{\mathbf{x}}^\top c_t (\mathbf{x}_2, \mathbf{u}_2) (\mathbf{x}_1 - \mathbf{x}_2) + \nabla_{\mathbf{u}}^\top c_t (\mathbf{x}_2, \mathbf{u}_2) (\mathbf{u}_1 - \mathbf{u}_2)$$
$$+ \frac{\mu}{2} \left( \|\mathbf{x}_1 - \mathbf{x}_2\|^2 + \|\mathbf{u}_1 - \mathbf{u}_2\|^2 \right), \forall \mathbf{x}_1, \mathbf{x}_2 \in \mathbb{R}^{d_x}, \mathbf{u}_1, \mathbf{u}_2 \in \mathbb{R}^{d_u}.$$

**A5** *(Smoothness). The per stage cost function $c_t (\mathbf{x}, \mathbf{u})$ is $\varsigma$ smooth such that*

$$\|\nabla_{\mathbf{x}} c_t (\mathbf{x}_1, \mathbf{u}_1) - \nabla_{\mathbf{x}} c_t (\mathbf{x}_2, \mathbf{u}_1)\| + \|\nabla_{\mathbf{u}} c_t (\mathbf{x}_1, \mathbf{u}_1) - \nabla_{\mathbf{u}} c_t (\mathbf{x}_1, \mathbf{u}_2)\|$$
$$\leq \varsigma (\|\mathbf{x}_1 - \mathbf{x}_2\| + \|\mathbf{u}_1 - \mathbf{u}_2\|), \forall \mathbf{x}_1, \mathbf{x}_2 \in \mathbb{R}^{d_x}, \mathbf{u}_1, \mathbf{u}_2 \in \mathbb{R}^{d_u}.$$

**A6** *(Boundedness). There exists a positive constant $G$ such that*

$$\|\nabla_{\mathbf{x}} c_t (\mathbf{x}, \mathbf{u})\|, \|\nabla_{\mathbf{u}} c_t (\mathbf{x}, \mathbf{u})\| \leq GD, \forall \|\mathbf{x}\|, \|\mathbf{u}\| \leq D.$$

The above Assumptions A4-A6 on the per stage cost function $c_t (\mathbf{x}, \mathbf{u})$ are quite standard, which hold for broad classes of costs such as the quadratic costs.

To facilitate our discussions, we define an expected total cost with distribution shift as

$$C_T (\mathbf{M}; \mathbf{M}') := \mathbb{E}_{\mathbf{x}_0, \{\boldsymbol{\Delta}_t \sim \mathcal{D}_t(\mathbf{M}')\}, \{\mathbf{w}_t\}} \left[ \sum_{t=0}^{T} c_t \left( \mathbf{x}_t^{(\mathbf{M})}, \mathbf{u}_t^{(\mathbf{M})} \right) \right],$$

where, under policy $\mathbf{M}$, the distribution of policy-dependent perturbation is changed from $\boldsymbol{\Delta}_t \sim \mathcal{D}_t(\mathbf{M})$ to $\boldsymbol{\Delta}_t \sim \mathcal{D}_t(\mathbf{M}')$, $\forall 0 \leq t < T$. For the rest of this paper, unless otherwisespecified, $\nabla C_T(\mathbf{M}; \mathbf{M}')$ denote the gradients taken w.r.t. the first argument $\mathbf{M}$.

Our main results rely on the strong convexity of the expected total cost $C_T(\mathbf{M}; \mathbf{M}')$ with respect to its first argument $\mathbf{M}$. However, the strong convexity of the per stage cost function $c_t(\mathbf{x}, \mathbf{u})$ over the state-action space in Assumption A4 does not by itself imply the strong convexity of the expected total cost $C_T(\mathbf{M}; \mathbf{M}')$ over the space of policies $\mathbf{M}$. This is becasue the policy $\mathbf{M}$, which maps from a space of dimensionality $H \times d_x \times d_u$ to that of $d_x + d_u$, is not necessarily full column-rank. Our next lemma, which forms the core of our analysis, shows that this is not the case using the inherent stochastic nature of the policy-dependent dynamics.

**Lemma 2** *Under A1-A6, fix any $\mathbf{M}' \in \mathbb{M}$, the expected total cost $C_T(\mathbf{M}; \mathbf{M}')$ is $\widetilde{\mu}$-strongly convex in its first argument $\mathbf{M}$ such that $\forall \mathbf{M}_1, \mathbf{M}_2 \in \mathbb{M}$,*

$$
C_T(\mathbf{M}_1; \mathbf{M}') \geq C_T(\mathbf{M}_2; \mathbf{M}') + \mathrm{Tr}\left( (\nabla C_T(\mathbf{M}_2; \mathbf{M}'))^\top (\mathbf{M}_1 - \mathbf{M}_2) \right) + \frac{\widetilde{\mu}}{2} \|\mathbf{M}_1 - \mathbf{M}_2\|_F^2,
$$
(13)

*where $\widetilde{\mu} = \min\left\{ \frac{(T-H+1)\mu\sigma^2}{2}, \frac{(T-H+1)\mu\sigma^2\gamma^2}{64\kappa^{10}} \right\}$.*

For a concise presentation of the smoothness of expected total control cost, we next define a collection of constants as follows:

$$
c_1 := d_x \varsigma H^{\frac{3}{2}} W \left(1 + (\kappa^2 + \kappa^3)\|\mathbf{B}\|\right)(\kappa^2 + \kappa^3)(1 - \gamma)^{-1},
$$

$$
c_2 := d_x H^{\frac{3}{2}} W G (1 - \gamma)^{-1}(\kappa^4 + \kappa^5)\|\mathbf{B}\|, c_3 := (HM\|\mathbf{B}\| + 1)W, c_4 := HW(1 - \gamma)c_1,
$$

$$
c_5 := HW(\kappa^2 + \kappa^3)^{-1}(1 - \gamma)c_1.
$$

The smoothness of the expected total cost $C_T(\mathbf{M}; \mathbf{M}')$ is summarized below.

**Lemma 3** *Under A1-A6, the expected total cost $C_T$ is smooth in the sense that, for any $\mathbf{M}, \mathbf{M}', \mathbf{M}_1, \mathbf{M}_2 \in \mathbb{M}$ and $\forall 1 \leq t \leq T$, the following inequality holds*

$$
\|\nabla C_T(\mathbf{M}_1; \mathbf{M}) - \nabla C_T(\mathbf{M}_2; \mathbf{M}')\|_F
$$
$$
\leq \sum_{t=1}^{T} \lambda_t \|\mathbf{M}_1 - \mathbf{M}_2\|_F + \sum_{t=0}^{T-1}\left(\varepsilon_t \sum_{i=t+1}^{T} \nu_i\right)\|\mathbf{M} - \mathbf{M}'\|_F,
$$
(14)

*where $\lambda_t = c_1(c_4\beta_t + c_5), \nu_t = (c_1 + c_2\beta_t)(x_0\alpha_t + c_3\beta_t), \forall 1 \leq t \leq T$, and we recall that $\alpha_t = \prod_{i=0}^{t-1}(1 - \gamma + \kappa^2\xi_i)$ and $\beta_t = \sum_{i=0}^{t-1}\prod_{j=i+1}^{t-1}\left(\mathbf{1}_{j<t}(1 - \gamma + \kappa^2\xi_j) + \mathbf{1}_{j=t}\right)$ from Lemma 1, which characterize the growth of the norm of system state $\|\mathbf{x}_t\|$.*

**Existence and Uniqueness of $\mathbf{M}^{PS}$.** Our first main result establishes a sufficient condition for the existence and uniqueness of the performative stable policy $\mathbf{M}^{PS}$ that solves the *PSC* problem (12).

**Lemma 4** *Under A1-A6, consider the fixed-point iteration*

$$
\mathbf{M}_{n+1} = \Phi(\mathbf{M}_n), \forall n \geq 0,
$$
(15)

*where the map $\Phi$ is defined in (12). If the following condition is satisfied*

$$
\sum_{t=0}^{T-1}\left(\varepsilon_t \sum_{i=t+1}^{T} \nu_i\right) < \widetilde{\mu},
$$
(16)

*then iterates $\mathbf{M}_n$ converge to a unique performatively stable point $\mathbf{M}^{PS}$ at a linear rate, i.e.,*

$$
\|\mathbf{M}_n - \mathbf{M}^{PS}\|_F \leq \rho \text{ for } n \geq \left(1 - \frac{\sum_{t=0}^{T-1}\left(\varepsilon_t \sum_{i=t+1}^{T} \nu_i\right)}{\widetilde{\mu}}\right)^{-1} \log\left(\frac{1}{\rho}\|\mathbf{M}_0 - \mathbf{M}^{PS}\|_F\right).
$$

The sufficient condition (16) delivers a fact that that the existence and uniqueness of $\mathbf{M}^{PS}$ is jointly determined by all the sensitivities $\{\varepsilon_t, 0 \le t < T\}$ in the temporal domain. The sensitivity $\varepsilon_t$ at the $t$-th time step is propagated starting from time step $t + 1$ to the last time step $T$ with a sequence of weights $\{\nu_{t+1}, \cdots, \nu_T\}$. The aggregated impact of all the policy-dependent disturbances $\{\boldsymbol{\Delta}_t, 0 \le t < T\}$ is captured by total sum in L.H.S. of (16). This is very different from the existing performative prediction works [21, 18, 5, 3, 13, 29, 12, 19, 25], where only one distribution $\mathcal{D}$ and one sensitivity $\varepsilon$ are involved.

The sufficient condition (16) also implies that it is preferable for the initial policy-dependent disturbance $\boldsymbol{\Delta}_0$ to be small because it propagates and aggregates for the longest time steps of $T$.

**Impacts of System Stability.** The sufficient condition (16) also reveals the impacts of system stability on the existence and uniqueness of the performative stable solution $\mathbf{M}^{PS}$, which are summarized below.

**Proposition 1** *(Almost Surely Strongly Stable Case) Let A1-A6 hold. If policy-dependent state transition matrix $\mathbf{A}_t$ is $\left(\kappa, \gamma - \kappa^2 \xi_t\right)$-strongly stable for real numbers $\kappa \ge 1, \gamma - \kappa^2 \xi_t < 1, \forall 0 \le t < T$, almost surely. Let $\zeta = \max\left\{1 - \gamma + \kappa^2 \xi_t, \forall 0 \le t < T\right\}$, then $\mathbf{M}^{PS}$ exists and is unique if $\sum_{t=0}^{T-1} \varepsilon_t < \phi\left(1 - \frac{H}{T}\right)\overline{\mu}$, where $\overline{\mu} = \min\left\{\frac{\mu\sigma^2}{2}, \frac{\mu\sigma^2\gamma^2}{64\kappa^{10}}\right\}$ and $\phi$ is some positive constant.*

Proposition 1 points out that when the policy-dependent dynamics are almost surely strongly stable, we only need to make sure that the sum of all the sensitivities $\{\varepsilon_t, 0 \le t < T\}$ is below a certain threshold.

**Proposition 2** *(Almost Surely Unstable Case) Let A1-A6 hold. If the policy-dependent state transition matrix $\mathbf{A}_t$ is almost surely unstable, i.e., there exists a positive constant $\widetilde{\zeta} > 1$ such that $\widetilde{\zeta} \le \|\mathbf{A}_t\| \le 1 - \gamma + \kappa^2 \xi_t, \forall 0 \le t < T$. In this case, to guarantee the sufficient condition (16) can be satifised, we must have $\varepsilon_t < \frac{\overline{\phi}(T-H+1)\overline{\mu}}{\widetilde{\zeta}^{T-t}-1}, \forall 0 \le t < T$, where $\overline{\phi}$ is some positive constant.*

For unstable policy-dependent dynamics, condition (16) will impose a necessary requirement on the temporally backwards decaying of the sensitivities. Particularly, more restrictive requirements are placed on the the sensitivies in the early time steps, i.e., smallt, which should decay exponentially fast w.r.t. the control time horzion $T$.

For more general applications, where $\mathbf{A}_t$ can be either stable or unstable for different time steps, the weighted sum requirement of the sensitivities $\{\varepsilon_t, 0 \le t < T\}$ in (16) is sufficient to guarantee the existence and uniqueness of the performative stable solution $\mathbf{M}^{PS}$.

**Convergence of RSGD Scheme.** Our next theorem establishes the convergence rate of the proposed RSGD algorithm.

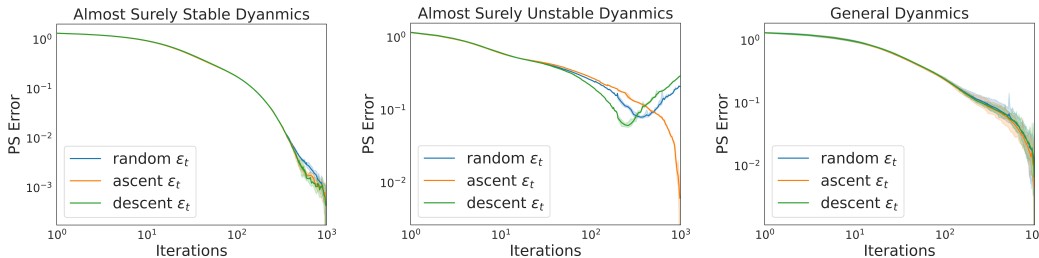

**Figure 1:** PS Error $\left\|\mathbf{M}_N - \mathbf{M}^{PS}\right\|_F^2$

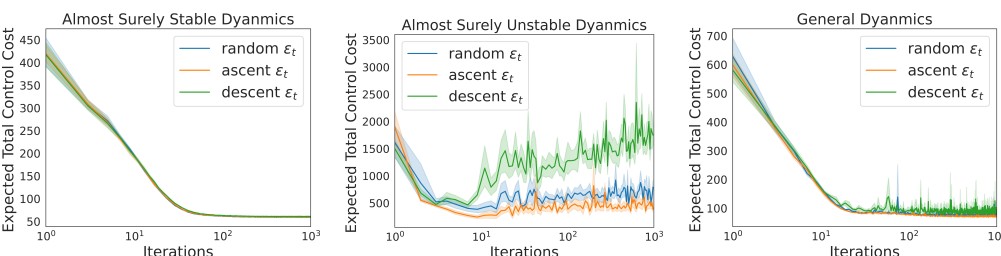

**Figure 2:** Expected Total Control Cost $C_T\left(\mathbf{M}_N; \mathbf{M}_N\right)$

---

**Theorem 1** *Choose two positive constants $\phi_1 > 0$ and $\phi_2 \geq 1$ such that the following two conditions are satisfied simualtneously*

$$\frac{\phi_1}{\phi_2} \leq \min\left\{ \frac{\widetilde{\mu} - \sum_{t=0}^{T-1}\left(\varepsilon_t \sum_{i=t+1}^{T} \nu_i\right)}{2\left(\sum_{t=1}^{T} \lambda_t + \sum_{t=0}^{T-1}\left(\varepsilon_t \sum_{i=t+1}^{T} \nu_i\right)\right)^2}, \frac{1}{\widetilde{\mu} - \sum_{t=0}^{T-1}\left(\varepsilon_t \sum_{i=t+1}^{T} \nu_i\right)} \right\}, \quad (17)$$

$$\frac{\phi_1}{1 + \frac{1}{\phi_2}} \geq \frac{2}{\widetilde{\mu} - \sum_{t=0}^{T-1}\left(\varepsilon_t \sum_{i=t+1}^{T} \nu_i\right)}. \quad (18)$$

*Consider a sequence of non-negative step sizes $\left\{\eta_n = \frac{\phi_1}{n+\phi_2}, n \geq 0\right\}$. Then, the iterates generated by RSGD admit the following bound for any $N \geq 1$:*

$$\mathbb{E}\left[\left\|\mathbf{M}_N - \mathbf{M}^{PS}\right\|_F^2\right] \leq \mathrm{e}^{-\sum_{n=1}^{N} \frac{\phi_1}{n}\left(\widetilde{\mu} - \sum_{t=0}^{T-1}\left(\varepsilon_t \sum_{i=t+1}^{T} \nu_i\right)\right)} \mathbb{E}\left[\left\|\mathbf{M}_0 - \mathbf{M}^{PS}\right\|_F^2\right] + \frac{\phi_3}{N}, \quad (19)$$

*where $\vartheta_t = \kappa^3 G\left(\left(HW + \kappa^2\right)\kappa \|\mathbf{B}\| \beta_t + 1\right)\left(x_0 \alpha_t + c_3 \beta_t\right) + GHWM\left(\kappa^3 \beta_t + 1\right), \forall 1 \leq t \leq T$, and $\phi_3 = \frac{4\phi_1 T \sum_{t=1}^{T} \vartheta_t^2}{\widetilde{\mu} - \sum_{t=0}^{T-1}\left(\varepsilon_t \sum_{i=t+1}^{T} \nu_i\right)}$ is a positive constant.*

---

Note that, under the sufficient condition (16) in Lemma 4, there always exists a pair of $(\phi_1, \phi_2)$ that satisfies (17) and (18) simultaneously by letting $\phi_2$ be sufficiently large. The first term on the R. H. S. of (19) decays at the rate of $\mathcal{O}(\mathrm{e}^{-\sum_{n=1}^{N} \frac{\phi_1}{n}\left(\widetilde{\mu} - \sum_{t=0}^{T-1}\left(\varepsilon_t \sum_{i=t+1}^{T} \nu_i\right)\right)})$ and is scaled by the initial error $\mathbb{E}\left[\left\|\mathbf{M}_0 - \mathbf{M}^{PS}\right\|_F^2\right]$. The second term is a fluctuation term that only depends on the variance of the stochastic gradient, which decays at the rate $\mathcal{O}\left(1/N\right)$. For more general types of step sizes and the associated nonasymptotic convergence rate analysis, please refer to Lemma 5 in the appendix G.1.

## 5 Numerical Experiments

We consider an application of stock investment risk minimization problem to verify our algorithm and theoretical results. Consider an investor trading a total number of 10 stocks over a period of $T = 60$ trading days. The detailed system setups are described in Example 1 in Appendix

A of the supplementary. We compare the performative error (PS error) $\left\|\mathbf{M}_N - \mathbf{M}^{PS}\right\|_F^2$ and the expected total cost $C_T\left(\mathbf{M}_N;\mathbf{M}_N\right)$ against the iteration number $N$, respectively. We consider a fixed distributional sensitivity value set $\boldsymbol{\varepsilon} = \{\varepsilon_i, i = 0, 1, \ldots, T-1\}$. We assign three different patterns of the sensitivity sequence as $\varepsilon_{\mathrm{d}} = \mathrm{descend}\left(\boldsymbol{\varepsilon}\right)$, $\varepsilon_{\mathrm{a}} = \mathrm{ascend}\left(\boldsymbol{\varepsilon}\right)$ and $\varepsilon_{\mathrm{r}} = \mathrm{random}\left(\boldsymbol{\varepsilon}\right)$ in descending, ascending, and random order, respectively, over the time steps. We first observe from Figure 1 (left) and Figure 2 (left) that when the policy-dependent system dynamics $\mathbf{A}_t, \forall 0 \le t < T$, are almost surely strongly stable, the gap $\left\|\mathbf{M}_N - \mathbf{M}^{PS}\right\|_F^2$ of three different patterns $\varepsilon_{\mathrm{d}}$, $\varepsilon_{\mathrm{a}}$ and $\varepsilon_{\mathrm{r}}$ all decay at $\mathcal{O}\left(\frac{1}{N}\right)$ as $N \to \infty$, and the expected total control cost also converges. These coincide exactly with Proposition 1 and Theorem 1, where the existence of $\mathbf{M}^{PS}$ only requires that the sum of the distributional sensitivities $\sum_{t=0}^{T-1} \varepsilon_t$ is small enough, and the temporal order of each $\varepsilon_t$ is negligible. We next observe from Figure 1 (middle) and Figure 2 (middle), that when the $\mathbf{A}_t, \forall 0 \le t < T$, are almost surely unstable, the iterates $\mathbf{M}_n$ of both $\varepsilon_{\mathrm{d}}$ and $\varepsilon_{\mathrm{r}}$ diverge. The expected total control costs associated with $\varepsilon_{\mathrm{d}}$ and $\varepsilon_{\mathrm{r}}$ are significantly larger than that of $\varepsilon_{\mathrm{a}}$. This is because large initial distributional sensitivities in $\varepsilon_{\mathrm{d}}$ and $\varepsilon_{\mathrm{r}}$ rule out the existence of $\mathbf{M}^{PS}$, which matches Proposition 2. For the general case of $\mathbf{A}_t$ in Figure 1 (right) and Figure 2 (right), all three cases converge due to the relatively mild sufficient condition (16).

## Conclusion

In this work, we have introduced the framework of performative control and studied the conditions under which a PSC policy exists. We have analyzed the impact of system stability on the existence of the PSC policy, and proposed a condition on the sum of the distributional sensitivities and the temporally backwards decaying of sensitivities for almost surely strongly stable and almost surely unstable systems, respectively. We have also proposed an RSGD algorithm that converges to the PSC policy in a mean-square sense. The extension of our current results to general control policies and general control costs [cf. A4], will be explored in future work.

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

# A   Application Example and Experiment Detail.

In this section, we elaborate a concrete example of stock market risk minimization to justify the policy-dependent state transition model in (8). We also conduct numerical experiment based on this example to demonstrate the efficacy of our developed theory.

**Example 1** *(Stock Market Risk Minimization). We describe a risk minimization problem for stock market investment to illustrate the application of (11). Consider an investor trading a total number of $L$ stocks over a period of $T$ trading days. The observed market price $s_t^{(l)}$ of the $n$-th stock at the $t$-th day follows a stochastic volatility model of $\log s_{t+1}^{(l)} = \log s_t^{(l)} + \frac{r - \frac{1}{2}\left(v_t^{(l)}\right)^2}{T} + \frac{v_t^{(l)}}{\sqrt{T}}, s_1^{(l)} > 0, \forall 1 \le t < T, 1 \le l \le L$, where $r > 0$ is the riskless interest rate per day, $v_t^{(l)}$ and $s_1^{(l)}$ are the unobservable independent random volatility process and the constant initial stock price associated with the $l$-th stock, respectively [22, 27]. Let $q_t^{(l)}$ and $q_1^{(l)}$ be the the total return at the $t$-th day before the market opens and the initial investment associated with the $l$-th stock, respectively. The investor maintains a portfolio for each $q_t^{(l)}$, which is parameterized by a row vector $\mathbf{m}^{(l)} = \left[m^{l,1}, \cdots, m^{l,M}\right]$ with $m^{l,i} \in [0,1], \forall 1 \le i \le L$, being the weight of allocation and $\sum_{i=1}^{L} m^{l,i} = 1$. Specifically, at the $t$-th day when the market opens, the investor immediately allocates $q_t^{(l)}$ proportionally to buy the $N$ stocks according to the weight vector $\mathbf{m}^{(l)}$. The total systemic risk associated with $q_t^l$ during the period between the $t$-th day right before market opening and the $(t+1)$-th day right before market opening is $\sum_{i=1}^{L}\left(m^{l,i}w_t^{(i)}h\right) + 1 \cdot w_{t+1}^{(l)}(24 - h)$, where $w_t^i$ is the per hour risk associated with the $i$-th stock in the $t$-th day and $h$ is the total number of trading hours per day. is $\sum_{i=1}^{L} m^{l,i} e^{\left(\left(r - \frac{1}{2}\left(v_t^{(i)}\right)^2\right)\frac{1}{T} + v_t^{(i)}\frac{1}{\sqrt{T}}\right)} q_t^l$. The investor then spends all this amount to buy back the $l$-th stock and then the $t$-th trading day ends. Each stock in the portfolio suffers from a proportionate random i.i.d. systemic risk $w_t^i$ per hour, i.e., holding the $i$-th stock with a ratio of $m^{l,i}$ in a portfolio for a total number of $h$ trading hours in the $t$-th day will incur a systemic risk of $m^{l,i}w_t^{(i)}h$. Denote the mean-shifted return as $\mathbf{r}_t = \mathbf{q}_t - \mathbb{E}\left[\mathbf{q}_t\right]$ and let $\mathbf{x}_t = \begin{bmatrix} \mathbf{r}_t \\ \mathbf{q}_t \end{bmatrix}$.*

*The evolution of $\mathbf{x}_t$ can be characterized by the canonical form of*

$$\mathbf{x}_{t+1} = \left(\mathbf{A} + \boldsymbol{\Delta}_t\right)\mathbf{x}_t + \mathbf{u}_t^{(\mathbf{M})} + \widetilde{\mathbf{w}}_t, \tag{20}$$

$$\mathbf{u}_t^{(\mathbf{M})} = \mathbf{M}\widetilde{\mathbf{w}}_{t-1} \ \ with \ \ \mathbf{B} = \mathbf{I}, \mathbf{K} = \mathbf{0}, \forall 1 \le t < T. \tag{21}$$

*where $\mathbf{M} = \frac{h}{24-h}\left[\mathbf{m}^{(1)}; \cdots; \mathbf{m}^{(L)}\right]$, $\mathbf{w}_{t-1} = (24 - h)\left[w_t^{(1)}, \cdots, w_t^{(L)}\right]^T$, $\widetilde{\mathbf{w}}_t = \begin{bmatrix} \mathbf{w}_t - \mathbb{E}\left[\mathbf{w}_t\right] \\ \mathbf{w}_t \end{bmatrix}$, the state transition matrix $\mathbf{A} = \mathbf{I}_{2L \times 2L}$ and the policy-dependent state transition perturbation $\boldsymbol{\Delta}_t$ is given by (22) and (23).*

$$\boldsymbol{\Delta}_t = \begin{bmatrix} \mathbb{E}\left[\mathbf{V}_t^{(\mathbf{M})}\right] - \mathbf{I}_{L \times L} & \mathbf{V}_t^{(\mathbf{M})} - \mathbb{E}\left[\mathbf{V}_t^{(\mathbf{M})}\right] \\ \mathbf{0}_{L \times L} & \mathbf{V}_t^{(\mathbf{M})} - \mathbf{I}_{L \times L} \end{bmatrix}, \tag{22}$$

$$\mathbf{V}_t^{(\mathbf{M})} = \text{diag}\left(\left[\sum_{i=1}^{L} m^{1,i} e^{\left(\frac{r - \frac{1}{2}\left(v_t^{(i)}\right)^2}{T} + \frac{v_t^{(i)}}{\sqrt{T}}\right)}, \cdots, \sum_{i=1}^{L} m^{L,i} e^{\left(\frac{r - \frac{1}{2}\left(v_t^{(i)}\right)^2}{T} + \frac{v_t^{(i)}}{\sqrt{T}}\right)}\right]\right). \tag{23}$$

*The investment risk minimization problem can thus be casted as*

$$\min_{\mathbf{M} \in \mathbb{M}} C_T^{\mathbf{M}} = \mathbb{E}_{\{\boldsymbol{\Delta}_t\}, \{\widetilde{\mathbf{w}}_t\}}\left[\sum_{t=1}^{T}\left(\left\|[\mathbf{I}_L, \mathbf{0}_{L \times L}] \cdot \mathbf{x}_t\right\|^2 + \left\|[\mathbf{I}_L, \mathbf{0}_{L \times L}] \cdot \mathbf{u}_t^{(\mathbf{M})}\right\|^2\right)\right], \quad s.t. \ (20) - (23).$$

*Tackling (11) leads to an optimized stock investment policy $\mathbf{M}$ which takes the effects of policy-dependent random perturbations to the dynamics of stock returns into account.*

In the simulation, we set the number of stocks $L = 10$ and the number of trading days $T = 60$. The initial policy $\mathbf{M}_0$ is randomly chosen within the feasible set $\mathbb{M} = \{\mathbf{M} : \sum_{i=1}^{10} m^{l,i} = 1, \forall 1 \leq l \leq 10\}$. The entries of noise term $\mathbf{w}_t$ are independently and uniformly drawn from the interval $[0, 1]$. For each $1 \leq i \leq 10$ and each $0 \leq t < 60$, we first obtain $\tilde{v}_t^{(i)}$ by sampling from a Gaussian distribution, i.e., $\tilde{v}_t^{(i)} \sim \mathcal{N}(\log(\varepsilon_t), 0.2)$, where $\varepsilon_t$ denotes the sensitivity at $t$-th trading day. The $\tilde{v}_t^{(i)}$ is then projected to the interval $[-0.6, 0.6]$ to obtain $v_t^{(i)}$ in (23). We let the total number iterations $N = 1000$, and the stepsize $\eta_n$ in Algorithm 1 is set to be 0.01, $\forall 0 \leq n \leq 1000$.

The sensitivities are shown as follows:

**Sensitivities with Ascending Sequence.** $\varepsilon_{\mathrm{a}}$=[1.25797477e-07 5.03189910e-07 1.13217730e-06 2.01275964e-06 3.14493693e-06 4.52870919e-06 6.16407639e-06 8.05103855e-06 1.01895957e-05 1.25797477e-05 1.52214948e-05 1.81148367e-05 2.12597737e-05 2.46563056e-05 2.83044324e-05 3.22041542e-05 3.63554710e-05 4.07583827e-05 4.54128893e-05 5.03189910e-05 4.66004175e-03 5.35796616e-03 6.12231163e-03 6.95609731e-03 7.86234234e-03 8.84406585e-03 9.90428699e-03 1.10460249e-02 1.22722987e-02 1.35861276e-02 1.49905306e-02 1.64885270e-02 1.80831358e-02 1.97773762e-02 2.15742674e-02 2.34768284e-02 2.54880785e-02 2.76110367e-02 2.98487222e-02 3.22041542e-02 4.33504397e-02 4.66004175e-02 5.00089002e-02 5.35796616e-02 5.73164756e-02 6.12231163e-02 6.53033575e-02 6.95609731e-02 7.39997371e-02 7.86234234e-02 8.34358059e-02 8.84406585e-02 9.36417552e-02 9.90428699e-02 1.04647776e-01 1.10460249e-01 1.16484061e-01 1.22722987e-01 1.29180801e-01 1.35861276e-01]

**Sensitivities with Descending Sequence.** $\varepsilon_{\mathrm{d}}$=[1.35861276e-01 1.29180801e-01 1.22722987e-01 1.16484061e-01 1.10460249e-01 1.04647776e-01 9.90428699e-02 9.36417552e-02 8.84406585e-02 8.34358059e-02 7.86234234e-02 7.39997371e-02 6.95609731e-02 6.53033575e-02 6.12231163e-02 5.73164756e-02 5.35796616e-02 5.00089002e-02 4.66004175e-02 4.33504397e-02 3.22041542e-02 2.98487222e-02 2.76110367e-02 2.54880785e-02 2.34768284e-02 2.15742674e-02 1.97773762e-02 1.80831358e-02 1.64885270e-02 1.49905306e-02 1.35861276e-02 1.22722987e-02 1.10460249e-02 9.90428699e-03 8.84406585e-03 7.86234234e-03 6.95609731e-03 6.12231163e-03 5.35796616e-03 4.66004175e-03 5.03189910e-05 4.54128893e-05 4.07583827e-05 3.63554710e-05 3.22041542e-05 2.83044324e-05 2.46563056e-05 2.12597737e-05 1.81148367e-05 1.52214948e-05 1.25797477e-05 1.01895957e-05 8.05103855e-06 6.16407639e-06 4.52870919e-06 3.14493693e-06 2.01275964e-06 1.13217730e-06 5.03189910e-07 1.25797477e-07]

**Sensitivities with Random Sequence.** $\varepsilon_{\mathrm{r}}$=[2.54880785e-02 3.63554710e-05 2.34768284e-02 1.64885270e-02 1.25797477e-07 9.36417552e-02 1.10460249e-02 6.16407639e-06 2.76110367e-02 5.00089002e-02 1.16484061e-01 1.04647776e-01 2.15742674e-02 2.46563056e-05 3.14493693e-06 4.66004175e-03 4.54128893e-05 3.22041542e-02 1.22722987e-02 4.33504397e-02 8.84406585e-03 6.95609731e-03 5.03189910e-05 1.01895957e-05 7.39997371e-02 1.97773762e-02 7.86234234e-02 1.35861276e-02 8.84406585e-02 1.22722987e-01 4.07583827e-05 8.05103855e-06 2.98487222e-02 9.90428699e-03 6.95609731e-02 8.34358059e-02 6.12231163e-03 1.81148367e-05 1.52214948e-05 2.12597737e-05 6.53033575e-02 5.03189910e-07 5.35796616e-03 9.90428699e-02 7.86234234e-03 1.10460249e-01 4.66004175e-02 2.83044324e-05 3.22041542e-05 2.01275964e-06 1.13217730e-06 6.12231163e-02 5.35796616e-02 1.49905306e-02 1.25797477e-05 1.80831358e-02 4.52870919e-06 1.35861276e-01 1.29180801e-01 5.73164756e-02]

The non-ordered lists of sensitivity values in epsilon $\varepsilon_{\mathrm{a}}$,$\varepsilon_{\mathrm{d}}$, and $\varepsilon_{\mathrm{r}}$ are the same. The state trajectory $\mathbf{x}_t$ and control action $\mathbf{u}_t$ are generated according to (20)- (23) based on the above system parameter configurations.

## B  Auxiliariy Lemmas for Per Stage Cost Function

In this part, we provide several lemmas to characterize the gradient properties and the smoothness of the per stage cost $c_t$. We first summarize a collection of notations. We denote $\nabla_{\mathbf{x}} c_t \left( \mathbf{x}_t^{(\mathbf{M})}, \mathbf{u}_t^{(\mathbf{M})} \right)$ and $\nabla_{\mathbf{u}} c_t \left( \mathbf{x}_t^{(\mathbf{M})}, \mathbf{u}_t^{(\mathbf{M})} \right)$ as the gradient of $c_t$ w.r.t. the system state $\mathbf{x}_t^{(\mathbf{M})}$ and control action

$\mathbf{u}_t^{(\mathbf{M})}$, respectively. We denote $\nabla_{\mathbf{M}} c_t \left( \mathbf{x}_t^{(\mathbf{M})}, \mathbf{u}_t^{(\mathbf{M})} \right)$ as the gradient of $c_t$ w.r.t. the variable $\mathbf{M}$ in $\mathbf{x}_t^{(\mathbf{M})}$ and $\mathbf{u}_t^{(\mathbf{M})}$ under any given realization of the policy-dependent state transition perturbations $\{ \boldsymbol{\Delta}_i, 0 \leq i < t \}$.

Based on the chain rule and the relationship between $\mathbf{M}$ and the pair $(\mathbf{x}_t, \mathbf{u}_t)$ in (7) and (10), the expression of $\nabla_{\mathbf{M}} c_t \left( \mathbf{x}_t^{(\mathbf{M})}, \mathbf{u}_t^{(\mathbf{M})} \right)$ is given by

$$
\begin{aligned}
\nabla_{\mathbf{M}} c_t &\left( \mathbf{x}_t^{(\mathbf{M})}, \mathbf{u}_t^{(\mathbf{M})} \right) \\
&= \sum_{i=0}^{t-1} \left( \prod_{j=i+1}^{t-1} \left( \mathbf{1}_{j<t} \left( \widetilde{\mathbf{A}} + \boldsymbol{\Delta}_j \right) + \mathbf{1}_{j=t} \mathbf{I} \right) \mathbf{B} \right)^{\top} \nabla_{\mathbf{x}} c_t \left( \mathbf{x}_t^{(\mathbf{M})}, \mathbf{u}_t^{(\mathbf{M})} \right) \left( [\mathbf{w}]_{i-1}^H \right)^{\top} \\
&\quad - \sum_{i=0}^{t-1} \left\{ \left( \mathbf{K} \prod_{j=i+1}^{t-1} \left( \mathbf{1}_{j<t} \left( \widetilde{\mathbf{A}} + \boldsymbol{\Delta}_j \right) + \mathbf{1}_{j=t} \mathbf{I} \right) \mathbf{B} \right)^{\top} \nabla_{\mathbf{u}} c_t \left( \mathbf{x}_t^{(\mathbf{M})}, \mathbf{u}_t^{(\mathbf{M})} \right) \left( [\mathbf{w}]_{i-1}^H \right)^{\top} \right\} \\
&\quad + \nabla_{\mathbf{u}} c_t \left( \mathbf{x}_t^{(\mathbf{M})}, \mathbf{u}_t^{(\mathbf{M})} \right) \left( [\mathbf{w}]_{t-1}^H \right)^{\top}.
\end{aligned}
\tag{24}
$$

For the ease of notation, without ambiguity we also occasionally use the notation $c_t \left( \mathbf{M}; \{ \boldsymbol{\Delta}_i \}_{0 \leq i < t} \right)$ to denote $c_t \left( \mathbf{x}_t^{(\mathbf{M})}, \mathbf{u}_t^{(\mathbf{M})} \right)$ under a sequence of policy-dependent state transition perturbations $\{ \boldsymbol{\Delta}_i, 0 \leq i < t \}$. In this case, $\nabla_{\mathbf{M}} c_t \left( \mathbf{M}; \{ \boldsymbol{\Delta}_i \}_{0 \leq i < t} \right)$ denote the gradient taken w.r.t. the first argument $\mathbf{M}$.

Moreover, the total cost function can be represented as

$$
J_T \left( \mathbf{M}; \{ \boldsymbol{\Delta}_t \}_{0 \leq t < T} \right) = \sum_{t=0}^{T} c_t \left( \mathbf{M}; \{ \boldsymbol{\Delta}_i \}_{0 \leq i < t} \right).
\tag{25}
$$

It is clear that

$$
C_T \left( \mathbf{M}; \mathbf{M}' \right) = \mathbb{E}_{\mathbf{x}_0, \{ \boldsymbol{\Delta}_t \sim \mathcal{D}_t(\mathbf{M}') \}_{0 \leq i < T}, \{ \mathbf{w}_i \}_{0 \leq i < T}} J_T \left( \mathbf{M}; \{ \boldsymbol{\Delta}_t \}_{0 \leq t < T} \right).
\tag{26}
$$

Unless otherwise specified, $\nabla J_T \left( \mathbf{M}; \{ \boldsymbol{\Delta}_t \}_{0 \leq t < T} \right)$ and $\nabla C_T \left( \mathbf{M}; \mathbf{M}' \right)$ denotes the gradient taken w.r.t. the first argument $\mathbf{M}$.

**Convexity.** Denote the per stage expected cost

$$
f_t \left( \mathbf{M}; \mathbf{M}_1 \right) = \mathbb{E}_{\mathbf{x}_0, \{ \boldsymbol{\Delta}_i \sim \mathcal{D}_i(\mathbf{M}_1) \}_{0 \leq i < t}, \{ \mathbf{w}_i \}_{0 \leq i < t}} \left[ c_t \left( \mathbf{M}; \{ \boldsymbol{\Delta}_i \}_{0 \leq i < t} \right) \right], \forall 1 \leq t \leq T,
$$

where the distribution of policy-dependent perturbation is changed from $\boldsymbol{\Delta}_i \sim \mathcal{D}_i(\mathbf{M})$ to $\boldsymbol{\Delta}_i \sim \mathcal{D}_i(\mathbf{M}_1), \forall 0 \leq i < t$.

We have the following lemma characterizing the convexity of $f_t \left( \mathbf{M}; \mathbf{M}_1 \right), \forall 1 \leq t \leq T$.

**Lemma B.1** *The per stage expected cost $f_t \left( \mathbf{M}; \mathbf{M}' \right)$ is a convex function of $\mathbf{M}$, $\forall \mathbf{M}, \mathbf{M}' \in \mathbb{M}$, $\forall 1 \leq t \leq T$.*

*Proof.* We prove this lemma following the convex property of the function $c_t(\mathbf{x}, \mathbf{u})$ in Assumption A 4. Let $0 < \theta < 1$ and $\mathbf{M}_1, \mathbf{M}_2, \mathbf{M}' \in \mathbb{M}$. Consder the weighted policy $\theta \mathbf{M}_1 + (1 - \theta) \mathbf{M}_2$, we

have

$$\mathbf{x}_t^{(\theta\mathbf{M}_1+(1-\theta)\mathbf{M}_2)} = \prod_{i=0}^{t-1}\left(\widetilde{\mathbf{A}}+\boldsymbol{\Delta}_i\right)\mathbf{x}_0 + \sum_{i=0}^{t-1}\prod_{j=i+1}^{t-1}\left(\mathbf{1}_{j<t}\left(\widetilde{\mathbf{A}}+\boldsymbol{\Delta}_j\right)+\mathbf{1}_{j=t}\mathbf{I}\right)\mathbf{B} \quad (27)$$

$$\cdot\left(\theta\mathbf{M}_1+(1-\theta)\mathbf{M}_2\right)\mathbf{M}\left[\mathbf{w}\right]_{i-1}^{H} + \sum_{i=0}^{t-1}\prod_{j=i+1}^{t-1}\left(\mathbf{1}_{j<t}\left(\widetilde{\mathbf{A}}+\boldsymbol{\Delta}_j\right)+\mathbf{1}_{j=t}\mathbf{I}\right)\mathbf{w}_i$$

$$= \theta\left(\prod_{i=0}^{t-1}\left(\widetilde{\mathbf{A}}+\boldsymbol{\Delta}_i\right)\mathbf{x}_0 + \sum_{i=0}^{t-1}\prod_{j=i+1}^{t-1}\left(\mathbf{1}_{j<t}\left(\widetilde{\mathbf{A}}+\boldsymbol{\Delta}_j\right)+\mathbf{1}_{j=t}\mathbf{I}\right)\mathbf{B}\mathbf{M}_1\left[\mathbf{w}\right]_{i-1}^{H}\right.$$

$$\left. + \sum_{i=0}^{t-1}\prod_{j=i+1}^{t-1}\left(\mathbf{1}_{j<t}\left(\widetilde{\mathbf{A}}+\boldsymbol{\Delta}_j\right)+\mathbf{1}_{j=t}\mathbf{I}\right)\mathbf{w}_i\right) + (1-\theta)$$

$$\cdot\left(\prod_{i=0}^{t-1}\left(\widetilde{\mathbf{A}}+\boldsymbol{\Delta}_i\right)\mathbf{x}_0 + \sum_{i=0}^{t-1}\prod_{j=i+1}^{t-1}\left(\mathbf{1}_{j<t}\left(\widetilde{\mathbf{A}}+\boldsymbol{\Delta}_j\right)+\mathbf{1}_{j=t}\mathbf{I}\right)\mathbf{B}\mathbf{M}_2\left[\mathbf{w}\right]_{i-1}^{H}\right.$$

$$\left. + \sum_{i=0}^{t-1}\prod_{j=i+1}^{t-1}\left(\mathbf{1}_{j<t}\left(\widetilde{\mathbf{A}}+\boldsymbol{\Delta}_j\right)+\mathbf{1}_{j=t}\mathbf{I}\right)\mathbf{w}_i\right)$$

$$= \theta\mathbf{x}_t^{\mathbf{M}_1} + (1-\theta)\mathbf{x}_t^{\mathbf{M}_2}.$$

As a result,

$$f_t\left(\theta\mathbf{M}_1+(1-\theta)\mathbf{M}_2;\mathbf{M}'\right) \quad (28)$$

$$= \mathbb{E}_{\mathbf{x}_0,\{\boldsymbol{\Delta}_i\sim\mathcal{D}_i(\mathbf{M}')\}_{0\le i<t},\{\mathbf{w}_i\}_{0\le i<t}}\left[c_t\left(\mathbf{x}_t^{(\mathbf{M}_1+(1-\theta)\mathbf{M}_2)},\mathbf{u}_t^{(\mathbf{M}_1+(1-\theta)\mathbf{M}_2)}\right)\right]$$

$$= \mathbb{E}_{\mathbf{x}_0,\{\boldsymbol{\Delta}_i\sim\mathcal{D}_i(\mathbf{M}')\}_{0\le i<t},\{\mathbf{w}_i\}_{0\le i<t}}\left[c_t\left(\theta\mathbf{x}_t^{\mathbf{M}_1}+(1-\theta)\mathbf{x}_t^{\mathbf{M}_2},\theta\mathbf{u}_t^{\mathbf{M}_1}+(1-\theta)\mathbf{u}_t^{\mathbf{M}_2}\right)\right]$$

$$\overset{(28.a)}{\le} \mathbb{E}_{\mathbf{x}_0,\{\boldsymbol{\Delta}_i\sim\mathcal{D}_i(\mathbf{M}')\}_{0\le i<t},\{\mathbf{w}_i\}_{0\le i<t}}\left[\theta c_t\left(\mathbf{x}_t^{\mathbf{M}_1},\mathbf{u}_t^{\mathbf{M}_1}\right)+(1-\theta)c_t\left(\mathbf{x}_t^{\mathbf{M}_2},\mathbf{u}_t^{\mathbf{M}_2}\right)\right]$$

$$= \theta f_t\left(\mathbf{M}_1;\mathbf{M}'\right)+(1-\theta)f_t\left(\mathbf{M}_2;\mathbf{M}'\right),$$

where inequality $(28.a)$ holds becasue of the convexity of $c_t(\mathbf{x},\mathbf{u})$. $\qquad\square$

We next have a key lemma characterizing the strong convexity of $f_t(\mathbf{M};\mathbf{M}'),\forall H\le t\le T$.

**Lemma B.2** *The per stage expected cost $f_t(\mathbf{M};\mathbf{M}')$ is a $\min\left\{\frac{\mu}{2},\frac{\mu\sigma^2\gamma^2}{64\kappa^{10}}\right\}$-strongly convex function of $\mathbf{M}$, $\forall\mathbf{M},\mathbf{M}'\in\mathbb{M},\forall H\le t\le T$.*

*Proof.* We prove this lemma following the strong convexity of the function $c_t(\mathbf{x},\mathbf{u})$ in Assumption A 4. Let $\mathbf{M}_1,\mathbf{M}_2,\mathbf{M}'\in\mathbb{M}$. Consider a time step $t$ such that $H\le t\le T$. For any given realizations of $\{\boldsymbol{\Delta}_i\}_{0\le i<t}$, where $\boldsymbol{\Delta}_i\sim\mathcal{D}_i(\mathbf{M}'),\forall 0\le i<t$, we have

$$c_t\left(\mathbf{x}_t^{(\mathbf{M}_1)},\mathbf{u}_t^{(\mathbf{M}_1)}\right)-c_t\left(\mathbf{x}_t^{(\mathbf{M}_2)},\mathbf{u}_t^{(\mathbf{M}_2)}\right) \quad (29)$$

$$\overset{(29.a)}{\ge} \left[\begin{array}{c}\nabla_{\mathbf{x}}c_t\left(\mathbf{x}_t^{(\mathbf{M}_2)},\mathbf{u}_t^{(\mathbf{M}_2)}\right)\\\nabla_{\mathbf{u}}c_t\left(\mathbf{x}_t^{(\mathbf{M}_2)},\mathbf{u}_t^{(\mathbf{M}_2)}\right)\end{array}\right]^T\left[\begin{array}{c}\mathbf{x}_t^{(\mathbf{M}_1)}-\mathbf{x}_t^{(\mathbf{M}_2)}\\\mathbf{u}_t^{(\mathbf{M}_1)}-\mathbf{u}_t^{(\mathbf{M}_2)}\end{array}\right]+\frac{\mu}{2}\left\|\begin{array}{c}\mathbf{x}_t^{(\mathbf{M}_1)}-\mathbf{x}_t^{(\mathbf{M}_2)}\\\mathbf{u}_t^{(\mathbf{M}_1)}-\mathbf{u}_t^{(\mathbf{M}_2)}\end{array}\right\|^2,$$

where inequality $(29.a)$ holds because of the strong convexity assumption of $c_t$ in Assumption A4.

Based on the representations of the pair $(\mathbf{x}_t, \mathbf{u}_t)$ in (7) and (10), it follows that

$$\mathbf{x}_t^{(\mathbf{M}_1)} - \mathbf{x}_t^{(\mathbf{M}_2)} = \sum_{i=0}^{t-1} \prod_{j=i+1}^{t-1} \left( \mathbf{1}_{j<t} \left( \widetilde{\mathbf{A}} + \mathbf{\Delta}_j \right) + \mathbf{1}_{j=t}\mathbf{I} \right) \mathbf{B} \left( \mathbf{M}_1 - \mathbf{M}_2 \right) [\mathbf{w}]_{i-1}^{H} \tag{30}$$

$$\mathbf{u}_t^{(\mathbf{M}_1)} - \mathbf{u}_t^{(\mathbf{M}_2)} = -\mathbf{K} \sum_{i=0}^{t-1} \prod_{j=i+1}^{t-1} \left( \mathbf{1}_{j<t} \left( \widetilde{\mathbf{A}} + \mathbf{\Delta}_j \right) + \mathbf{1}_{j=t}\mathbf{I} \right) \mathbf{B} \left( \mathbf{M}_1 - \mathbf{M}_2 \right) [\mathbf{w}]_{i-1}^{H} \tag{31}$$

$$+ \left( \mathbf{M}_1 - \mathbf{M}_2 \right) [\mathbf{w}]_{t-1}^{H}.$$

Therefore, we obtain

$$\left[ \begin{array}{c} \nabla_{\mathbf{x}} c_t \left( \mathbf{x}_t^{(\mathbf{M}_2)}, \mathbf{u}_t^{(\mathbf{M}_2)} \right) \\ \nabla_{\mathbf{u}} c_t \left( \mathbf{x}_t^{(\mathbf{M}_2)}, \mathbf{u}_t^{(\mathbf{M}_2)} \right) \end{array} \right]^{T} \left[ \begin{array}{c} \mathbf{x}_t^{(\mathbf{M}_1)} - \mathbf{x}_t^{(\mathbf{M}_2)} \\ \mathbf{u}_t^{(\mathbf{M}_1)} - \mathbf{u}_t^{(\mathbf{M}_2)} \end{array} \right] \tag{32}$$

$$= \sum_{i=0}^{t-1} \nabla_{\mathbf{x}}^{\top} c_t \left( \mathbf{x}_t^{(\mathbf{M}_2)}, \mathbf{u}_t^{(\mathbf{M}_2)} \right) \left( \prod_{j=i+1}^{t-1} \left( \mathbf{1}_{j<t} \left( \widetilde{\mathbf{A}} + \mathbf{\Delta}_j \right) + \mathbf{1}_{j=t}\mathbf{I} \right) \mathbf{B} \left( \mathbf{M}_1 - \mathbf{M}_2 \right) [\mathbf{w}]_{i-1}^{H} \right)$$

$$+ \sum_{i=0}^{t-1} \nabla_{\mathbf{u}}^{\top} c_t \left( \mathbf{x}_t^{(\mathbf{M}_2)}, \mathbf{u}_t^{(\mathbf{M}_2)} \right) \left( -\mathbf{K} \sum_{i=0}^{t-1} \prod_{j=i+1}^{t-1} \left( \mathbf{1}_{j<t} \left( \widetilde{\mathbf{A}} + \mathbf{\Delta}_j \right) + \mathbf{1}_{j=t}\mathbf{I} \right) \mathbf{B} \left( \mathbf{M}_1 - \mathbf{M}_2 \right) [\mathbf{w}]_{i-1}^{H} \right.$$

$$+ \left( \mathbf{M}_1 - \mathbf{M}_2 \right) [\mathbf{w}]_{t-1}^{H} \bigg)$$

$$= \mathrm{Tr} \left( \left( \sum_{i=0}^{t-1} [\mathbf{w}]_{i-1}^{H} \nabla_{\mathbf{x}}^{\top} c_t \left( \mathbf{x}_t^{(\mathbf{M}_2)}, \mathbf{u}_t^{(\mathbf{M}_2)} \right) \prod_{j=i+1}^{t-1} \left( \mathbf{1}_{j<t} \left( \widetilde{\mathbf{A}} + \mathbf{\Delta}_j \right) + \mathbf{1}_{j=t}\mathbf{I} \right) \mathbf{B} \right) \left( \mathbf{M}_1 - \mathbf{M}_2 \right) \right)$$

$$+ \mathrm{Tr} \left( \left( -\sum_{i=0}^{t-1} [\mathbf{w}]_{i-1}^{H} \nabla_{\mathbf{u}}^{\top} c_t \left( \mathbf{x}_t^{(\mathbf{M}_2)}, \mathbf{u}_t^{(\mathbf{M}_2)} \right) \mathbf{K} \prod_{j=i+1}^{t-1} \left( \mathbf{1}_{j<t} \left( \widetilde{\mathbf{A}} + \mathbf{\Delta}_j \right) + \mathbf{1}_{j=t}\mathbf{I} \right) \mathbf{B} \right.$$

$$+ [\mathbf{w}]_{t-1}^{H} \nabla_{\mathbf{u}}^{\top} c_t \left( \mathbf{x}_t^{(\mathbf{M}_2)}, \mathbf{u}_t^{(\mathbf{M}_2)} \right) \bigg) \left( \mathbf{M}_1 - \mathbf{M}_2 \right) \right)$$

$$= \mathrm{Tr} \left( \nabla_{\mathbf{M}_2}^{\top} c_t \left( \mathbf{x}_t^{(\mathbf{M}_2)}, \mathbf{u}_t^{(\mathbf{M}_2)} \right) \left( \mathbf{M}_1 - \mathbf{M}_2 \right) \right).$$

Substitute (32) back into (29), we have

$$c_t \left( \mathbf{x}_t^{(\mathbf{M}_1)}, \mathbf{u}_t^{(\mathbf{M}_1)} \right) - c_t \left( \mathbf{x}_t^{(\mathbf{M}_2)}, \mathbf{u}_t^{(\mathbf{M}_2)} \right) \geq \mathrm{Tr} \left( \nabla_{\mathbf{M}_2}^{\top} c_t \left( \mathbf{x}_t^{(\mathbf{M}_2)}, \mathbf{u}_t^{(\mathbf{M}_2)} \right) \left( \mathbf{M}_1 - \mathbf{M}_2 \right) \right) \tag{33}$$

$$+ \frac{\mu}{2} \left( \left\| \mathbf{x}_t^{(\mathbf{M}_1)} - \mathbf{x}_t^{(\mathbf{M}_2)} \right\|^2 + \left\| \mathbf{u}_t^{(\mathbf{M}_1)} - \mathbf{u}_t^{(\mathbf{M}_2)} \right\|^2 \right).$$

Taking full expectation on both sides of (33), it follows

$$f_t \left( \mathbf{M}_1; \mathbf{M}' \right) - f_t \left( \mathbf{M}_1; \mathbf{M}' \right) \geq \mathrm{Tr} \left( \nabla^{\top} f_t \left( \mathbf{M}_2; \mathbf{M}' \right) \cdot \left( \mathbf{M}_1 - \mathbf{M}_2 \right) \right) \tag{34}$$

$$+ \frac{\mu}{2} \mathbb{E}_{\mathbf{x}_0, \{\mathbf{\Delta}_i \sim \mathcal{D}_i(\mathbf{M}')\}_{0 \leq i < t}, \{\mathbf{w}_i\}_{0 \leq i < t}} \left[ \left\| \mathbf{x}_t^{(\mathbf{M}_1)} - \mathbf{x}_t^{(\mathbf{M}_2)} \right\|^2 + \left\| \mathbf{u}_t^{(\mathbf{M}_1)} - \mathbf{u}_t^{(\mathbf{M}_2)} \right\|^2 \right].$$

We next analyze the last term in (34). Consider the conditional expectation on $\{\mathbf{w}_i\}_{0 \leq i < t}$ as follows

$$\mathbb{E}_{\mathbf{x}_0, \{\boldsymbol{\Delta}_i \sim \mathcal{D}_i(\mathbf{M}')\}_{0 \leq i < t}, \{\mathbf{w}_i\}_{0 \leq i < t}} \left[ \left\| \mathbf{x}_t^{(\mathbf{M}_1)} - \mathbf{x}_t^{(\mathbf{M}_2)} \right\|^2 + \left\| \mathbf{u}_t^{(\mathbf{M}_1)} - \mathbf{u}_t^{(\mathbf{M}_2)} \right\|^2 \right] \tag{35}$$

$$= \mathbb{E}_{\{\mathbf{w}_i\}_{0 \leq i < t}} \left[ \mathbb{E}_{\mathbf{x}_0, \{\boldsymbol{\Delta}_i \sim \mathcal{D}_i(\mathbf{M}')\}_{0 \leq i < t}} \left[ \left\| \mathbf{x}_t^{(\mathbf{M}_1)} - \mathbf{x}_t^{(\mathbf{M}_2)} \right\|^2 + \left\| \mathbf{u}_t^{(\mathbf{M}_1)} - \mathbf{u}_t^{(\mathbf{M}_2)} \right\|^2 \Big| \{\mathbf{w}_i\}_{0 \leq i < t} \right] \right]$$

$$\geq \mathbb{E}_{\{\mathbf{w}_i\}_{0 \leq i < t}} \left[ \left\| \mathbb{E}_{\mathbf{x}_0, \{\boldsymbol{\Delta}_i \sim \mathcal{D}_i(\mathbf{M}')\}_{0 \leq i < t}} \left[ \mathbf{x}_t^{(\mathbf{M}_1)} - \mathbf{x}_t^{(\mathbf{M}_2)} \Big| \{\mathbf{w}_i\}_{0 \leq i < t} \right] \right\|^2 \right.$$

$$+ \left. \left\| \mathbb{E}_{\mathbf{x}_0, \{\boldsymbol{\Delta}_i \sim \mathcal{D}_i(\mathbf{M}')\}_{0 \leq i < t}} \left[ \mathbf{u}_t^{(\mathbf{M}_1)} - \mathbf{u}_t^{(\mathbf{M}_2)} \Big| \{\mathbf{w}_i\}_{0 \leq i < t} \right] \right\|^2 \right]$$

$$\overset{(35.a)}{=} \mathbb{E}_{\{\mathbf{w}_i\}_{0 \leq i < t}} \left[ \left\| \sum_{i=0}^{t-1} \widetilde{\mathbf{A}}^i \mathbf{B} (\mathbf{M}_1 - \mathbf{M}_2) [\mathbf{w}]_{i-1}^H \right\|^2 \right.$$

$$+ \left. \left\| \mathbf{K} \sum_{i=0}^{t-1} \widetilde{\mathbf{A}}^i \mathbf{B} (\mathbf{M}_1 - \mathbf{M}_2) [\mathbf{w}]_{i-1}^H - (\mathbf{M}_1 - \mathbf{M}_2) [\mathbf{w}]_{t-1}^H \right\|^2 \right],$$

where $(35.a)$ holds becasue $\mathbb{E}[\boldsymbol{\Delta}_t] = \mathbf{0}, \forall 0 \leq t < T$.

We next consider each term on R.H.S. of $(35.a)$. We notice that

$$\text{vec} \left( \sum_{i=0}^{t-1} \widetilde{\mathbf{A}}^i \mathbf{B} (\mathbf{M}_1 - \mathbf{M}_2) [\mathbf{w}]_{i-1}^H \right) = \left( \sum_{i=0}^{t-1} \left( [\mathbf{w}]_{i-1}^H \right)^\top \otimes \left( \widetilde{\mathbf{A}}^i \mathbf{B} \right) \right) \text{vec} (\mathbf{M}_1 - \mathbf{M}_2) \tag{36}$$

Therefore,

$$\mathbb{E}_{\{\mathbf{w}_i\}_{0 \leq i < t}} \left[ \left\| \sum_{i=0}^{t-1} \widetilde{\mathbf{A}}^i \mathbf{B} (\mathbf{M}_1 - \mathbf{M}_2) [\mathbf{w}]_{i-1}^H \right\|^2 \right] \tag{37}$$

$$= \delta_{\mathbf{M}}^\top \mathbb{E}_{\{\mathbf{w}_i\}_{0 \leq i < t}} \left[ \left( \sum_{i=0}^{t-1} \left( [\mathbf{w}]_{i-1}^H \right)^\top \otimes \left( \widetilde{\mathbf{A}}^i \mathbf{B} \right) \right)^\top \left( \sum_{i=0}^{t-1} \left( [\mathbf{w}]_{i-1}^H \right)^\top \otimes \left( \widetilde{\mathbf{A}}^i \mathbf{B} \right) \right) \right] \delta_{\mathbf{M}}$$

$$= \delta_{\mathbf{M}}^\top \left( \left( \sum_{i_1=1}^{t} \sum_{i_2=1}^{t} \left( \Lambda_{i_1 - i_2} \otimes \left( \widetilde{\mathbf{A}}^{i_1-1} \mathbf{B} \right)^\top \widetilde{\mathbf{A}}^{i_2-1} \mathbf{B} \right) \right) \otimes \mathbb{E} \left[ \mathbf{w}_t \mathbf{w}_t^\top \right] \right) \delta_{\mathbf{M}}$$

$$= \delta_{\mathbf{M}}^\top \left( \left( (\mathbf{I}_H \otimes \mathbf{B}^\top) \Psi (\mathbf{I}_H \otimes \mathbf{B}) \right) \otimes \mathbb{E} \left[ \mathbf{w}_t \mathbf{w}_t^\top \right] \right) \delta_{\mathbf{M}},$$

where $\delta_{\mathbf{M}} = \text{vec} (\mathbf{M}_1 - \mathbf{M}_2)$, $\Lambda_t \in \mathbb{R}^{H \times H}$ with $[\Lambda_l]_{i,j} = 1$ if and only if $i - j = l$ and $0$ otherwise and $\Psi = \sum_{i_1=1}^{t} \sum_{i_2=1}^{t} \left( \Lambda_{i_1 - i_2} \otimes \left( \widetilde{\mathbf{A}}^{i_1-1} \mathbf{B} \right)^\top \widetilde{\mathbf{A}}^{i_2-1} \mathbf{B} \right)$.

Using similar approach, we can obtain

$$\mathbb{E}_{\{\mathbf{w}_i\}_{0 \leq i < t}} \left[ \left\| \mathbf{K} \sum_{i=0}^{t-1} \widetilde{\mathbf{A}}^i \mathbf{B} (\mathbf{M}_1 - \mathbf{M}_2) [\mathbf{w}]_{i-1}^H - (\mathbf{M}_1 - \mathbf{M}_2) [\mathbf{w}]_{t-1}^H \right\|^2 \right] \tag{38}$$

$$= \delta_{\mathbf{M}}^\top \left( \left( (\mathbf{I}_H \otimes \mathbf{B}^\top) \widetilde{\Psi} (\mathbf{I}_H \otimes \mathbf{B}^\top) - \Theta (\mathbf{I}_H \otimes \mathbf{B}) - (\mathbf{I}_H \otimes \mathbf{B}^\top) \Theta^\top + \mathbf{I}_{H d_u} \right) \otimes \mathbb{E} \left[ \mathbf{w}_t \mathbf{w}_t^\top \right] \right) \delta_{\mathbf{M}},$$

where $\widetilde{\Psi} = \sum_{i_1=1}^{t} \sum_{i_2=1}^{t} \left( \Lambda_{i_1 - i_2} \otimes \left( \mathbf{K} \widetilde{\mathbf{A}}^{i_1-1} \mathbf{B} \right)^\top \mathbf{K} \widetilde{\mathbf{A}}^{i_2-1} \mathbf{B} \right)$ and $\Theta = (\mathbf{I}_H \otimes \mathbf{K}) \otimes \left( \sum_{i=1}^{t} \Lambda_{-i} \otimes \left( \widetilde{\mathbf{A}}^{i-1} \mathbf{B} \right) \right)$

Substitue (38) and (37) back into (35), it follows that

$$\mathbb{E}_{\mathbf{x}_0, \{\boldsymbol{\Delta}_i \sim \mathcal{D}_i(\mathbf{M}')\}_{0 \leq i < t}, \{\mathbf{w}_i\}_{0 \leq i < t}} \left[ \left\| \mathbf{x}_t^{(\mathbf{M}_1)} - \mathbf{x}_t^{(\mathbf{M}_2)} \right\|^2 + \left\| \mathbf{u}_t^{(\mathbf{M}_1)} - \mathbf{u}_t^{(\mathbf{M}_2)} \right\|^2 \right] \tag{39}$$

$$= \delta_{\mathbf{M}}^\top \left( \Omega \otimes \mathbb{E} \left[ \mathbf{w}_t \mathbf{w}_t^\top \right] \right) \delta_{\mathbf{M}}.$$

where

$$\Omega = \left(\mathbf{I}_H \otimes \mathbf{B}^\top\right) \Psi \left(\mathbf{I}_H \otimes \mathbf{B}\right) + \left(\mathbf{I}_H \otimes \mathbf{B}^\top\right) \widetilde{\Psi} \left(\mathbf{I}_H \otimes \mathbf{B}^\top\right) \tag{40}$$
$$- \Theta \left(\mathbf{I}_H \otimes \mathbf{B}\right) - \left(\mathbf{I}_H \otimes \mathbf{B}^\top\right) \Theta^\top + \mathbf{I}_{Hd_u}.$$

Based on Lemma F.1 and F.2 in [2], we have, $\forall H \leq t \leq T$, $\Psi \geq \frac{1}{4\kappa^4}\mathbf{I}_{Hd_x}$ and $\|\Theta\| \leq \gamma^{-1}\kappa^3$. Therefore, if $\|\mathbf{I}_H \otimes \mathbf{B}\| \geq \frac{\gamma}{4\kappa^3}$, then

$$\Omega \geq \frac{1}{4\kappa^4}\mathbf{I}_{Hd_x}\left(\frac{\gamma}{4\kappa^3}\right)^2 = \frac{\gamma^2}{64\kappa^{10}}\mathbf{I}_{Hd_x}. \tag{41}$$

Otherwise, if $\|\mathbf{I}_H \otimes \mathbf{B}\| < \frac{\gamma}{4\kappa^3}$, then

$$\Omega \geq \mathbf{I}_{Hd_u} - \Theta \left(\mathbf{I}_H \otimes \mathbf{B}\right) - \left(\mathbf{I}_H \otimes \mathbf{B}^\top\right)\Theta^\top, \tag{42}$$

$$\|\Omega\| \geq 1 - \gamma^{-1}\kappa^3\frac{\gamma}{4\kappa^3} - \gamma^{-1}\kappa^3\frac{\gamma}{4\kappa^3} = \frac{1}{2}. \tag{43}$$

Combine (41) and (43), we have $\|\Omega\| \geq \min\left\{\frac{1}{2}, \frac{\gamma^2}{64\kappa^{10}}\right\}$. As a result,

$$\mathbb{E}_{\mathbf{x}_0, \{\boldsymbol{\Delta}_i \sim \mathcal{D}_i(\mathbf{M}')\}_{0 \leq i < t}, \{\mathbf{w}_i\}_{0 \leq i < t}}\left[\left\|\mathbf{x}_t^{(\mathbf{M}_1)} - \mathbf{x}_t^{(\mathbf{M}_2)}\right\|^2 + \left\|\mathbf{u}_t^{(\mathbf{M}_1)} - \mathbf{u}_t^{(\mathbf{M}_2)}\right\|^2\right] \tag{44}$$

$$= \delta_{\mathbf{M}}^\top\left(\Omega \otimes \mathbb{E}\left[\mathbf{w}_t\mathbf{w}_t^\top\right]\right)\delta_{\mathbf{M}} \geq \delta_{\mathbf{M}}\|\Omega\|\sigma^2 \geq \min\left\{\frac{\sigma^2}{2}, \frac{\gamma^2\sigma^2}{64\kappa^{10}}\right\}\|\mathbf{M}_1 - \mathbf{M}_2\|_F^2.$$

Substitue (44) back into (34), it follows that

$$f_t\left(\mathbf{M}_1; \mathbf{M}'\right) - f_t\left(\mathbf{M}_1; \mathbf{M}'\right) \geq \text{Tr}\left(\nabla^\top f_t\left(\mathbf{M}_2; \mathbf{M}'\right) \cdot \left(\mathbf{M}_1 - \mathbf{M}_2\right)\right) \tag{45}$$

$$+ \frac{\min\left\{\frac{\mu\sigma^2}{2}, \frac{\mu\gamma^2\sigma^2}{64\kappa^{10}}\right\}}{2}\|\mathbf{M}_1 - \mathbf{M}_2\|_F^2.$$

Therefore, Lemma B.2 is proved. $\qquad\square$

**Properties of Gradient.** The following lemma provides upper bounds for the norm of gradients $\left\|\nabla_{\mathbf{x}}c_t\left(\mathbf{x}_t^{(\mathbf{M})}, \mathbf{u}_t^{(\mathbf{M})}\right)\right\|$ and $\left\|\nabla_{\mathbf{u}}c_t\left(\mathbf{x}_t^{(\mathbf{M})}, \mathbf{u}_t^{(\mathbf{M})}\right)\right\|$.

**Lemma B.3** *The gradients of the per stage cost* $c_t\left(\mathbf{x}_t^{(\mathbf{M})}, \mathbf{u}_t^{(\mathbf{M})}\right), \forall 1 \leq t \leq T$, *satifies*

$$\left\|\nabla_{\mathbf{x}}c_t\left(\mathbf{x}_t^{(\mathbf{M})}, \mathbf{u}_t^{(\mathbf{M})}\right)\right\| \leq x_0\kappa^2 G\alpha_t + \kappa^2 GW\left(\|\mathbf{B}\|HM + 1\right)\beta_t, \tag{46}$$

$$\left\|\nabla_{\mathbf{u}}c_t\left(\mathbf{x}_t^{(\mathbf{M})}, \mathbf{u}_t^{(\mathbf{M})}\right)\right\| \leq x_0\kappa^3 G\alpha_t + \kappa^3 GW\left(\|\mathbf{B}\|HM + 1\right)\beta_t + GMHW, \tag{47}$$

*where* $\alpha_t = \prod_{i=0}^{t-1}\left(1 - \gamma + \kappa^2\xi_i\right)$ *and* $\beta_t = \sum_{i=0}^{t-1}\prod_{j=i+1}^{t-1}\left(\mathbf{1}_{j<t}\left(1 - \gamma + \kappa^2\xi_j\right) + \mathbf{1}_{j=t}\right)$.

*Proof.* We first provide upper bounds for $\|\mathbf{x}_t\|$ and $\|\mathbf{u}_t\|$. From the evolution of $\mathbf{x}_t$ in (10), we obtain

$$\|\mathbf{x}_t\| \leq \left\|\prod_{i=0}^{t-1}\left(\widetilde{\mathbf{A}} + \boldsymbol{\Delta}_i\right)\mathbf{x}_0\right\| + \left\|\sum_{i=0}^{t-1}\prod_{j=i+1}^{t-1}\left(\mathbf{1}_{j<t}\left(\widetilde{\mathbf{A}} + \boldsymbol{\Delta}_j\right) + \mathbf{1}_{j=t}\mathbf{I}\right)\mathbf{B}M[\mathbf{w}]_{i-1}^H\right\| \tag{48}$$

$$+ \left\|\sum_{i=0}^{t-1}\prod_{j=i+1}^{t-1}\left(\mathbf{1}_{j<t}\left(\widetilde{\mathbf{A}} + \boldsymbol{\Delta}_j\right) + \mathbf{1}_{j=t}\mathbf{I}\right)\mathbf{w}_i\right\| \leq x_0\kappa^2\prod_{i=0}^{t-1}\left(1 - \gamma + \kappa^2\|\boldsymbol{\Delta}_i\|\right)$$

$$+ \kappa^2 W\left(\|\mathbf{B}\|HM + 1\right)\sum_{i=0}^{t-1}\prod_{j=i+1}^{t-1}\left\|\mathbf{1}_{j<t}\left(1 - \gamma + \kappa^2\|\boldsymbol{\Delta}_j\|\right) + \mathbf{1}_{j=t}\mathbf{I}\right\|,$$

$$\|\mathbf{u}_t\| \leq \kappa\|\mathbf{x}_t\| + MHW. \tag{49}$$

Lemma B.3 follows by noting that $\|\mathbf{\Delta}_t\| \leq \xi_t, \forall 0 \leq t \leq T, \forall \mathbf{M} \in \mathbb{M}$, and whenever $\|\mathbf{x}_t\|, \|\mathbf{u}_t\| \leq D$, we have $\left\|\nabla_{\mathbf{x}} c_t \left(\mathbf{x}_t^{(\mathbf{M})}, \mathbf{u}_t^{(\mathbf{M})}\right)\right\|, \left\|\nabla_{\mathbf{u}} c_t \left(\mathbf{x}_t^{(\mathbf{M})}, \mathbf{u}_t^{(\mathbf{M})}\right)\right\| \leq GD$. $\qquad\square$

We next prove that the variance of the gradient $\nabla_{\mathbf{M}} c_t \left(\mathbf{x}_t^{(\mathbf{M})}, \mathbf{u}_t^{(\mathbf{M})}\right)$ is bounded.

**Lemma B.4** *There exists a $\vartheta_t > 0, \forall 1 \leq t \leq T$, such that*

$$\mathbb{E}_{\mathbf{x}_0, \{\mathbf{\Delta}_i \sim \mathcal{D}_i(\mathbf{M})\}_{0 \leq i < t}, \{\mathbf{w}_i\}_{0 \leq i < t}} \left[\left\|\nabla_{\mathbf{M}} c_t \left(\mathbf{x}_t^{(\mathbf{M})}, \mathbf{u}_t^{(\mathbf{M})}\right)\right\| \right. \tag{50}$$
$$\left. - \mathbb{E}_{\mathbf{x}_0, \{\mathbf{\Delta}_i \sim \mathcal{D}_i(\mathbf{M})\}_{0 \leq i < t}, \{\mathbf{w}_i\}_{0 \leq i < t}} \left[\nabla_{\mathbf{M}} c_t \left(\mathbf{x}_t^{(\mathbf{M})}, \mathbf{u}_t^{(\mathbf{M})}\right)\right]\right\|_F^2 \right] \leq \vartheta_t.$$

*Proof.* Based on the expression of $\nabla_{\mathbf{M}} c_t \left(\mathbf{x}_t^{(\mathbf{M})}, \mathbf{u}_t^{(\mathbf{M})}\right)$ in (24), we have

$$\left\|\nabla_{\mathbf{M}} c_t \left(\mathbf{x}_t^{(\mathbf{M})}, \mathbf{u}_t^{(\mathbf{M})}\right)\right\| \tag{51}$$
$$\leq \|\mathbf{B}\| HW \sum_{i=0}^{t-1} \left\|\prod_{j=i+1}^{t-1} \left(\mathbf{1}_{j<t} \left(\widetilde{\mathbf{A}} + \mathbf{\Delta}_j\right) + \mathbf{1}_{j=t}\mathbf{I}\right)\right\| \left\|\nabla_{\mathbf{x}} c_t \left(\mathbf{x}_t^{(\mathbf{M})}, \mathbf{u}_t^{(\mathbf{M})}\right)\right\|$$
$$+ \kappa \|\mathbf{B}\| HW \sum_{i=0}^{t-1} \left\|\prod_{j=i+1}^{t-1} \left(\mathbf{1}_{j<t} \left(\widetilde{\mathbf{A}} + \mathbf{\Delta}_j\right) + \mathbf{1}_{j=t}\mathbf{I}\right)\right\| \left\|\nabla_{\mathbf{u}} c_t \left(\mathbf{x}_t^{(\mathbf{M})}, \mathbf{u}_t^{(\mathbf{M})}\right)\right\|$$
$$+ HW \left\|\nabla_{\mathbf{u}} c_t \left(\mathbf{x}_t^{(\mathbf{M})}, \mathbf{u}_t^{(\mathbf{M})}\right)\right\|$$
$$\leq \kappa^2 \|\mathbf{B}\| HW \beta_t \left\|\nabla_{\mathbf{x}} c_t \left(\mathbf{x}_t^{(\mathbf{M})}, \mathbf{u}_t^{(\mathbf{M})}\right)\right\| + \left(\kappa^3 \|\mathbf{B}\| \beta_t + 1\right) HW \left\|\nabla_{\mathbf{u}} c_t \left(\mathbf{x}_t^{(\mathbf{M})}, \mathbf{u}_t^{(\mathbf{M})}\right)\right\|$$
$$\overset{(51.a)}{\leq} \left(\kappa \|\mathbf{B}\| HW \beta_t + \kappa^3 \|\mathbf{B}\| \beta_t + 1\right) \left(x_0 \kappa^3 G \alpha_t + \kappa^3 GW \left(\|\mathbf{B}\| HM + 1\right) \beta_t\right)$$
$$+ \left(\kappa^3 \|\mathbf{B}\| \beta_t + 1\right) GMHW,$$

where inequality $(51.a)$ holds becasue of the gradient norm bounds in Lemma B.3.

Further note that

$$\mathbb{E}\left[\left\|\nabla_{\mathbf{M}} c_t \left(\mathbf{x}_t^{(\mathbf{M})}, \mathbf{u}_t^{(\mathbf{M})}\right) - \mathbb{E}\left[\nabla_{\mathbf{M}} c_t \left(\mathbf{x}_t^{(\mathbf{M})}, \mathbf{u}_t^{(\mathbf{M})}\right)\right]\right\|_F^2\right] \leq \mathbb{E}\left[\left\|\nabla_{\mathbf{M}} c_t \left(\mathbf{x}_t^{(\mathbf{M})}, \mathbf{u}_t^{(\mathbf{M})}\right)\right\|_F^2\right]$$
$$\tag{52}$$
$$\leq d_x^2 H \mathbb{E}\left[\left\|\nabla_{\mathbf{M}} c_t \left(\mathbf{x}_t^{(\mathbf{M})}, \mathbf{u}_t^{(\mathbf{M})}\right)\right\|^2\right]$$

The desired result follows by substituting (51) into (52) and letting

$$\vartheta_t = d_x^2 H \left(\left(\kappa \|\mathbf{B}\| HW \beta_t + \kappa^3 \|\mathbf{B}\| \beta_t + 1\right) \left(x_0 \kappa^3 G \alpha_t + \kappa^3 GW \left(\|\mathbf{B}\| HM + 1\right) \beta_t\right)\right. \tag{53}$$
$$\left. + \left(\kappa^3 \|\mathbf{B}\| \beta_t + 1\right) GMHW\right).$$

$\qquad\square$

The next lemma characterizes the difference between the gradients under any two different polices $\mathbf{M}$ and $\mathbf{M}'$.

**Lemma B.5** *For any given two policies* $\mathbf{M}, \mathbf{M}' \in \mathbb{M}$, *the differences between the gradients satify,* $\forall 1 \le t \le T$,

$$
\left\| \nabla_{\mathbf{x}} c_t \left( \mathbf{x}_t^{(\mathbf{M})}, \mathbf{u}_t^{(\mathbf{M})} \right) - \nabla_{\mathbf{x}} c_t \left( \mathbf{x}_t^{(\mathbf{M}')}, \mathbf{u}_t^{(\mathbf{M}')} \right) \right\|
$$
$$
+ \left\| \nabla_{\mathbf{u}} c_t \left( \mathbf{x}_t^{(\mathbf{M})}, \mathbf{u}_t^{(\mathbf{M})} \right) - \nabla_{\mathbf{u}} c_t \left( \mathbf{x}_t^{(\mathbf{M}')}, \mathbf{u}_t^{(\mathbf{M}')} \right) \right\| \tag{54}
$$
$$
\le \varsigma \left( (1 + \kappa) \kappa^2 H W \beta_t + H W \right) \|\mathbf{M} - \mathbf{M}'\|
$$
$$
+ \varsigma (1 + \kappa) \kappa^2 (1 - \gamma)^{-1} (x_0 \alpha_t + (\|\mathbf{B}\| H M + 1) W \beta_t) \sum_{i=0}^{t-1} \|\boldsymbol{\Delta}_i - \boldsymbol{\Delta}'_i\|,
$$

*where* $\boldsymbol{\Delta}_i$ *and* $\boldsymbol{\Delta}'_i, \forall 0 \le i < t$, *denotes the policy-dependent state transition perturbations under the policy* $\mathbf{M}$ *and* $\mathbf{M}'$, *respectively.*

*Proof.* Recall the smoothness of the cost function $c_t(\mathbf{x}, \mathbf{u})$ in A 5, i.e.,

$$
\|\nabla_{\mathbf{x}} c_t(\mathbf{x}_1, \mathbf{u}_1) - \nabla_{\mathbf{x}} c_t(\mathbf{x}_2, \mathbf{u}_2)\| + \|\nabla_{\mathbf{u}} c_t(\mathbf{x}_1, \mathbf{u}_1) - \nabla_{\mathbf{u}} c_t(\mathbf{x}_2, \mathbf{u}_2)\| \tag{55}
$$
$$
\le \varsigma (\|\mathbf{x}_1 - \mathbf{x}_2\| + \|\mathbf{u}_1 - \mathbf{u}_2\|).
$$

To prove Lemma B.5, it suffices to quantify $\left\| \mathbf{x}_t^{(\mathbf{M})} - \mathbf{x}_t^{(\mathbf{M}')} \right\|$ and $\left\| \mathbf{u}_t^{(\mathbf{M})} - \mathbf{u}_t^{(\mathbf{M}')} \right\|$.

Based on the evolution of the system state $\mathbf{x}_t^{(\mathbf{M})}$, we have

$$
\mathbf{x}_t^{(\mathbf{M})} - \mathbf{x}_t^{(\mathbf{M}')} = E_1 + E_2 + E_3, \tag{56}
$$

where

$$
E_1 = \left( \prod_{i=0}^{t-1} \left( \widetilde{\mathbf{A}} + \boldsymbol{\Delta}_i \right) - \prod_{i=0}^{t-1} \left( \widetilde{\mathbf{A}} + \boldsymbol{\Delta}'_i \right) \right) \mathbf{x}_0, \tag{57}
$$

$$
E_2 = \sum_{i=0}^{t-1} \prod_{j=i+1}^{t-1} \left( \mathbf{1}_{j<t} \left( \widetilde{\mathbf{A}} + \boldsymbol{\Delta}_j \right) + \mathbf{1}_{j=t} \mathbf{I} \right) \mathbf{B} \mathbf{M} [\mathbf{w}]_{i-1}^{H} \tag{58}
$$
$$
- \sum_{i=0}^{t-1} \prod_{j=i+1}^{t-1} \left( \mathbf{1}_{j<t} \left( \widetilde{\mathbf{A}} + \boldsymbol{\Delta}'_j \right) + \mathbf{1}_{j=t} \mathbf{I} \right) \mathbf{B} \mathbf{M}' [\mathbf{w}]_{i-1}^{H},
$$

$$
E_3 = \sum_{i=0}^{t-1} \left( \prod_{j=i+1}^{t-1} \left( \mathbf{1}_{j<t} \left( \widetilde{\mathbf{A}} + \boldsymbol{\Delta}_j \right) + \mathbf{1}_{j=t} \mathbf{I} \right) - \prod_{j=i+1}^{t-1} \left( \mathbf{1}_{j<t} \left( \widetilde{\mathbf{A}} + \boldsymbol{\Delta}'_j \right) + \mathbf{1}_{j=t} \mathbf{I} \right) \right) \mathbf{w}_i. \tag{59}
$$

We next analyze each term in (56) one by one. Specifically,

$$
\|E_1\| \le x_0 \left\| \prod_{i=0}^{t-1} \left( \widetilde{\mathbf{A}} + \boldsymbol{\Delta}_i \right) - \prod_{i=0}^{t-1} \left( \widetilde{\mathbf{A}} + \boldsymbol{\Delta}'_i \right) \right\| \tag{60}
$$
$$
\le x_0 \left\| (\boldsymbol{\Delta}_0 - \boldsymbol{\Delta}'_0) \prod_{i=1}^{t-1} \left( \widetilde{\mathbf{A}} + \boldsymbol{\Delta}_i \right) + \left( \widetilde{\mathbf{A}} + \boldsymbol{\Delta}'_0 \right) (\boldsymbol{\Delta}_1 - \boldsymbol{\Delta}'_1) \prod_{i=2}^{t-1} \left( \widetilde{\mathbf{A}} + \boldsymbol{\Delta}_i \right) \right.
$$
$$
\left. + \cdots + \prod_{i=0}^{t-2} \left( \widetilde{\mathbf{A}} + \boldsymbol{\Delta}'_i \right) (\boldsymbol{\Delta}_{t-1} - \boldsymbol{\Delta}'_{t-1}) \right\|
$$
$$
\le x_0 \kappa^2 (1 - \gamma)^{-1} \prod_{i=0}^{t-1} \left( 1 - \gamma + \kappa^2 \xi_i \right) \sum_{i=0}^{t-1} \|\boldsymbol{\Delta}_i - \boldsymbol{\Delta}'_i\|
$$
$$
= x_0 \kappa^2 (1 - \gamma)^{-1} \alpha_t \sum_{i=0}^{t-1} \|\boldsymbol{\Delta}_i - \boldsymbol{\Delta}'_i\|.
$$

For the term $E_2$, we use the similar approach for analyzing $E_1$ in (74) and obtain the following bound

$$\|E_2\| \tag{61}$$

$$\leq \left\| \sum_{i=0}^{t-1} \prod_{j=i+1}^{t-1} \left(\mathbf{1}_{j<t}\left(\widetilde{\mathbf{A}} + \boldsymbol{\Delta}_j\right) + \mathbf{1}_{j=t}\mathbf{I}\right) \mathbf{B}\left(\mathbf{M} - \mathbf{M}'\right) [\mathbf{w}]_{i-1}^{H} \right.$$

$$\left. + \sum_{i=0}^{t-1} \left( \prod_{j=i+1}^{t-1} \left(\mathbf{1}_{j<t}\left(\widetilde{\mathbf{A}} + \boldsymbol{\Delta}_j\right) + \mathbf{1}_{j=t}\mathbf{I}\right) - \prod_{j=i+1}^{t-1} \left(\mathbf{1}_{j<t}\left(\widetilde{\mathbf{A}} + \boldsymbol{\Delta}'_j\right) + \mathbf{1}_{j=t}\mathbf{I}\right) \right) \mathbf{B}\mathbf{M}' [\mathbf{w}]_{i-1}^{H} \right\|$$

$$\leq \kappa^2 HW\beta_t \|\mathbf{M} - \mathbf{M}'\|$$

$$+ \|\mathbf{B}\| MHW \sum_{i=0}^{t-1} \left\| \prod_{j=i+1}^{t-1} \left(\mathbf{1}_{j<t}\left(\widetilde{\mathbf{A}} + \boldsymbol{\Delta}_j\right) + \mathbf{1}_{j=t}\mathbf{I}\right) - \prod_{j=i+1}^{t-1} \left(\mathbf{1}_{j<t}\left(\widetilde{\mathbf{A}} + \boldsymbol{\Delta}'_j\right) + \mathbf{1}_{j=t}\mathbf{I}\right) \right\|$$

$$\overset{(61.a)}{\leq} \kappa^2 HW\beta_t \|\mathbf{M} - \mathbf{M}'\| + \kappa^2 (1-\gamma)^{-1} \|\mathbf{B}\| MHW\beta_t \sum_{i=0}^{t-1} \|\boldsymbol{\Delta}_i - \boldsymbol{\Delta}'_i\|,$$

where inequality $(61.a)$ is obtained by using (76).

Finally, still using (76), an upper bound of $\|E_3\|$ is obtained as follows

$$\|E_3\| \leq W \sum_{i=0}^{t-1} \left\| \prod_{j=i+1}^{t-1} \left(\mathbf{1}_{j<t}\left(\widetilde{\mathbf{A}} + \boldsymbol{\Delta}_j\right) + \mathbf{1}_{j=t}\mathbf{I}\right) - \prod_{j=i+1}^{t-1} \left(\mathbf{1}_{j<t}\left(\widetilde{\mathbf{A}} + \boldsymbol{\Delta}'_j\right) + \mathbf{1}_{j=t}\mathbf{I}\right) \right\| \tag{62}$$

$$\leq \kappa^2 (1-\gamma)^{-1} W\beta_t \sum_{i=0}^{t-1} \|\boldsymbol{\Delta}_i - \boldsymbol{\Delta}'_i\|.$$

It follow directly that

$$\left\| \mathbf{x}_t^{(\mathbf{M})} - \mathbf{x}_t^{(\mathbf{M}')} \right\| \leq \|E_1\| + \|E_2\| + \|E_3\| \tag{63}$$

$$\leq \kappa^2 HW\beta_t \|\mathbf{M} - \mathbf{M}'\| + \kappa^2 (1-\gamma)^{-1} \left(x_0\alpha_t + (\|\mathbf{B}\| HM + 1) W\beta_t\right) \sum_{i=0}^{t-1} \|\boldsymbol{\Delta}_i - \boldsymbol{\Delta}'_i\|.$$

We next observe the relationship

$$\left\| \mathbf{u}_t^{(\mathbf{M})} - \mathbf{u}_t^{(\mathbf{M}')} \right\| = \left\| -\mathbf{K}\mathbf{x}_t^{(\mathbf{M})} + \mathbf{K}\mathbf{x}_t^{(\mathbf{M}')} + \mathbf{M}[\mathbf{w}]_{t-1}^{H} - \mathbf{M}'[\mathbf{w}]_{t-1}^{H} \right\| \tag{64}$$

$$\leq \kappa \left\| \mathbf{x}_t^{(\mathbf{M})} - \mathbf{x}_t^{(\mathbf{M}')} \right\| + HW \|\mathbf{M} - \mathbf{M}'\|.$$

Therefore, it follows that

$$\left\| \mathbf{x}_t^{(\mathbf{M})} - \mathbf{x}_t^{(\mathbf{M}')} \right\| + \left\| \mathbf{u}_t^{(\mathbf{M})} - \mathbf{u}_t^{(\mathbf{M}')} \right\| \leq (1+\kappa) \left\| \mathbf{x}_t^{(\mathbf{M})} - \mathbf{x}_t^{(\mathbf{M}')} \right\| + HW \|\mathbf{M} - \mathbf{M}'\| \tag{65}$$

$$\leq \left((1+\kappa)\kappa^2 HW\beta_t + HW\right) \|\mathbf{M} - \mathbf{M}'\|$$

$$+ (1+\kappa)\kappa^2 (1-\gamma)^{-1} \left(x_0\alpha_t + (\|\mathbf{B}\| HM + 1) W\beta_t\right) \sum_{i=0}^{t-1} \|\boldsymbol{\Delta}_i - \boldsymbol{\Delta}'_i\|.$$

Combining (65) and (55), we obatin the desired result. $\qquad\square$

**Smoothness**: We analyze the smoothness of the per stage cost and the per stage expected cost in the following Lemma B.6 and B.8, respectively.

**Lemma B.6** *The per stage cost function* $c_t\left(\mathbf{M}; \{\boldsymbol{\Delta}_i\}_{0\le i<t}\right), \forall 1 \le t \le T$, *is smooth in the sense that*

$$\left\|\nabla_{\mathbf{M}} c_t\left(\mathbf{M}; \{\boldsymbol{\Delta}_i\}_{0\le i<t}\right) - \nabla_{\mathbf{M}'} c_t\left(\mathbf{M}'; \{\boldsymbol{\Delta}_i'\}_{0\le i<t}\right)\right\|_F \tag{66}$$

$$\le \lambda_t \|\mathbf{M} - \mathbf{M}'\|_F + \nu_t \sum_{i=0}^{t-1} \|\boldsymbol{\Delta}_i - \boldsymbol{\Delta}_i'\|_F,$$

*where*

$$\lambda_t = d_x \sqrt{H} \varsigma H^2 W^2 \left(1 + \|\mathbf{B}\| \left(\kappa^2 + \kappa^3\right)\right) \left(\left(\kappa^2 + \kappa^3\right) \beta_t + 1\right), \tag{67}$$

$$\nu_t = d_x \sqrt{H} \left(\varsigma HW \left(1 + \|\mathbf{B}\| \left(\kappa^2 + \kappa^3\right)\right) \left(\kappa^2 + \kappa^3\right) (1 - \gamma)^{-1} + G (1 - \gamma)^{-1} \right. \tag{68}$$

$$\left. \cdot \left(\kappa^4 + \kappa^5\right) HW \|\mathbf{B}\| \beta_t\right) (x_0 \alpha_t + (\|\mathbf{B}\| HM + 1) W \beta_t),$$

*and* $\boldsymbol{\Delta}_i$ *and* $\boldsymbol{\Delta}_i', \forall 0 \le i < t$, *denotes the policy-dependent state transition perturbations under the policy* $\mathbf{M}$ *and* $\mathbf{M}'$, *respectively.*

*Proof.* We prove this lemma based on the smoothness of the cost function $c_t(\mathbf{x}, \mathbf{u})$ in A 5, i.e.,

$$\|\nabla_{\mathbf{x}} c_t(\mathbf{x}_1, \mathbf{u}_1) - \nabla_{\mathbf{x}} c_t(\mathbf{x}_2, \mathbf{u}_1)\| + \|\nabla_{\mathbf{u}} c_t(\mathbf{x}_1, \mathbf{u}_1) - \nabla_{\mathbf{u}} c_t(\mathbf{x}_1, \mathbf{u}_2)\|$$
$$\le \varsigma \left(\|\mathbf{x}_1 - \mathbf{x}_2\| + \|\mathbf{u}_1 - \mathbf{u}_2\|\right). \tag{69}$$

Based on the expression of $\nabla_{\mathbf{M}} c_t\left(\mathbf{M}; \{\boldsymbol{\Delta}_i\}_{0\le i<t}\right)$ in (24), it follows that

$$\nabla_{\mathbf{M}} c_t\left(\mathbf{M}; \{\boldsymbol{\Delta}_i\}_{0\le i<t}\right) - \nabla_{\mathbf{M}'} c_t\left(\mathbf{M}'; \{\boldsymbol{\Delta}_i'\}_{0\le i<t}\right) \le F_1 + F_2 + F_3, \tag{70}$$

where

$$F_1 = \sum_{i=0}^{t-1} \left(\left(\prod_{j=i+1}^{t-1} \left(\mathbf{1}_{j<t}\left(\widetilde{\mathbf{A}} + \boldsymbol{\Delta}_j\right) + \mathbf{1}_{j=t}\mathbf{I}\right)\mathbf{B}\right)^\top \nabla_{\mathbf{x}} c_t\left(\mathbf{x}_t^{(\mathbf{M})}, \mathbf{u}_t^{(\mathbf{M})}\right)\right. \tag{71}$$

$$\left. - \left(\prod_{j=i+1}^{t-1} \left(\mathbf{1}_{j<t}\left(\widetilde{\mathbf{A}} + \boldsymbol{\Delta}_j'\right) + \mathbf{1}_{j=t}\mathbf{I}\right)\mathbf{B}\right)^\top \nabla_{\mathbf{x}} c_t\left(\mathbf{x}_t^{(\mathbf{M}')}, \mathbf{u}_t^{(\mathbf{M}')}\right)\right) \left([\mathbf{w}]_{i-1}^H\right)^\top,$$

$$F_2 = \sum_{i=0}^{t-1} \left(\left(\mathbf{K} \prod_{j=i+1}^{t-1} \left(\mathbf{1}_{j<t}\left(\widetilde{\mathbf{A}} + \boldsymbol{\Delta}_j\right) + \mathbf{1}_{j=t}\mathbf{I}\right)\mathbf{B}\right)^\top \nabla_{\mathbf{x}} c_t\left(\mathbf{x}_t^{(\mathbf{M})}, \mathbf{u}_t^{(\mathbf{M})}\right)\right. \tag{72}$$

$$\left. - \left(\mathbf{K} \prod_{j=i+1}^{t-1} \left(\mathbf{1}_{j<t}\left(\widetilde{\mathbf{A}} + \boldsymbol{\Delta}_j'\right) + \mathbf{1}_{j=t}\mathbf{I}\right)\mathbf{B}\right)^\top \nabla_{\mathbf{x}} c_t\left(\mathbf{x}_t^{(\mathbf{M}')}, \mathbf{u}_t^{(\mathbf{M}')}\right)\right) \left([\mathbf{w}]_{i-1}^H\right)^\top,$$

$$F_3 = \left(\nabla_{\mathbf{u}} c_t\left(\mathbf{x}_t^{(\mathbf{M})}, \mathbf{u}_t^{(\mathbf{M})}\right) - \nabla_{\mathbf{u}} c_t\left(\mathbf{x}_t^{(\mathbf{M}')}, \mathbf{u}_t^{(\mathbf{M}')}\right)\right) \left([\mathbf{w}]_{t-1}^H\right)^\top. \tag{73}$$

We next analyze each term in (70) one by one. Specifically,

$$
\|F_1\| = \left\| \sum_{i=0}^{t-1} \left( \left( \prod_{j=i+1}^{t-1} \left( \mathbf{1}_{j<t} \left( \widetilde{\mathbf{A}} + \mathbf{\Delta}_j \right) + \mathbf{1}_{j=t}\mathbf{I} \right) \mathbf{B} \right)^{\top} \nabla_{\mathbf{x}} c_t \left( \mathbf{x}_t^{(\mathbf{M})}, \mathbf{u}_t^{(\mathbf{M})} \right) \right. \right. \tag{74}
$$

$$
- \left( \prod_{j=i+1}^{t-1} \left( \mathbf{1}_{j<t} \left( \widetilde{\mathbf{A}} + \mathbf{\Delta}_j \right) + \mathbf{1}_{j=t}\mathbf{I} \right) \mathbf{B} \right)^{\top} \nabla_{\mathbf{x}} c_t \left( \mathbf{x}_t^{(\mathbf{M'})}, \mathbf{u}_t^{(\mathbf{M'})} \right)
$$

$$
+ \left( \prod_{j=i+1}^{t-1} \left( \mathbf{1}_{j<t} \left( \widetilde{\mathbf{A}} + \mathbf{\Delta}_j \right) + \mathbf{1}_{j=t}\mathbf{I} \right) \mathbf{B} \right)^{\top} \nabla_{\mathbf{x}} c_t \left( \mathbf{x}_t^{(\mathbf{M'})}, \mathbf{u}_t^{(\mathbf{M'})} \right)
$$

$$
\left. \left. - \left( \prod_{j=i+1}^{t-1} \left( \mathbf{1}_{j<t} \left( \widetilde{\mathbf{A}} + \mathbf{\Delta}'_j \right) + \mathbf{1}_{j=t}\mathbf{I} \right) \mathbf{B} \right)^{\top} \nabla_{\mathbf{x}} c_t \left( \mathbf{x}_t^{(\mathbf{M'})}, \mathbf{u}_t^{(\mathbf{M'})} \right) \right) \left( [\mathbf{w}]_{i-1}^H \right)^{\top} \right\|
$$

$$
\overset{(74.a)}{\leq} HW \|\mathbf{B}\| \kappa^2 \beta_t \left\| \nabla_{\mathbf{x}} c_t \left( \mathbf{x}_t^{(\mathbf{M})}, \mathbf{u}_t^{(\mathbf{M})} \right) - \nabla_{\mathbf{x}} c_t \left( \mathbf{x}_t^{(\mathbf{M'})}, \mathbf{u}_t^{(\mathbf{M'})} \right) \right\|
$$

$$
+ HW \|\mathbf{B}\| \kappa^2 \left( x_0 G \alpha_t + GW \left( \|\mathbf{B}\| HM + 1 \right) \beta_t \right)
$$

$$
\cdot \sum_{i=0}^{t-1} \left\| \prod_{j=i+1}^{t-1} \left( \mathbf{1}_{j<t} \left( \widetilde{\mathbf{A}} + \mathbf{\Delta}_j \right) + \mathbf{1}_{j=t}\mathbf{I} \right) - \prod_{j=i+1}^{t-1} \left( \mathbf{1}_{j<t} \left( \widetilde{\mathbf{A}} + \mathbf{\Delta}'_j \right) + \mathbf{1}_{j=t}\mathbf{I} \right) \right\|,
$$

where inequality $(74.a)$ holds due to the definition of $\beta_t$ and the upper bound of $\left\| \nabla_{\mathbf{x}} c_t \left( \mathbf{x}_t^{(\mathbf{M'})}, \mathbf{u}_t^{(\mathbf{M'})} \right) \right\|$ provided in Lemma B.3.

For the last term in (74), we first observe that

$$\left\| \prod_{j=i+1}^{t-1} \left( \mathbf{1}_{j<t} \left( \widetilde{\mathbf{A}} + \boldsymbol{\Delta}'_j \right) + \mathbf{1}_{j=t}\mathbf{I} \right) - \prod_{j=i+1}^{t-1} \left( \mathbf{1}_{j<t} \left( \widetilde{\mathbf{A}} + \boldsymbol{\Delta}_j \right) + \mathbf{1}_{j=t}\mathbf{I} \right) \right\| \tag{75}$$

$$= \left\| \left( \boldsymbol{\Delta}'_{i+1} - \boldsymbol{\Delta}_{i+1} \right) \prod_{j=i+2}^{t-1} \left( \mathbf{1}_{j<t} \left( \widetilde{\mathbf{A}} + \boldsymbol{\Delta}'_j \right) + \mathbf{1}_{j=t}\mathbf{I} \right) \right.$$

$$+ \left( \widetilde{\mathbf{A}} + \boldsymbol{\Delta}_{i+1} \right) \left( \boldsymbol{\Delta}'_{i+2} - \boldsymbol{\Delta}_{i+2} \right) \prod_{j=i+3}^{t-1} \left( \mathbf{1}_{j<t} \left( \widetilde{\mathbf{A}} + \boldsymbol{\Delta}'_j \right) + \mathbf{1}_{j=t}\mathbf{I} \right) + \cdots$$

$$+ \prod_{k=i+1}^{i+1+l} \left( \mathbf{1}_{k<t} \left( \widetilde{\mathbf{A}} + \boldsymbol{\Delta}_k \right) + \mathbf{1}_{k=t}\mathbf{I} \right) \left( \boldsymbol{\Delta}'_{i+1+l+1} - \boldsymbol{\Delta}_{i+1+l+1} \right) \prod_{j=i+1+l+2}^{t-1} \left( \mathbf{1}_{j<t} \left( \widetilde{\mathbf{A}} + \boldsymbol{\Delta}'_j \right) + \mathbf{1}_{j=t}\mathbf{I} \right)$$

$$+ \cdots + \prod_{k=i+1}^{t-2} \left( \mathbf{1}_{k<t} \left( \widetilde{\mathbf{A}} + \boldsymbol{\Delta}_k \right) + \mathbf{1}_{k=t}\mathbf{I} \right) \left( \boldsymbol{\Delta}'_{t-1} - \boldsymbol{\Delta}_{t-1} \right) \right\|$$

$$\leq \left( \prod_{j=i+1}^{t-1} \left( \mathbf{1}_{j<t} \left( 1 - \gamma + \kappa^2 \xi_j \right) + \mathbf{1}_{j=t} \right) \right) \left( \kappa^2 \left\| \boldsymbol{\Delta}'_{i+1} - \boldsymbol{\Delta}_{i+1} \right\| \left( 1 - \gamma + \kappa^2 \xi_{i+1} \right)^{-1} \right.$$

$$+ \cdots + \kappa^2 \left\| \left( \boldsymbol{\Delta}'_{i+1+l+1} - \boldsymbol{\Delta}_{i+1+l+1} \right) \right\| \left( 1 - \gamma + \kappa^2 \xi_{i+1+l+1} \right)^{-1} + \cdots$$

$$+ \kappa^2 \left\| \boldsymbol{\Delta}'_{t-1} - \boldsymbol{\Delta}_{t-1} \right\| \left( 1 - \gamma + \kappa^2 \xi_{t-1} \right)^{-1} \right)$$

$$\leq \left( \prod_{j=i+1}^{t-1} \left( \mathbf{1}_{j<t} \left( 1 - \gamma + \kappa^2 \xi_j \right) + \mathbf{1}_{j=t} \right) \right) \sum_{k=i+1}^{t-1} \kappa^2 \left( 1 - \gamma \right)^{-1} \left\| \boldsymbol{\Delta}'_k - \boldsymbol{\Delta}_k \right\|$$

$$\leq \left( \prod_{j=i+1}^{t-1} \left( \mathbf{1}_{j<t} \left( 1 - \gamma + \kappa^2 \xi_j \right) + \mathbf{1}_{j=t} \right) \right) \sum_{k=0}^{t-1} \kappa^2 \left( 1 - \gamma \right)^{-1} \left\| \boldsymbol{\Delta}'_k - \boldsymbol{\Delta}_k \right\|.$$

As a result,

$$\sum_{i=0}^{t-1} \left\| \prod_{j=i+1}^{t-1} \left( \mathbf{1}_{j<t} \left( \widetilde{\mathbf{A}} + \boldsymbol{\Delta}'_j \right) + \mathbf{1}_{j=t}\mathbf{I} \right) - \prod_{j=i+1}^{t-1} \left( \mathbf{1}_{j<t} \left( \widetilde{\mathbf{A}} + \boldsymbol{\Delta}_j \right) + \mathbf{1}_{j=t}\mathbf{I} \right) \right\| \tag{76}$$

$$\leq \sum_{i=0}^{t-1} \prod_{j=i+1}^{t-1} \left( \mathbf{1}_{j<t} \left( 1 - \gamma + \kappa^2 \xi_j \right) + \mathbf{1}_{j=t} \right) \left( \sum_{k=0}^{t-1} \kappa^2 \left( 1 - \gamma \right)^{-1} \left\| \boldsymbol{\Delta}'_k - \boldsymbol{\Delta}_k \right\| \right)$$

$$= \kappa^2 \left( 1 - \gamma \right)^{-1} \beta_t \sum_{i=0}^{t-1} \left\| \boldsymbol{\Delta}'_i - \boldsymbol{\Delta}_i \right\|.$$

Substitue (76) into (74), it follows

$$\|E_1\| \leq HW \left\| \mathbf{B} \right\| \kappa^2 \beta_t \left( \left\| \nabla_{\mathbf{x}} c_t \left( \mathbf{x}_t^{(\mathbf{M})}, \mathbf{u}_t^{(\mathbf{M})} \right) - \nabla_{\mathbf{x}} c_t \left( \mathbf{x}_t^{(\mathbf{M}')}, \mathbf{u}_t^{(\mathbf{M}')} \right) \right\| \right. \tag{77}$$

$$\left. + \left( x_0 G \alpha_t + GW \left( \left\| \mathbf{B} \right\| HM + 1 \right) \beta_t \right) \kappa^2 \left( 1 - \gamma \right)^{-1} \sum_{i=0}^{t-1} \left\| \boldsymbol{\Delta}'_i - \boldsymbol{\Delta}_i \right\| \right)$$

The upper bounds for $\|E_2\|$ and $\|E_3\|$ can be readily obtained as follows

$$\|E_2\| \leq \kappa HW \left\| \mathbf{B} \right\| \kappa^2 \beta_t \left( \left\| \nabla_{\mathbf{x}} c_t \left( \mathbf{x}_t^{(\mathbf{M})}, \mathbf{u}_t^{(\mathbf{M})} \right) - \nabla_{\mathbf{x}} c_t \left( \mathbf{x}_t^{(\mathbf{M}')}, \mathbf{u}_t^{(\mathbf{M}')} \right) \right\| \right. \tag{78}$$

$$\left. + \left( x_0 G \alpha_t + GW \left( \left\| \mathbf{B} \right\| HM + 1 \right) \beta_t \right) \kappa^2 \left( 1 - \gamma \right)^{-1} \sum_{i=0}^{t-1} \left\| \boldsymbol{\Delta}'_i - \boldsymbol{\Delta}_i \right\| \right)$$

and

$$\|E_3\| \leq HW \left\| \nabla_{\mathbf{u}} c_t \left( \mathbf{x}_t^{(\mathbf{M})}, \mathbf{u}_t^{(\mathbf{M})} \right) - \nabla_{\mathbf{u}} c_t \left( \mathbf{x}_t^{(\mathbf{M}')}, \mathbf{u}_t^{(\mathbf{M}')} \right) \right\|. \tag{79}$$

Substitute (77), (78) and (79) into (70), we have

$$\left\| \nabla_{\mathbf{M}} c_t \left( \mathbf{M}; \{\boldsymbol{\Delta}_i\}_{0 \leq i < t} \right) - \nabla_{\mathbf{M}'} c_t \left( \mathbf{M}'; \{\boldsymbol{\Delta}_i'\}_{0 \leq i < t} \right) \right\| \tag{80}$$
$$\leq \left( 1 + \|\mathbf{B}\| \left( \kappa^2 + \kappa^3 \right) \right) HW \left( \left\| \nabla_{\mathbf{x}} c_t \left( \mathbf{x}_t^{(\mathbf{M})}, \mathbf{u}_t^{(\mathbf{M})} \right) - \nabla_{\mathbf{x}} c_t \left( \mathbf{x}_t^{(\mathbf{M}')}, \mathbf{u}_t^{(\mathbf{M}')} \right) \right\| \right.$$
$$\left. + \left\| \nabla_{\mathbf{u}} c_t \left( \mathbf{x}_t, \mathbf{u}_t^{(\mathbf{M})} \right) - \nabla_{\mathbf{u}} c_t \left( \mathbf{x}_t, \mathbf{u}_t^{(\mathbf{M}')} \right) \right\| \right) + \widetilde{\beta}_t \sum_{i=0}^{t-1} \|\boldsymbol{\Delta}_i' - \boldsymbol{\Delta}_i\|,$$

where $\widetilde{\beta}_t = (1 - \gamma)^{-1} \left( \kappa^4 + \kappa^5 \right) HW \|\mathbf{B}\| \beta_t \left( x_0 G \alpha_t + GW \left( \|\mathbf{B}\| HM + 1 \right) \beta_t \right).$

Applying Lemma B.5 to (80) leads to the desired result. $\qquad\square$

Before we proceed to analyze the gradient difference of the expected per stage cost, we cite the following tool lemma in [21].

**Lemma B.7** *(Kantorovich-Rubinstein (Lemma D.3 in [21])) A distribution map $\mathcal{D}(\cdot)$ is $\varepsilon$-sensitive if and only if for all $\mathbf{M}, \mathbf{M}' \in \mathbb{M}$:*

$$\sup \left\{ \mathbb{E}_{\mathbf{A} \sim \mathcal{D}(\mathbf{M})} \left[ g \left( \mathbf{A} \right) \right] - \mathbb{E}_{\mathbf{A}' \sim \mathcal{D}(\mathbf{M}')} \left[ g \left( \mathbf{A}' \right) \right] : g : \mathbb{R}^{d_x \times d_x} \to \mathbb{R}, g \ 1 - Lipschitz \right\}$$
$$\leq \varepsilon \|\mathbf{M} - \mathbf{M}'\|_F. \tag{81}$$

The smoothness regarding the expected per stage cost is summarized below.

**Lemma B.8** *For any $\mathbf{M}, \mathbf{M}', \mathbf{M}_1, \mathbf{M}_2 \in \mathbb{M}$ and $\forall 1 \leq t \leq T$, the following inequality holds*

$$\left\| \mathbb{E}_{\mathbf{x}_0, \{\boldsymbol{\Delta}_i \sim \mathcal{D}_i(\mathbf{M}_1)\}_{0 \leq i < t}, \{\mathbf{w}_i\}_{0 \leq i < t}} \left[ \nabla_{\mathbf{M}} c_t \left( \mathbf{M}; \{\boldsymbol{\Delta}_i\}_{0 \leq i < t} \right) \right] \right. \tag{82}$$
$$\left. - \mathbb{E}_{\mathbf{x}_0, \{\boldsymbol{\Delta}_i' \sim \mathcal{D}_i(\mathbf{M}_2)\}_{0 \leq i < t}, \{\mathbf{w}_i\}_{0 \leq i < t}} \left[ \nabla_{\mathbf{M}'} c_t \left( \mathbf{M}'; \{\boldsymbol{\Delta}_i'\}_{0 \leq i < t} \right) \right] \right\|_F$$
$$\leq \lambda_t \|\mathbf{M} - \mathbf{M}'\|_F + \nu_t \sum_{i=0}^{t-1} \varepsilon_i \|\mathbf{M}_1 - \mathbf{M}_2\|_F.$$

*Proof.* The norm of the gradient difference in (82) can be expanded as

$$\left\| \mathbb{E} \left[ \nabla_{\mathbf{M}} c_t \left( \mathbf{M}; \{\boldsymbol{\Delta}_i\}_{0 \leq i < t} \right) \right] - \nabla_{\mathbf{M}'} c_t \left( \mathbf{M}'; \{\boldsymbol{\Delta}_i'\}_{0 \leq i < t} \right) \right\|_F \tag{83}$$
$$\leq \left\| \mathbb{E} \left[ \nabla_{\mathbf{M}} c_t \left( \mathbf{M}; \{\boldsymbol{\Delta}_i\}_{0 \leq i < t} \right) - \nabla_{\mathbf{M}'} c_t \left( \mathbf{M}', \{\boldsymbol{\Delta}_i\}_{0 \leq i < t} \right) \right] \right\|_F$$
$$+ \left\| \mathbb{E} \left[ \nabla_{\mathbf{M}'} c_t \left( \mathbf{M}', \{\boldsymbol{\Delta}_i\}_{0 \leq i < t} \right) - \nabla_{\mathbf{M}'} c_t \left( \mathbf{M}', \boldsymbol{\Delta}_1', \{\boldsymbol{\Delta}_i\}_{1 \leq i < t} \right) \right] \right\|_F + \cdots$$
$$+ \left\| \mathbb{E} \left[ \nabla_{\mathbf{M}'} c_t \left( \mathbf{M}', \{\boldsymbol{\Delta}_j'\}_{0 \leq j \leq k-1}, \{\boldsymbol{\Delta}_i\}_{k \leq i < t} \right) - \nabla_{\mathbf{M}'} c_t \left( \mathbf{M}', \{\boldsymbol{\Delta}_j'\}_{0 \leq j \leq k}, \{\boldsymbol{\Delta}_i\}_{k+1 \leq i < t} \right) \right] \right\|_F$$
$$+ \cdots + \left\| \mathbb{E} \left[ \nabla_{\mathbf{M}'} c_t \left( \mathbf{M}', \{\boldsymbol{\Delta}_i'\}_{0 \leq i < t-1}, \boldsymbol{\Delta}_{t-1} \right) - \nabla_{\mathbf{M}'} c_t \left( \mathbf{M}', \{\boldsymbol{\Delta}_i'\}_{0 \leq i < t-1} \right) \right] \right\|_F,$$

where we drop the subscript in $\mathbb{E}[\cdot]$ since the associated randomness is clear from the context.

We next analyze a general term in R.H.S. of inequality (83). Specifically,

$$\left\| \mathbb{E}\left[ \nabla_{\mathbf{M}'} c_t \left( \mathbf{M}', \{\boldsymbol{\Delta}'_j\}_{0\le j\le k-1}, \{\boldsymbol{\Delta}_i\}_{k\le i<t} \right) - \nabla_{\mathbf{M}'} c_t \left( \mathbf{M}', \{\boldsymbol{\Delta}'_j\}_{0\le j\le k}, \{\boldsymbol{\Delta}_i\}_{k+1\le i<t} \right) \right] \right\|_F \tag{84}$$

$$= \left\| \mathbb{E}\left[ \mathbb{E}\left[ \nabla_{\mathbf{M}'} c_t \left( \mathbf{M}', \{\boldsymbol{\Delta}'_j\}_{0\le j\le k-1}, \boldsymbol{\Delta}_k, \{\boldsymbol{\Delta}_i\}_{k+1\le i<t} \right) \right.\right.\right.$$
$$\left.\left.\left. - \nabla_{\mathbf{M}'} c_t \left( \mathbf{M}', \{\boldsymbol{\Delta}'_j\}_{0\le j\le k-1}, \boldsymbol{\Delta}'_k, \{\boldsymbol{\Delta}_i\}_{k+1\le i<t} \right) \right| \{\boldsymbol{\Delta}'_j\}_{0\le j\le k-1}, \{\boldsymbol{\Delta}_i\}_{k+1\le i<t} \right] \right] \right\|_F$$

$$\le \mathbb{E}\left[ \left\| \mathbb{E}\left[ \nabla_{\mathbf{M}'} c_t \left( \mathbf{M}', \{\boldsymbol{\Delta}'_j\}_{0\le j\le k-1}, \boldsymbol{\Delta}_k, \{\boldsymbol{\Delta}_i\}_{k+1\le i<t} \right) \right.\right.\right.$$
$$\left.\left.\left. - \nabla_{\mathbf{M}'} c_t \left( \mathbf{M}', \{\boldsymbol{\Delta}'_j\}_{0\le j\le k-1}, \boldsymbol{\Delta}'_k, \{\boldsymbol{\Delta}_i\}_{k+1\le i<t} \right) \right| \{\boldsymbol{\Delta}'_j\}_{0\le j\le k-1}, \{\boldsymbol{\Delta}_i\}_{k+1\le i<t} \right] \right\|_F \right].$$

Given $\left\{ \{\boldsymbol{\Delta}'_j\}_{0\le j\le k-1}, \{\boldsymbol{\Delta}_i\}_{k+1\le i<t} \right\}$, for notation conciseness, we abbreviate

$\nabla_{\mathbf{M}'} c_t \left( \mathbf{M}', \{\boldsymbol{\Delta}'_j\}_{0\le j\le k-1}, \boldsymbol{\Delta}_k, \{\boldsymbol{\Delta}_i\}_{k+1\le i<t} \right)$ and $\nabla_{\mathbf{M}'} c_t \left( \mathbf{M}', \{\boldsymbol{\Delta}'_j\}_{0\le j\le k-1}, \boldsymbol{\Delta}'_k, \{\boldsymbol{\Delta}_i\}_{k+1\le i<t} \right)$
as $\nabla_{\mathbf{M}'} c_t \left( \mathbf{M}', \boldsymbol{\Delta}_k \right)$ and $\nabla_{\mathbf{M}'} c_t \left( \mathbf{M}', \boldsymbol{\Delta}'_k \right)$, respectively. It follows that

$$\left\| \mathbb{E}\left[ \nabla_{\mathbf{M}'} c_t \left( \mathbf{M}', \boldsymbol{\Delta}_k \right) - \nabla_{\mathbf{M}'} c_t \left( \mathbf{M}', \boldsymbol{\Delta}'_k \right) \right] \right\|_F^2 \tag{85}$$

$$= \mathrm{Tr}\left( \left( \mathbb{E}\left[ \nabla_{\mathbf{M}'} c_t \left( \mathbf{M}', \boldsymbol{\Delta}_k \right) \right] - \mathbb{E}\left[ \nabla_{\mathbf{M}'} c_t \left( \mathbf{M}', \boldsymbol{\Delta}'_k \right) \right] \right)^\top \left( \mathbb{E}\left[ \nabla_{\mathbf{M}'} c_t \left( \mathbf{M}', \boldsymbol{\Delta}_k \right) \right] - \mathbb{E}\left[ \nabla_{\mathbf{M}'} c_t \left( \mathbf{M}', \boldsymbol{\Delta}'_k \right) \right] \right) \right)$$

$$= \left\| \mathbb{E}\left[ \nabla_{\mathbf{M}'} c_t \left( \mathbf{M}', \boldsymbol{\Delta}_k \right) - \nabla_{\mathbf{M}'} c_t \left( \mathbf{M}', \boldsymbol{\Delta}'_k \right) \right] \right\|_F \mathrm{Tr}\left( V^\top \mathbb{E}\left[ \nabla_{\mathbf{M}'} c_t \left( \mathbf{M}', \boldsymbol{\Delta}_k \right) \right] - V^\top \mathbb{E}\left[ \nabla_{\mathbf{M}'} c_t \left( \mathbf{M}', \boldsymbol{\Delta}'_k \right) \right] \right),$$

where

$$V = \frac{\mathbb{E}\left[ \nabla_{\mathbf{M}'} c_t \left( \mathbf{M}', \boldsymbol{\Delta}_k \right) \right] - \mathbb{E}\left[ \nabla_{\mathbf{M}'} c_t \left( \mathbf{M}', \boldsymbol{\Delta}'_k \right) \right]}{\left\| \mathbb{E}\left[ \nabla_{\mathbf{M}'} c_t \left( \mathbf{M}', \boldsymbol{\Delta}_k \right) - \nabla_{\mathbf{M}'} c_t \left( \mathbf{M}', \boldsymbol{\Delta}'_k \right) \right] \right\|_F}, \ \|V\|_F = 1. \tag{86}$$

Based on Lemma B.6, given $\left\{ \{\boldsymbol{\Delta}'_j\}_{0\le j\le k-1}, \{\boldsymbol{\Delta}_i\}_{k+1\le i<t} \right\}$, the conditional gradient $V^\top \mathbb{E}\left[ \nabla_{\mathbf{M}'} c_t \left( \mathbf{M}', \boldsymbol{\Delta}_k \right) \right]$ is $\nu_t$-Lipschitz in $\boldsymbol{\Delta}_k$. Further note that $\boldsymbol{\Delta}_k \sim \mathcal{D}_k(\mathbf{M}_1)$ and $\boldsymbol{\Delta}'_k \sim \mathcal{D}_k(\mathbf{M}_2)$, which are both $\varepsilon_k$-sensitive. Applying Lemma B.7, it follows that

$$\mathrm{Tr}\left( V^\top \mathbb{E}\left[ \nabla_{\mathbf{M}'} c_t \left( \mathbf{M}', \boldsymbol{\Delta}_k \right) \right] - V^\top \mathbb{E}\left[ \nabla_{\mathbf{M}'} c_t \left( \mathbf{M}', \boldsymbol{\Delta}'_k \right) \right] \right) \le \nu_t \varepsilon_k \|\mathbf{M}_1 - \mathbf{M}_2\|_F. \tag{87}$$

Substitute (87) into (85), it follows that

$$\left\| \mathbb{E}\left[ \nabla_{\mathbf{M}'} c_t \left( \mathbf{M}', \boldsymbol{\Delta}_k \right) - \nabla_{\mathbf{M}'} c_t \left( \mathbf{M}', \boldsymbol{\Delta}'_k \right) \right] \right\|_F \le \nu_t \varepsilon_k \|\mathbf{M}_1 - \mathbf{M}_2\|_F. \tag{88}$$

Using the similar approach, we can obtain

$$\left\| \mathbb{E}\left[ \nabla_{\mathbf{M}} c_t \left( \mathbf{M}; \{\boldsymbol{\Delta}_i\}_{0\le i<t} \right) - \nabla_{\mathbf{M}'} c_t \left( \mathbf{M}', \{\boldsymbol{\Delta}_i\}_{0\le i<t} \right) \right] \right\| \le \lambda_t \|\mathbf{M} - \mathbf{M}'\|_F; \tag{89}$$

$$\left\| \mathbb{E}\left[ \nabla_{\mathbf{M}'} c_t \left( \mathbf{M}', \{\boldsymbol{\Delta}_i\}_{0\le i<t} \right) - \nabla_{\mathbf{M}'} c_t \left( \mathbf{M}', \mathbf{A}'_1, \{\boldsymbol{\Delta}_i\}_{1\le i<t} \right) \right] \right\| \le \nu_t \varepsilon_1 \|\mathbf{M}_1 - \mathbf{M}_2\|_F; \tag{90}$$

$$\vdots$$

$$\left\| \mathbb{E}\left[ \nabla_{\mathbf{M}'} c_t \left( \mathbf{M}', \{\boldsymbol{\Delta}'_i\}_{0\le i<t-1}, \boldsymbol{\Delta}_{t-1} \right) - \nabla_{\mathbf{M}'} c_t \left( \mathbf{M}', \{\boldsymbol{\Delta}'_i\}_{0\le i<t-1} \right) \right] \right\|$$
$$\le \nu_t \varepsilon_{t-1} \|\mathbf{M}_1 - \mathbf{M}_2\|_F. \tag{91}$$

Combining (83), (89)-(91), it follow inequality (82). □

## C  Properties of the Total Cost Function

We define a total cost function $J_T$ as

$$J_T\left( \mathbf{M}; \{\boldsymbol{\Delta}_t\}_{0\le t<T} \right) = \sum_{t=0}^{T} c_t\left( \mathbf{M}; \{\boldsymbol{\Delta}_i\}_{0\le i<t} \right). \tag{92}$$

It is clear that

$$C_T\left(\mathbf{M};\mathbf{M}_1\right) = \mathbb{E}_{\mathbf{x}_0,\{\boldsymbol{\Delta}_t\sim\mathcal{D}_t(\mathbf{M}_1)\}_{0\leq i<T},\{\mathbf{w}_i\}_{0\leq i<T}} J_T\left(\mathbf{M};\{\boldsymbol{\Delta}_t\}_{0\leq t<T}\right). \tag{93}$$

We next analyze the gradient properties and the smoothness of the total cost function $J_T\left(\mathbf{M};\{\boldsymbol{\Delta}_t\}_{0\leq t<T}\right)$. We again note that $\nabla J_T\left(\mathbf{M};\{\boldsymbol{\Delta}_t\}_{0\leq t<T}\right)$ denotes the gradient taken w.r.t. the first argument $\mathbf{M}$.

**Lemma C.1** *The variance of the gradient of the total cost function satisfies*

$$\mathbb{E}_{\mathbf{x}_0,\{\boldsymbol{\Delta}_t\sim\mathcal{D}_t(\mathbf{M})\}_{0\leq i<T},\{\mathbf{w}_i\}_{0\leq i<T}}\left[\left\|\nabla J_T\left(\mathbf{M};\{\boldsymbol{\Delta}_t\}_{0\leq t<T}\right) - \nabla C_T\left(\mathbf{M};\mathbf{M}\right)\right\|_F^2\right] \leq T\sum_{t=1}^T \vartheta_t^2. \tag{94}$$

*Proof.* Note that $\nabla J_0\left(\mathbf{M};\boldsymbol{\Delta}_0\right) = \nabla_{\mathbf{M}}c_0\left(\mathbf{x}_0,\mathbf{u}_0\right) = \nabla_{\mathbf{M}}c_0\left(\mathbf{x}_0,-\mathbf{K}\mathbf{x}_0\right) = \mathbf{0}$. Inequality (94) then follows directly from the norm triangular inequality:

$$\mathbb{E}\left[\left\|\nabla J_T\left(\mathbf{M};\{\boldsymbol{\Delta}_t\}_{0\leq t<T}\right) - \mathbb{E}\left[\nabla J_T\left(\mathbf{M};\{\boldsymbol{\Delta}_t\}_{0\leq t<T}\right)\right]\right\|_F^2\right] \tag{95}$$

$$\leq T\sum_{t=1}^T \mathbb{E}_{\mathbf{x}_0,\{\boldsymbol{\Delta}_i\sim\mathcal{D}_i(\mathbf{M})\}_{0\leq i<t},\{\mathbf{w}_i\}_{0\leq i<t}}\left[\left\|\nabla_{\mathbf{M}}c_t\left(\mathbf{x}_t^{(\mathbf{M})},\mathbf{u}_t^{(\mathbf{M})}\right)\right\|\right.$$

$$\left. - \mathbb{E}_{\mathbf{x}_0,\{\boldsymbol{\Delta}_i\sim\mathcal{D}_i(\mathbf{M})\}_{0\leq i<t},\{\mathbf{w}_i\}_{0\leq i<t}}\left[\nabla_{\mathbf{M}}c_t\left(\mathbf{x}_t^{(\mathbf{M})},\mathbf{u}_t^{(\mathbf{M})}\right)\right]\right\|_F^2\right]$$

$$\overset{(95.a)}{\leq} T\sum_{t=1}^T \vartheta_t^2,$$

where inequality $(95.a)$ holds becasue of Lemma B.4. $\qquad\square$

## D   Proof of Lemma 2

Denote $f_t\left(\mathbf{M};\mathbf{M}_1\right) = \mathbb{E}_{\mathbf{x}_0,\{\boldsymbol{\Delta}_i\sim\mathcal{D}_i(\mathbf{M}_1)\}_{0\leq i<t},\{\mathbf{w}_i\}_{0\leq i<t}}\left[c_t\left(\mathbf{M};\{\boldsymbol{\Delta}_i\}_{0\leq i<t}\right)\right]$ as the per stage expected cost with the distribution of policy-dependent perturbation is shifted from $\boldsymbol{\Delta}_i\sim\mathcal{D}_i\left(\mathbf{M}\right)$ to $\boldsymbol{\Delta}_i\sim\mathcal{D}_i\left(\mathbf{M}_1\right), \forall 0\leq i<t$.

From the convexity of per stage expected cost in Lemma B.1, we know that, $\forall 1\leq t_1<H$, $f_{t_1}\left(\mathbf{M};\mathbf{M}_1\right)$ is a convex function of $\mathbf{M}$. It follows that

$$f_{t_1}\left(\mathbf{M};\mathbf{M}_1\right) \geq f_{t_1}\left(\mathbf{M}';\mathbf{M}_1\right) + \mathrm{Tr}\left(\left(\nabla f_{t_1}\left(\mathbf{M}';\mathbf{M}_1\right)\right)^\top\left(\mathbf{M}-\mathbf{M}'\right)\right), \forall 1\leq t_1<H, \tag{96}$$

where the gradient $\nabla f_{t_1}\left(\mathbf{M}';\mathbf{M}_1\right)$ is taken w.r.t. the first argument $\mathbf{M}'$.

From stong convexity of per stage expected cost in Lemma B.2, we know that, $\forall H\leq t_2\leq T$, $f_{t_2}\left(\mathbf{M};\mathbf{M}_1\right)$ is $\min\left\{\frac{\mu\sigma^2}{2},\frac{\mu\gamma^2\sigma^2}{64\kappa^{10}}\right\}$-strongly convex function of $\mathbf{M}$. Therefore, it follows

$$f_{t_2}\left(\mathbf{M};\mathbf{M}_1\right) \geq f_{t_2}\left(\mathbf{M}';\mathbf{M}_1\right) + \mathrm{Tr}\left(\left(\nabla f_{t_2}\left(\mathbf{M}';\mathbf{M}_1\right)\right)^\top\left(\mathbf{M}-\mathbf{M}'\right)\right) \tag{97}$$

$$+ \min\left\{\frac{\mu\sigma^2}{2},\frac{\mu\gamma^2\sigma^2}{64\kappa^{10}}\right\}\left\|\mathbf{M}-\mathbf{M}'\right\|_F^2, \forall H\leq t_2\leq T.$$

Sum up (96) from $t_1 = 1$ to $t_1 = H - 1$ and (97) from $t_2 = H$ to $t_2 = T$, we obatin

$$\sum_{t_1=1}^{H-1} f_{t_1}(\mathbf{M}; \mathbf{M}_1) + \sum_{t_2=H}^{T} f_{t_1}(\mathbf{M}; \mathbf{M}_1) \geq \sum_{t_1=1}^{H-1} f_{t_1}(\mathbf{M}'; \mathbf{M}_1) + \sum_{t_2=H}^{T} f_{t_1}(\mathbf{M}'; \mathbf{M}_1) \qquad (98)$$

$$+ \mathrm{Tr}\left(\left(\nabla\left(\sum_{t_1=1}^{H-1} f_{t_1}(\mathbf{M}'; \mathbf{M}_1) + \sum_{t_2=H}^{T} f_{t_1}(\mathbf{M}'; \mathbf{M}_1)\right)\right)^{\top}(\mathbf{M} - \mathbf{M}')\right)$$

$$+ \frac{\widetilde{\mu}}{2}\|\mathbf{M} - \mathbf{M}'\|_F^2.$$

Inequality (13) follows directly by noting that

$$C_T(\mathbf{M}; \mathbf{M}_1) = \sum_{t_1=1}^{H-1} f_{t_1}(\mathbf{M}; \mathbf{M}_1) + \sum_{t_2=H}^{T} f_{t_1}(\mathbf{M}; \mathbf{M}_1), \qquad (99)$$

$$\nabla C_T(\mathbf{M}'; \mathbf{M}_1) = \nabla\left(\sum_{t_1=1}^{H-1} f_{t_1}(\mathbf{M}'; \mathbf{M}_1) + \sum_{t_2=H}^{T} f_{t_1}(\mathbf{M}'; \mathbf{M}_1)\right). \qquad (100)$$

## E   Proof of Lemma 3

Note that $\nabla C_0(\mathbf{M}; \mathbf{M}_1) = \nabla_{\mathbf{M}}\mathbb{E}[c_t(\mathbf{x}_0, \mathbf{u}_0)] = \nabla_{\mathbf{M}}\mathbb{E}[c_t(\mathbf{x}_0, -\mathbf{K}\mathbf{x}_0)] = \mathbf{0}$. Therefore, inequality (14) follows directly by combining the norm triangular inequalityand the results in Lemma B.6.

$$\|\nabla C_T(\mathbf{M}; \mathbf{M}_1) - \nabla C_T(\mathbf{M}'; \mathbf{M}_2)\|_F \qquad (101)$$

$$\leq \sum_{t=1}^{T}\left\|\mathbb{E}_{\mathbf{x}_0, \{\boldsymbol{\Delta}_i \sim \mathcal{D}_i(\mathbf{M}_1)\}_{0\leq i<t}, \{\mathbf{w}_i\}_{0\leq i<t}}\left[\nabla_{\mathbf{M}}c_t\left(\mathbf{M}; \{\boldsymbol{\Delta}_i\}_{0\leq i<t}\right)\right]\right.$$

$$\left. -\mathbb{E}_{\mathbf{x}_0, \{\boldsymbol{\Delta}'_i \sim \mathcal{D}_i(\mathbf{M}_2)\}_{0\leq i<t}, \{\mathbf{w}_i\}_{0\leq i<t}}\left[\nabla_{\mathbf{M}'}c_t\left(\mathbf{M}'; \{\boldsymbol{\Delta}'_i\}_{0\leq i<t}\right)\right]\right\|_F$$

$$\leq \left(\sum_{t=1}^{T}\lambda_t\right)\|\mathbf{M} - \mathbf{M}'\|_F + \sum_{t=1}^{T}\left(\nu_t\sum_{i=0}^{t-1}\varepsilon_i\right)\|\mathbf{M}_1 - \mathbf{M}_2\|_F$$

$$= \left(\sum_{t=1}^{T}\lambda_t\right)\|\mathbf{M} - \mathbf{M}'\|_F + \sum_{t=0}^{T-1}\varepsilon_t\left(\sum_{i=t+1}^{T}\nu_i\right)\|\mathbf{M}_1 - \mathbf{M}_2\|_F.$$

## F   Proof of Lemma 4

**Proposition 3** *(First-order optimality condition in [18]). Let $F(\boldsymbol{x})$ be convex and let $\mathcal{X}^n$ be a closed convex set on which $F(\boldsymbol{x})$ is differentiable, then*

$$\boldsymbol{x}^* \in \operatorname*{arg\,min}_{\boldsymbol{x} \in \mathcal{X}^n} F(\boldsymbol{x})$$

*if and only if*

$$\nabla F(\boldsymbol{x}^*)^{\top}(\boldsymbol{y} - \boldsymbol{x}^*) \geqslant 0, \quad \forall \boldsymbol{y} \in \mathcal{X}^n.$$

Fix any $\mathbf{M}, \mathbf{M}' \in \mathbb{M}$, applying the first order optimality condition in Proposition 3, the optimality condition to (15) implies that

$$\langle \nabla C_T(\Phi(\mathbf{M}); \mathbf{M}), \Phi(\mathbf{M}') - \Phi(\mathbf{M})\rangle \geq 0, \langle \nabla C_T(\Phi(\mathbf{M}'); \mathbf{M}'), \Phi(\mathbf{M}) - \Phi(\mathbf{M}')\rangle \geq 0, \quad (102)$$

where the gradients are again taken w.r.t. the first argument in the function $C_T(\cdot, \cdot)$. Observe the fact

$$0 \leq \langle \nabla C_T(\Phi(\mathbf{M}'); \mathbf{M}') - \nabla C_T(\Phi(\mathbf{M}); \mathbf{M}), \Phi(\mathbf{M}) - \Phi(\mathbf{M}')\rangle. \qquad (103)$$

Adding and subtracting the term $\nabla C_T \left( \Phi \left( \mathbf{M} \right) ; \mathbf{M}' \right)$ implies the equality

$$\mathrm{Tr} \left( \left( \nabla C_T \left( \Phi \left( \mathbf{M} \right) ; \mathbf{M}' \right) - \nabla C_T \left( \Phi \left( \mathbf{M} \right) ; \mathbf{M} \right) \right)^{\top} \left( \Phi \left( \mathbf{M} \right) - \Phi \left( \mathbf{M}' \right) \right) \right) \tag{104}$$

$$\geq \mathrm{Tr} \left( \left( \nabla C_T \left( \Phi \left( \mathbf{M} \right) ; \mathbf{M}' \right) - \nabla C_T \left( \Phi \left( \mathbf{M}' \right) ; \mathbf{M}' \right) \right)^{\top} \left( \Phi \left( \mathbf{M} \right) - \Phi \left( \mathbf{M}' \right) \right) \right).$$

Recall that $C_T \left( \Phi \left( \mathbf{M} \right) ; \mathbf{M}' \right)$ is $\widetilde{\mu}$-strongly convex w.r.t. $\Phi \left( \mathbf{M} \right)$, it follows

$$\mathrm{Tr} \left( \left( \nabla C_T \left( \Phi \left( \mathbf{M} \right) ; \mathbf{M}' \right) - \nabla C_T \left( \Phi \left( \mathbf{M}' \right) ; \mathbf{M}' \right) \right)^{\top} \left( \Phi \left( \mathbf{M} \right) - \Phi \left( \mathbf{M}' \right) \right) \right) \tag{105}$$

$$\geq \widetilde{\mu} \left\| \Phi \left( \mathbf{M} \right) - \Phi \left( \mathbf{M}' \right) \right\|_F^2 .$$

Meanwhile, applying Lemma 3 to the left hand side of (104) gives

$$\mathrm{Tr} \left( \left( \nabla C_T \left( \Phi \left( \mathbf{M} \right) ; \mathbf{M}' \right) - \nabla C_T \left( \Phi \left( \mathbf{M} \right) ; \mathbf{M} \right) \right)^{\top} \left( \Phi \left( \mathbf{M} \right) - \Phi \left( \mathbf{M}' \right) \right) \right) \tag{106}$$

$$\leq \sum_{t=0}^{T-1} \left( \varepsilon_t \sum_{i=t+1}^{T} \nu_i \right) \left\| \mathbf{M} - \mathbf{M}' \right\|_F \left\| \Phi \left( \mathbf{M} \right) - \Phi \left( \mathbf{M}' \right) \right\|_F .$$

Combining (104), (105) and (106), it follows

$$\left\| \Phi \left( \mathbf{M} \right) - \Phi \left( \mathbf{M}' \right) \right\|_F \leq \frac{\sum_{t=0}^{T-1} \left( \varepsilon_t \sum_{i=t+1}^{T} \nu_i \right)}{\widetilde{\mu}} \left\| \mathbf{M} - \mathbf{M}' \right\|_F . \tag{107}$$

We note that $\mathbf{M}_{n+1} = \Phi \left( \mathbf{M}_n \right)$ by the definition of (15), and $\mathbf{M}^{PS} = \Phi \left( \mathbf{M}^{PS} \right)$ by the definition of performative stability. Applying (107) yields

$$\left\| \mathbf{M}_n - \mathbf{M}^{PS} \right\|_F = \left\| \Phi \left( \mathbf{M}_{n-1} \right) - \Phi \left( \mathbf{M}^{PS} \right) \right\|_F \leq \frac{\sum_{t=0}^{T-1} \left( \varepsilon_t \sum_{i=t+1}^{T} \nu_i \right)}{\widetilde{\mu}} \left\| \mathbf{M}_{l-1} - \mathbf{M}^{PS} \right\|_F$$

$$\tag{108}$$

$$\leq \left( \frac{\sum_{t=0}^{T-1} \left( \varepsilon_t \sum_{i=t+1}^{T} \nu_i \right)}{\widetilde{\mu}} \right)^n \left\| \mathbf{M}_0 - \mathbf{M}^{PS} \right\|_F .$$

Setting $\left( \frac{\sum_{t=0}^{T-1} \left( \varepsilon_t \sum_{i=t+1}^{T} \nu_i \right)}{\widetilde{\mu}} \right)^n \left\| \mathbf{M}_0 - \mathbf{M}^{PS} \right\|_F$ to be at most $\rho$ and solving for $n$ completes the proof.

# G Convergence Analysis of Algorithm 1

## G.1 Non-asymptomatic Convergence Analysis for Algorithm 1 with General Step Sizes

We have the following Lemma 5 regarding the convergence results of the proposed RSGD in Algorithm 1.

**Lemma 5** *Under A1-A6. Consider a sequence of non-negative step sizes $\{ \eta_n, n \geq 0 \}$ satisfy*

$$\sup_{n \geq 0} \eta_n \leq \min \left\{ \frac{\widetilde{\mu} - \sum_{t=0}^{T-1} \left( \varepsilon_t \sum_{i=t+1}^{T} \nu_i \right)}{2 \left( \sum_{t=1}^{T} \lambda_t + \sum_{t=0}^{T-1} \left( \varepsilon_t \sum_{i=t+1}^{T} \nu_i \right) \right)^2}, \frac{2}{\widetilde{\mu} - \sum_{t=0}^{T-1} \left( \varepsilon_t \sum_{i=t+1}^{T} \nu_i \right)} \right\}, \tag{109}$$

*and*

$$\frac{\eta_n}{\eta_{n+1}} \leq \left( 1 + \frac{1}{2} \left( \widetilde{\mu} - \sum_{t=0}^{T-1} \left( \varepsilon_t \sum_{i=t+1}^{T} \nu_i \right) \right) \eta_{n+1} \right), \tag{110}$$

*for any $n \geq 0$. Then, the iterates generated by RSGD admit the following bound for any $N \geq 1$:*

$$\mathbb{E}\left[\left\|\mathbf{M}_N - \mathbf{M}^{PS}\right\|_F^2\right] \leq \prod_{n=0}^{N-1}\left(1 - \eta_n\left(\widetilde{\mu} - \sum_{t=0}^{T-1}\left(\varepsilon_t \sum_{i=t+1}^{T}\nu_i\right)\right)\right)\mathbb{E}\left[\left\|\mathbf{M}_0 - \mathbf{M}^{PS}\right\|_F^2\right] \quad (111)$$

$$+ \frac{4\eta_{N-1}T\sum_{t=1}^{T}\vartheta_t^2}{\widetilde{\mu} - \sum_{t=0}^{T-1}\left(\varepsilon_t \sum_{i=t+1}^{T}\nu_i\right)},$$

*where $\vartheta_t = \kappa^3 G\left(\left(HW + \kappa^2\right)\kappa\|\mathbf{B}\|\beta_t + 1\right)\left(x_0\alpha_t + c_3\beta_t\right) + GHWM\left(\kappa^3\beta_t + 1\right), \forall 1 \leq t \leq T$.*

*Proof.*   Since projecting onto a convex set can only bring two iterates closer together, it follows that

$$\mathbb{E}\left[\left\|\mathbf{M}_{n+1} - \mathbf{M}^{PS}\right\|_F^2\right] \leq \mathbb{E}\left[\left\|\mathbf{M}_n - \eta_n\nabla J_T\left(\mathbf{M}_n; \{\mathbf{\Delta}_t\}_{0 \leq t < T}\right) - \mathbf{M}^{PS}\right\|^2\right] \quad (112)$$

$$\leq I_1 + I_2 + I_3,$$

where

$$I_1 = \mathbb{E}\left[\left\|\mathbf{M}_n - \mathbf{M}^{PS}\right\|_F^2\right], \quad (113)$$

$$I_2 = -2\eta_n\mathbb{E}\left[\mathrm{Tr}\left(\left(\mathbf{M}_n - \mathbf{M}^{PS}\right)^\top\nabla J_T\left(\mathbf{M}_n; \{\mathbf{\Delta}_t\}_{0 \leq t < T}\right)\right)\right], \quad (114)$$

$$I_3 = \eta_n^2\mathbb{E}\left[\left\|\nabla J_T\left(\mathbf{M}_n; \{\mathbf{\Delta}_t\}_{0 \leq t < T}\right)\right\|_F^2\right], \quad (115)$$

and $\mathbf{\Delta}_t \sim \mathcal{D}_t\left(\mathbf{M}_n\right), \forall 0 \leq t < T$.

We first analyze the term $I_2$. Let $\mathbb{E}_n\left[\cdot\right]$ be the conditional expectation on $\mathbf{M}_n$. We have

$$\mathbb{E}_n\left[\mathrm{Tr}\left(\left(\mathbf{M}_n - \mathbf{M}^{PS}\right)^\top\nabla J_T\left(\mathbf{M}_n; \{\mathbf{\Delta}_t\}_{0 \leq t < T}\right)\right)\right] = \mathrm{Tr}\left(\left(\mathbf{M}_n - \mathbf{M}^{PS}\right)^\top\nabla C_T\left(\mathbf{M}_n; \mathbf{M}_n\right)\right)$$
$$(116)$$

$$\geq \mathrm{Tr}\left(\left(\mathbf{M}_n - \mathbf{M}^{PS}\right)^\top\left(\nabla C_T\left(\mathbf{M}_n; \mathbf{M}_n\right) - \nabla C_T\left(\mathbf{M}_n; \mathbf{M}^{PS}\right)\right)\right)$$

$$+ \mathrm{Tr}\left(\left(\mathbf{M}_n - \mathbf{M}^{PS}\right)^\top\left(\nabla C_T\left(\mathbf{M}_n; \mathbf{M}^{PS}\right) - \nabla C_T\left(\mathbf{M}^{PS}; \mathbf{M}^{PS}\right)\right)\right),$$

where we use the first order optimality condition in Proposition 3 that $\langle\nabla C_T\left(\mathbf{M}^{PS}; \mathbf{M}^{PS}\right), \mathbf{M}_n - \mathbf{M}^{PS}\rangle \geq 0$ in the last equality of (116).

Applying the Cauchy-Schwarz inequality and Lemma 3, we obtain

$$\mathrm{Tr}\left(\left(\mathbf{M}_n - \mathbf{M}^{PS}\right)^\top\left(\nabla C_T\left(\mathbf{M}_n; \mathbf{M}_n\right) - \nabla C_T\left(\mathbf{M}_n; \mathbf{M}^{PS}\right)\right)\right) \quad (117)$$

$$\geq -\left\|\mathbf{M}_n - \mathbf{M}^{PS}\right\|_F\left(\sum_{t=0}^{T-1}\left(\varepsilon_t \sum_{i=t+1}^{T}\nu_i\right)\left\|\mathbf{M}_n - \mathbf{M}^{PS}\right\|_F\right)$$

$$= -\sum_{t=0}^{T-1}\left(\varepsilon_t \sum_{i=t+1}^{T}\nu_i\right)\left\|\mathbf{M}_n - \mathbf{M}^{PS}\right\|_F^2.$$

Meanwhile, based on strongly convexity of $C_T$ in Lemma 2, we have

$$\mathrm{Tr}\left(\left(\mathbf{M}_n - \mathbf{M}^{PS}\right)^\top\left(\nabla C_T\left(\mathbf{M}_n; \mathbf{M}^{PS}\right) - \nabla C_T\left(\mathbf{M}^{PS}; \mathbf{M}^{PS}\right)\right)\right) \geq \widetilde{\mu}\left\|\mathbf{M}_n - \mathbf{M}^{PS}\right\|_F^2.$$
$$(118)$$

Substitute (117) and (118) into (116) and take the full expectation, it follows that $I_2$ satisfies

$$I_2 \leq -2\eta_n\left(\widetilde{\mu} - \sum_{t=0}^{T-1}\left(\varepsilon_t \sum_{i=t+1}^{T}\nu_i\right)\right)\mathbb{E}\left[\left\|\mathbf{M}_n - \mathbf{M}^{PS}\right\|_F^2\right]. \quad (119)$$

We next analyze the term $I_3$. We observe the following equivalent expression for $I_3$

$$
\left\| \nabla J_T \left( \mathbf{M}_n; \{\boldsymbol{\Delta}_t\}_{0 \le t < T} \right) \right\|_F^2 = \left\| \nabla J_T \left( \mathbf{M}_n; \{\boldsymbol{\Delta}_t\}_{0 \le t < T} \right) - \nabla C_T \left( \mathbf{M}_n; \mathbf{M}_n \right) \right. \tag{120}
$$
$$
\left. + \nabla C_T \left( \mathbf{M}_n; \mathbf{M}_n \right) - \nabla C_T \left( \mathbf{M}^{PS}; \mathbf{M}^{PS} \right) \right\|_F^2 .
$$

Therefore,

$$
\mathbb{E} \left[ \left\| \nabla J_T \left( \mathbf{M}_n; \{\boldsymbol{\Delta}_t\}_{0 \le t < T} \right) \right\|_F^2 \right] \tag{121}
$$
$$
\le 2 \mathbb{E} \left[ \left\| \nabla J_T \left( \mathbf{M}_l; \{\boldsymbol{\Delta}_t\}_{0 \le t < T} \right) - \nabla C_T \left( \mathbf{M}_n; \mathbf{M}_n \right) \right\|_F^2 \right]
$$
$$
+ 2 \mathbb{E} \left[ \left\| \nabla C_T \left( \mathbf{M}_n; \mathbf{M}_n \right) - \nabla C_T \left( \mathbf{M}^{PS}; \mathbf{M}^{PS} \right) \right\|_F^2 \right] .
$$

According to Lemma C.1, an upper bound of the variance of $\nabla J_T \left( \mathbf{M}_n; \{\boldsymbol{\Delta}_t\}_{0 \le t < T} \right)$ is given by (94). Moreover, based on Lemma 3, we have

$$
\mathbb{E} \left[ \left\| \nabla C_T \left( \mathbf{M}_n; \mathbf{M}_n \right) - \nabla C_T \left( \mathbf{M}^{PS}; \mathbf{M}^{PS} \right) \right\|_F^2 \right] \tag{122}
$$
$$
\le \left( \sum_{t=1}^{T} \lambda_t + \sum_{t=0}^{T-1} \left( \varepsilon_t \sum_{i=t+1}^{T} \nu_i \right) \right)^2 \mathbb{E} \left[ \left\| \mathbf{M}_n - \mathbf{M}^{PS} \right\|_F^2 \right] .
$$

Substitute (94) and (122) back into (121), it follows

$$
I_3 \le 2\eta_n^2 T \sum_{t=1}^{T} \vartheta_t^2 + 2\eta_n^2 \left( \sum_{t=1}^{T} \lambda_t + \sum_{t=0}^{T-1} \left( \varepsilon_t \sum_{i=t+1}^{T} \nu_i \right) \right)^2 \mathbb{E} \left[ \left\| \mathbf{M}_n - \mathbf{M}^{PS} \right\|_F^2 \right] . \tag{123}
$$

Now subtitute (119) and (123) back into (112), we obtain

$$
\mathbb{E} \left[ \left\| \mathbf{M}_{n+1} - \mathbf{M}^{PS} \right\|_F^2 \right] \le \mathbb{E} \left[ \left\| \mathbf{M}_n - \mathbf{M}^{PS} \right\|_F^2 \right] - 2\eta_n \left( \widetilde{\mu} - \sum_{t=0}^{T-1} \left( \varepsilon_t \sum_{i=t+1}^{T} \nu_i \right) \right) \tag{124}
$$
$$
\cdot \mathbb{E} \left[ \left\| \mathbf{M}_n - \mathbf{M}^{PS} \right\|_F^2 \right] + 2\eta_n^2 \left( \sum_{t=1}^{T} \lambda_t + \sum_{t=0}^{T-1} \left( \varepsilon_t \sum_{i=t+1}^{T} \nu_i \right) \right)^2 \mathbb{E} \left[ \left\| \mathbf{M}_n - \mathbf{M}^{PS} \right\|_F^2 \right] + 2\eta_n^2 T \sum_{t=1}^{T} \vartheta_t^2
$$
$$
= \left( 1 - 2\eta_n \left( \widetilde{\mu} - \sum_{t=0}^{T-1} \left( \varepsilon_t \sum_{i=t+1}^{T} \nu_i \right) \right) + 2\eta_n^2 \left( \sum_{t=1}^{T} \lambda_t + \sum_{t=0}^{T-1} \left( \varepsilon_t \sum_{i=t+1}^{T} \nu_i \right) \right)^2 \right) \mathbb{E} \left[ \left\| \mathbf{M}_n - \mathbf{M}^{PS} \right\|_F^2 \right]
$$
$$
+ 2\eta_n^2 T \sum_{t=1}^{T} \vartheta_t^2 .
$$

Let $\eta_n$ be choosen such that $\eta_n \left( \widetilde{\mu} - \sum_{t=0}^{T-1} \left( \varepsilon_t \sum_{i=t+1}^{T} \nu_i \right) \right) \ge 2\eta_n^2 \left( \sum_{t=1}^{T} \lambda_t + \sum_{t=0}^{T-1} \left( \varepsilon_t \sum_{i=t+1}^{T} \nu_i \right) \right)^2$. Then inequality (124) is reduced to

$$
\mathbb{E} \left[ \left\| \mathbf{M}_{n+1} - \mathbf{M}^{PS} \right\|_F^2 \right] \le \left( 1 - \eta_n \left( \widetilde{\mu} - \sum_{t=0}^{T-1} \left( \varepsilon_t \sum_{i=t+1}^{T} \nu_i \right) \right) \right) \mathbb{E} \left[ \left\| \mathbf{M}_n - \mathbf{M}^{PS} \right\|_F^2 \right] \tag{125}
$$
$$
+ 2\eta_n^2 T \sum_{t=1}^{T} \vartheta_t^2 .
$$

Therefore,

$$\mathbb{E}\left[\left\|\mathbf{M}_N - \mathbf{M}^{PS}\right\|_F^2\right] \leq \prod_{n=0}^{N-1}\left(1 - \eta_n\left(\widetilde{\mu} - \sum_{t=0}^{T-1}\left(\varepsilon_t\sum_{i=t+1}^{T}\nu_i\right)\right)\right)\mathbb{E}\left[\left\|\mathbf{M}_0 - \mathbf{M}^{PS}\right\|_F^2\right] \quad (126)$$

$$+ \sum_{n=0}^{N-1}\prod_{i=n+1}^{N-1}\left(1 - \eta_i\left(\widetilde{\mu} - \sum_{t=0}^{T-1}\left(\varepsilon_t\sum_{i=t+1}^{T}\nu_i\right)\right)\right)2\eta_n^2 T\sum_{t=1}^{T}\vartheta_t^2.$$

Let $\eta_0 < \frac{2}{\widetilde{\mu} - \sum_{t=0}^{T-1}\left(\varepsilon_t\sum_{i=t+1}^{T}\nu_i\right)}$. Based on Lemma 6 in [14], if $\frac{\eta_n}{\eta_{n+1}} \leq$ $\left(1 + \frac{1}{2}\left(\widetilde{\mu} - \sum_{t=0}^{T-1}\left(\varepsilon_t\sum_{i=t+1}^{T}\nu_i\right)\right)\eta_{n+1}\right)$ for any $n \geq 0$, then

$$\sum_{n=0}^{N-1}\eta_n^2\prod_{i=n+1}^{N-1}\left(1 - \eta_i\left(\widetilde{\mu} - \sum_{t=0}^{T-1}\left(\varepsilon_t\sum_{i=t+1}^{T}\nu_i\right)\right)\right) \leq \frac{2\eta_{N-1}}{\widetilde{\mu} - \sum_{t=0}^{T-1}\left(\varepsilon_t\sum_{i=t+1}^{T}\nu_i\right)}, \forall N \geq 1.$$
$$(127)$$

Substitute (127) into (126), we obtain the desired result. $\qquad\square$

Based on the step size requirements in Lemma 5, RSGD converges when the aggregated sensitivity $\sum_{t=0}^{T-1}\left(\varepsilon_t\sum_{i=t+1}^{T}\nu_i\right)$ is strictly below the threshold $\widetilde{\mu}$ defined in Lemma 4, i.e., the same sufficient condition that guarantees the existence of $\mathbf{M}^{PS}$.

The first term on the R. H. S. of (111) decays sub-geometrically and is scaled by the initial error $\mathbb{E}\left[\left\|\mathbf{M}_0 - \mathbf{M}^{PS}\right\|_F^2\right]$. The second term is a fluctuation term that only depends on the variance of the stochastic gradient, which decays at a slower rate as $\mathcal{O}\left(\eta_{N-1}\right)$.

### G.2 Convergence Analysis of Algorithm 1 with Diminishing Step Sizes

We select the diminishing step sizes $\eta_n = \frac{\phi_1}{n+\phi_2}, \forall n \geq 0$, where $\phi_1$ and $\phi_2$ satisfy (17) and (18) simultaneously. We can verify that condition (109) in Lemma 5 is reduced to (17) in Theorem 1. Besides, condition (110) in Lemma 5 can be further simplifed as

$$\frac{\eta_n}{\eta_{n+1}} = \frac{\frac{\phi_1}{n+\phi_2}}{\frac{\phi_1}{n+1+\phi_2}} = 1 + \frac{1}{n+\phi_2} \leq \left(1 + \frac{1}{2}\left(\widetilde{\mu} - \sum_{t=0}^{T-1}\left(\varepsilon_t\sum_{i=t+1}^{T}\nu_i\right)\right)\frac{\phi_1}{n+1+\phi_2}\right), \quad (128)$$

which is equivalent to

$$\frac{n+1+\phi_2}{n+\phi_2} \leq \frac{1}{2}\left(\widetilde{\mu} - \sum_{t=0}^{T-1}\left(\varepsilon_t\sum_{i=t+1}^{T}\nu_i\right)\right)\phi_1. \quad (129)$$

Note that $\forall n \geq 0$, to guarantee (129) holds, we only need to let

$$\frac{2}{\left(\widetilde{\mu} - \sum_{t=0}^{T-1}\left(\varepsilon_t\sum_{i=t+1}^{T}\nu_i\right)\right)} \leq \frac{\phi_1\phi_2}{1+\phi_2} = \frac{\phi_1}{1+\frac{1}{\phi_2}}, \quad (130)$$

which coincides with condition (18) in Theorem 1.

Lastly, not that $1 + x \leq e^x$ for all $x > 0$, we can verify the first term in R.H.S. of (111) can be upper bounded as

$$\prod_{n=0}^{N-1}\left(1 - \eta_n\left(\widetilde{\mu} - \sum_{t=0}^{T-1}\left(\varepsilon_t\sum_{i=t+1}^{T}\nu_i\right)\right)\right)\mathbb{E}\left[\left\|\mathbf{M}_0 - \mathbf{M}^{PS}\right\|_F^2\right]$$

$$\leq e^{-\sum_{n=0}^{N-1}\eta_n\left(\widetilde{\mu}-\sum_{t=0}^{T-1}\left(\varepsilon_t\sum_{i=t+1}^{T}\nu_i\right)\right)}\mathbb{E}\left[\left\|\mathbf{M}_0 - \mathbf{M}^{PS}\right\|_F^2\right]$$

$$= e^{-\sum_{n=0}^{N-1}\frac{\phi_1}{n+\phi_2}\left(\widetilde{\mu}-\sum_{t=0}^{T-1}\left(\varepsilon_t\sum_{i=t+1}^{T}\nu_i\right)\right)}\mathbb{E}\left[\left\|\mathbf{M}_0 - \mathbf{M}^{PS}\right\|_F^2\right]$$

$$\leq e^{-\sum_{n=1}^{N}\frac{\phi_1}{n}\left(\widetilde{\mu}-\sum_{t=0}^{T-1}\left(\varepsilon_t\sum_{i=t+1}^{T}\nu_i\right)\right)}\mathbb{E}\left[\left\|\mathbf{M}_0 - \mathbf{M}^{PS}\right\|_F^2\right]. \quad (131)$$

Therefore, Theorem 1 follows by combining (130) and (131).

# H  Proof of Proposition 1

Since $\|\mathbf{A}_t\| \leq 1 - \gamma + \kappa^2 \xi_t \leq \zeta < 1, \forall 0 \leq t < T$, it follows directly that $\alpha_t = \prod_{i=0}^{t-1} \left(1 - \gamma + \kappa^2 \xi_i\right) \leq \prod_{i=0}^{t-1} \zeta = \zeta^t$ and $\beta_t = \sum_{i=0}^{t-1} \prod_{j=i+1}^{t-1} \left(\mathbf{1}_{j<t} \left(1 - \gamma + \kappa^2 \xi_j\right) + \mathbf{1}_{j=t}\right) \leq \frac{1-\zeta^t}{1-\zeta}$.

Further calculation yeilds $\nu_t = (c_1 + c_2 \beta_t)(x_0 \alpha_t + c_3 \beta_t) \leq \left(c_1 + c_2 \frac{1}{1-\zeta}\right)\left(x_0 + c_3 \frac{1}{1-\zeta}\right)$.

As a result, the sufficient condition (16) for the existence and uniqueness of $\mathbf{M}^{PS}$ is reduced to

$$\left(c_1 + c_2 \frac{1}{1-\zeta}\right)\left(x_0 + c_3 \frac{1}{1-\zeta}\right) \sum_{t=0}^{T-1} \left(\varepsilon_t \left(T-t\right)\right) < \widetilde{\mu}. \tag{132}$$

This is equivalent to $\left(c_1 + c_2 \frac{1}{1-\zeta}\right)\left(x_0 + c_3 \frac{1}{1-\zeta}\right) \sum_{t=0}^{T-1} \frac{T-t}{T-H+1} \varepsilon_t < \overline{\mu}$, where $\overline{\mu} = \min\left\{\frac{\mu \sigma^2}{2}, \frac{\mu \sigma^2 \gamma^2}{64 \kappa^{10}}\right\}$. As a result, to guarantee condition (132) holds, it is sufficient to ensure that

$$\sum_{t=0}^{T-1} \varepsilon_t < \left(1 - \frac{H}{T}\right)\left(c_1 + c_2 \frac{1}{1-\zeta}\right)^{-1}\left(x_0 + c_3 \frac{1}{1-\zeta}\right)^{-1} \overline{\mu}.$$

Proposition 1 is thus proved by choosing $\phi = \left(c_1 + c_2 \frac{1}{1-\zeta}\right)^{-1}\left(x_0 + c_3 \frac{1}{1-\zeta}\right)^{-1}$.

# I  Proof of Proposition 2

Since $\widetilde{\zeta} \leq \|\mathbf{A}_t\| \leq 1 - \gamma + \kappa^2 \xi_t, \forall 0 \leq t < T$, for some a positive constant $\widetilde{\zeta} > 1$, it follows directly that $\alpha_t = \prod_{i=0}^{t-1} \left(1 - \gamma + \kappa^2 \xi_i\right) \geq \alpha_t \geq \prod_{i=0}^{t-1} \widetilde{\zeta} = \widetilde{\zeta}^t$ and $\beta_t = \sum_{i=0}^{t-1} \prod_{j=i+1}^{t-1} \left(\mathbf{1}_{j<t} \left(1 - \gamma + \kappa^2 \xi_j\right) + \mathbf{1}_{j=t}\right) \geq \frac{\widetilde{\zeta}^t - 1}{\widetilde{\zeta} - 1}$.

Further calculation yeilds $\nu_t = (c_1 + c_2 \beta_t)(x_0 \alpha_t + c_3 \beta_t) \geq \left(c_1 + c_2 \widetilde{\zeta}^{t-1}\right)\left(x_0 \widetilde{\zeta}^t + c_3 \widetilde{\zeta}^{t-1}\right)$. Therefore,

$$\sum_{i=t+1}^{T} \nu_i \geq \left(c_1 x_0 + (c_1 c_3 + c_2 c_3 + c_3 x_0) \widetilde{\zeta}^{-1}\right) \sum_{i=t+1}^{T} \widetilde{\zeta}^i$$

$$= \left(c_1 x_0 + (c_1 c_3 + c_2 c_3 + c_3 x_0) \widetilde{\zeta}^{-1}\right) \frac{\widetilde{\zeta}^{T-t} - 1}{\widetilde{\zeta} - 1}. \tag{133}$$

As a result, to guarantee the sufficient condition (16) can be satifised, we must have

$$\sum_{t=0}^{T-1} \frac{\varepsilon_t}{T-H+1}\left(c_1 x_0 + (c_1 c_3 + c_2 c_3 + c_3 x_0) \widetilde{\zeta}^{-1}\right) \frac{\widetilde{\zeta}^{T-t} - 1}{\widetilde{\zeta} - 1} < \overline{\mu}. \tag{134}$$

This means that

$$\varepsilon_t < \frac{(T-H+1)\left(\widetilde{\zeta} - 1\right)}{c_1 x_0 + (c_1 c_3 + c_2 c_3 + c_3 x_0) \widetilde{\zeta}^{-1}} \cdot \frac{\overline{\mu}}{\widetilde{\zeta}^{T-t} - 1}. \tag{135}$$

As a result, Proposition 2 is proved by letting $\phi = \frac{\widetilde{\zeta} - 1}{c_1 x_0 + (c_1 c_3 + c_2 c_3 + c_3 x_0) \widetilde{\zeta}^{-1}}$.

