# OpenReview forum: "Performative Control for Linear Dynamical Systems"
_NeurIPS.cc/2024/Conference — NeurIPS 2024 poster_

### Official Review · Reviewer_e5pZ · 2024-07-11

**Soundness:** 2
**Presentation:** 2
**Contribution:** 1
**Rating:** 3
**Confidence:** 4

**Summary:**

This paper presents the new framework of performative control based on the performantive prediction concept.  The performative stable control (PSC) problem is formulated and its solvability conditions are derived. This paper also presents the algorithm of finding the PSC solution and addresses the convergence analysis.

**Strengths:**

The concept of performative control is nicely motivated in Introduction.  The control policy, design method, and convergence analysis are stated mathematically rigorously.

**Weaknesses:**

The disturbance-action policy defined in Definition 1 implicitly assumes that the disturbance w is measurable.  This assumption is practically severe and limits the application of the control policy.

The results of this paper rely heavily on the performative prediction results from the work [22].  Under many practically severe assumptions, the authors provide a new control policy and convergence analysis.

**Questions:**

The assumption that the disturbance w is available for determining the control action u, is practically severe.  Could the authors relax the assumption?

Lemma 4, the convergence condition, is technically correct.  However, the reviewer cannot find any interpretation of the condition (11).  Could you give some remarks on the interpretation?

The reviewer cannot agree the statement of Introduction, "Most prior studies on control of linear dynamical systems rely on the key assumption that the system state transition model is static".  The pioneering works on state-space-based theory by R. E. Kalman, published 1960, handle time-varying system, meaning the transition is time-dependent.  Could you give more surveys on classical control problems?

**Limitations:**

The limitations of the proposed control policy such as measurable disturbance and perfect modeling on the state transition are not adequately stated in this paper.

---

> ### Author Rebuttal · Authors · 2024-08-06
>
> Thank you for your work in reviewing our manuscript, and for providing thorough feedback. We address your concerns as follows.
>
> **Weaknesses:**
>
> **[W1] "The disturbance-action policy...":** We want to clarify that the reviewer has a misunderstanding regarding the measuring of disturbance $\mathbf{w}\_{t}.$
>
> The measuring of the disturbance $\mathbf{w}\_{t}$ is practically not severe and is commonly advocated in the framework of non-stochastic control [1,8,10,11] initiated by Elad Hazan. The methods for measuring $\mathbf{w}\_{t}$ are mature and have been extensively studied for LTI systems with static $\mathbf{A}$ and $\mathbf{B}$ [1,8,10,11] and LVI systems with time-varying $\mathbf{A}\_{t}$ and $\mathbf{B}\_{t}$ ([Chapter 7, R1]).
>
> The control policy we adopt in our paper is the disturbance-action policy, which is popular and widely used in non-stochastic control applications [1,8,10,11]. There is no application limitation as suggested by the reviewer.
>
> **[W2] "The results of this paper rely heavily...":** We want to clarify that our work does not relies on the results from work [22]. In contrast, compared to [22], our work has totally different convexity assumption and convergence analysis, which are summarized below.
>
> **1. Convexity Assumption**: For the original performative prediction work [22], one can directly assume the objective function $\mathbb{E}[c_{t}(\mathbf{x},\mathbf{u})]$ is $\mu$-strongly convex w.r.t. optimization variable $\mathbf{M}$ (i.e., the disturbance-action policy) **without any problem-specific convexity analysis**.
>
> However, for the convexity in our work, we only have a mild assumption that the per stage cost $c_{t}(\mathbf{x},\mathbf{u})$ is **$\mu$-strongly convex** for the state-action pair $(\mathbf{x},\mathbf{u})$. We provide rigorous theoretical analysis that $\mathbb{E}[c_{t}(\mathbf{x},\mathbf{u})]$ is strongly convex w.r.t. $\mathbf{M}$ and the strongly convex constant is **not $\mu$, but min{$\frac{\mu\sigma^{2}}{2},\frac{\mu\sigma^{2}\gamma^{2}}{64\kappa^{10}}$}** , where $\kappa$ and $\gamma$ are problem-specific parameters related to the dynamical system parameters $\mathbf{A},\mathbf{B}$ and $\mathbf{K}$. **This is totally different from the results in [22]**.
>
> **2. Convergence Analysis**: To guarantee convergence of our proposed gradient descent algorithm, the designing of the step sizes relies on the problem-specific strongly convex constant min {$\frac{\mu\sigma^{2}}{2},\frac{\mu\sigma^{2}\gamma^{2}}{64\kappa^{10}}$} and the distributional sensitivity temporal propagation and aggregation $\mathbf{M}^{PS}$ existence condition. **These are unique to our performative dynamical systems, and have never been exploited in [22]**.
>
> **Questions:**
>
> **Q1. The assumption that the disturbance w ...?**
>
> A1. As we clarified in the Response to [W1], the measuring of the disturbance $\mathbf{w}\_{t}$ is practically not severe at all, and is commonly advocated in the framework of non-stochastic control [1,8,10,11] initiated by Elad Hazan. The control action $\mathbf{u}\_{t}^{(\mathbf{M})}$ in our paper is the disturbance-action policy, which is also popular and widely used in [1,8,10,11].
>
> **Q2. Lemma 4, the convergence condition...?**
>
> A2. **The interpretation of the condition (11) has already been provided in Line 238 -Line 246 right after condition (11)**. Condition (11) reveals the effects of the sensitivity propagation and aggregation, which are unique to our performative dynamical systems, and have never been exploited in the existing performative prediction works.
>
> **Q3. The reviewer cannot agree the statement ... ?**
>
> A3. Our statement is that **"most prior studies"** consider the static transition model. Of course, there are several
> other papers on time-varying transition model, such as the paper suggested
> by the reviewer and the papers on jump linear systems [R2, R3], etc. We have included more survey on time-varying transition models.
>
> However, the key message we want to deliver is that the system state
> transition model can be changed by control policies has not been considered
> in existing literature, either for static or time-varying transition
> models. To the best of our knowledge, our paper is the first work
> to consider such performative state transition model for linear dynamical
> systems.
>
> **Response to Limitation:** We want to clarify that the disturbance-action control policy adopted in our paper is not proposed by us. Instead, it was initiated by Elad Hazan et al. in the framework of non-stochastic control [1,8,10,11]. The methods for measuring $\mathbf{w}\_{t}$ are mature and have been extensively studied in [1,8,10,11]. Particularly, in [R4], the disturbance-action control policy is applied to the case of imperfect modelling on the state transition, where the underlying state transition model is unknown.
>
> [R1]. Hazan E, Singh K. Introduction to online nonstochastic control[J]. arXiv preprint arXiv:2211.09619, 2022.
>
> [R2]. Zhang L, Leng Y, Colaneri P. Stability and stabilization of discrete-time semi-Markov jump linear systems via semi-Markov kernel approach[J]. IEEE Transactions on Automatic Control, 2015, 61(2): 503-508.
>
> [R3]. Cheng J, Xie L, Zhang D, et al. Novel event-triggered protocol to sliding mode control for singular semi-Markov jump systems[J]. Automatica, 2023, 151: 110906.
>
> [R4]. Elad Hazan, Sham M. Kakade, and Karan Singh. The nonstochastic control problem. In Proceedings of the 31st International Conference on Algorithmic Learning Theory (ALT), pages 408--421, 2020.

---

> > ### Author Response · Authors · 2024-08-11
> > **Seeking Your Feedback**
> >
> > Dear Reviewer e5pZ,
> >
> > We appreciate your time and effort in reviewing our submission. We have now provided further clarification, explanation, and discussion to address your concerns. We hope you've had a chance to review them, consider whether our response addresses your concerns, and reevaluate your score accordingly. Please let us know if you have any further questions or need clarification. Thank you very much.
> >
> > Warmest regards,
> >
> > Authors

---

> > ### Comment · Reviewer_e5pZ · 2024-08-13
> >
> > Thanks for the reply comments.
> >
> > [W1] "The disturbance-action policy..."
> > The reviewer understands that the problem setting on the measurable disturbance is commonly used in some works (thanks for raising the literature again).
> >
> > [W2] "The results of this paper rely heavily..."
> > The reviewer finds some novelty of the work compared with the previous work [22].  However, he/she believes the contribution is insufficient, particularly not enough to be accepted at a high-level conference like NeurIPS.

---

> ### Author Response · Authors · 2024-08-12
> **Key differences between our PP on linear dynamical system and performative prediction results from the work [22]**
>
> As suggested by Reviewer wfwc, we here recap the key differences between our work and the existing PP work [22].
>
> **Original PP work [22]**: We first point out that the original PP does not involve temporal correlated performative sample data in the cost $\mathbb{E}[l(\mathbf{\theta};Z)]$, where all the data samples follow a static distribution $Z\sim\mathcal{D}(\theta).$
>
> **PP on on linear dynamical system (our paper)**: In addition to the **linear-system-dynamics-dependent strong convexity analysis**, the linear dynamic system will introduce a unique challenge
> of **temporally correlated performative state sample data $\mathbf{x}\_{t}$
> and $\mathbf{u}\_{t}$ in the cost $\mathbb{E}[c\_{t}(\mathbf{x}\_{t},\mathbf{u}\_{t})]$**, which cannot be covered by the above Original PP work [22]. Specifically,
>
> **$\bullet$** The cost $\mathbb{E}[c\_{t}(\mathbf{x}\_{t},\mathbf{u}\_{t})]$ is a function of policy $\mathbf{M},$ the form of which will depend on linear dynamics (i.e., $\mathbf{A},\mathbf{B}$ and $\mathbf{K}$). As a result, the linear dynamics will directly affect the strongly convex constant of $\mathbb{E}[c\_{t}(\mathbf{x}\_{t},\mathbf{u}\_{t})]$ in $\mathbf{M}$.
>
> **$\bullet$** The primary cause of the temporal correlation of the performative state data sample $(\mathbf{x}\_{t},\mathbf{u}\_{t})$ is also the linear dynamics (i.e., $\mathbf{A},\mathbf{B}$
> and $\mathbf{K}$).

---

> ### Author Response · Authors · 2024-08-13
>
> Dear Reviewer e5pZ,
>
> Thank you for participating in the discussion.
>
> **[W2] "The results of this paper rely heavily..." The reviewer finds some novelty of the work compared with the previous work [22]. However, he/she believes the contribution is insufficient, particularly not enough to be accepted at a high-level conference like NeurIPS.**：
>
> We thank the reviewer's comments. We believe there may be some misunderstandings regarding the contributions of our work.
> We summarize some key contributions that distinguish our work from the standard PP work [22].
>
> **$\bullet$ Strong Convexity Analysis**: The standard PP work [22] brute-forcely assumed that the optimization variable is strongly convex in the policy without any problem-specific theoretical analysis. However, in our paper, **we provide a linear-system-dynamics-dependent strong convexity analysis**, which is unique to our performative linear dynamic systems.
>
> **$\bullet$ Stateful vs Non-stateful**: The distribution map in the standard PP work [22] is non-stateful, where all the data samples follow a simple and static distribution $Z\sim\mathcal{D}(\theta).$ However, in our work, **we consider a more complicated stateful performative maps, where observed data distributions depend on the history of previously deployed polices**.
>
> **$\bullet$ Promote understanding of general PP**:  Our PP via linear dynamical system modeling is closely connected to the linearized general stateful distribution map in the seminal work of stateful PP [R1]. **Our work allows performative transition maps of performative distributions** and **extends** the technical results of the **fixed performative distribution transition map** in the stateful PP work [R1]. As a result, **our work has potential to enhance the understanding of general performative prediction**.
>
> The above three key differences serve as strong evidence that clearly justifies our contributions and distinguishes our work from standard PP work [22].
>
> [R1]. Gavin Brown, Shlomi Hod, and Iden Kalemaj. Performative prediction in a stateful world. In International conference on artificial intelligence and statistics, pages 6045–6061. PMLR, 2022.

---

### Official Review · Reviewer_nkz4 · 2024-07-11

**Soundness:** 3
**Presentation:** 2
**Contribution:** 4
**Rating:** 8
**Confidence:** 4

**Summary:**

The authors introduce the problem of performative linear control whereby one aims to control the evolution of a linear dynamical system whose dynamics are influenced by the choice of control policy. Apart from defining the problem, they define the concept of a performatively stable controller, extending the definition of performative stability introduced by Perdomo et al for the supervised learning setting. Given their model, their main technical results are results showing existence and uniqueness of a performatively stable controller under specific conditions, and a convergence result that illustrates how repeated stochastic gradient descent will converge to stability in this new setting at a O(1/T) rate.

**Strengths:**

The idea of connecting performative prediction and linear control is very exciting, and to the best of my knowledge novel. I really enjoyed this idea that the very choice of control policy might actively influence the underlying system dynamics, and hence we should start to think about new models that explicitly account for this feedback. The technical results seem solid, although I wasn’t able to go through the proofs line by line. I also enjoyed the insights regarding how sensitivities need to be smaller at the beginning of the time horizon versus at the end.

**Weaknesses:**

While the motivation regarding how control policies might influence the dynamics is exciting, I find that it is quite underdeveloped in the paper. The main illustrative example the authors have is this stock market example. The intuition behind how control policies can affect the dynamics is plausible to me, but I would have liked to see it written out in detail in the main body. The primary exposition of this example is deferred to the appendix and I felt that it lacked clarity.

This underdeveloped motivation then makes the formal model itself hard to interpret and justify. I didn’t quite understand why the policy only affects the state matrix A, why all policy induce disturbances that are zero-mean, or why the disturbances between time steps are uncorrelated for the same policy. It would be helpful if the authors spent more time explaining the model and how it maps onto the high level ideas explained in the introduction.

Lastly, I think the paper suffers from a lack of clarity in the technical writing. Several quantities are referenced without being previously defined and there are some claims in the paper than are unsubstantiated (specifically, the idea that performative stability implies performative optimality in this new control model). Please see below for a full list of comments and questions.

**Questions:**

Could the authors motivate why they assume that A_t is performative but B is time invariant?

Why are the performative disturbances for A_t assumed to all be zero mean?

In L171, what is \tilde{A?}? Is this just A? Or A - BK?

In Lemma 1, is K assumed to be strongly stable? If so, could you please make that explicit?

In L182: the authors claim that the PSC is approximately performative optimal, why is that?  In L236 they cite [22], but the result in [22] is shown in a different model with different assumptions

L205: there is a mistaken reference, the authors reference A1-6, but only A4-6 are about the costs

Why do the authors refer to the condition that the operator norms of the matrices are <1 as almost surely stable? I usually understand stability in linear control as the condition that the spectral radius (not the operator norm) is smaller than 1. Assuming that the operator norm is smaller than 1 is a strictly stronger condition.

Theorem 1 is not very readable. Please write this out as a lemma in the appendix and state the result formally with the step sizes plugged in so that the reader can appreciate the final convergence rate (stated formally).

What is reference 13 it doesn’t seem to have an author? Reference 18 was published in a conference.

**Limitations:**

These issues have been discussed to the extent necessary.

---

> ### Author Rebuttal · Authors · 2024-08-06
>
> We are glad that the reviewer liked our paper and recognized our exciting idea.
>
> **Weaknesses:**
>
> **[W1] "While the motivation...":** Thank you for recognizing and appreciating our key idea that control policies affect dynamics. To fully develop this idea, we have moved the main illustrative example of the stock market from the appendix to the main body of the paper. We have elaborated and discussed the intuitions behind how control policies can affect the dynamics in detail in the main body of the paper using this stock market example.
>
> **[W2.1] "Why the policy only affects the state matrix $\mathbf{A}$?":** The key motivation for letting only matrix $\mathbf{A}\_t$ be performative is that our work adopts the linear disturbance-action control policy initiated in the framework of **non-stochastic control** [1,8,10,11] by Elad Hazan et al, where the disturbance-action $\mathbf{B}\mathbf{M}[\mathbf{w}]_{t-1}^{H}$ is linear in policy $\mathbf{M}$. Such kind of linear disturbance-action control policy has various nice theoretical performance guarantees as substantiated in [1,8,10,11].
>
> On the other hand, if $\mathbf{B}$ is also performative, we will have a generalized disturbance-action $\mathbf{B}\_t(\mathbf{M})\mathbf{M}[\mathbf{w}]_{t-1}^{H}$, which can be possibly nonlinear in policy $\mathbf{M}$. To the best of our knowledge, such kind of generalized nonlinear disturbance-action control policy has received very few research attention, and it can serve as a very exciting future research direction.
>
> To the best of our knowledge, this paper is the very first work to investigate the control of performative dynamical systems, and we intend to provide a thorough theoretical investigation and aim at establishing various new theoretical results. Therefore, we choose to build our work on the linear disturbance-action control policy (i.e., policy only affects the state matrix $\mathbf{A}$ and not affects matrix $\mathbf{B}$), whose nice theoretical performance guarantees would facilitate and consolidate our theoretical analysis.
>
> **[W2.2] "Why all policy induce disturbances that are zero-mean?":** The non-zero mean and zero mean cases are mathematically equivalent. The mean value of the disturbance can be absorbed into the original state transition matrix $\mathbf{A}$, which is equivalent to a new $\mathbf{A}^{\prime}$ ($\mathbf{A}^{\prime}$ equals $\mathbf{A}$ plus the mean of disturbance) with zero-mean disturbances. We only need to choose a new controller $\mathbf{K}^{\prime}$ such that $\mathbf{A}^{\prime}-\mathbf{B}\mathbf{K}^{\prime}$ is $(\kappa,\gamma)$-strongly stabilizing as specified in Definition 2. In this way, the non-zero mean performative disturbances are converted equivalently to the zero-mean case.
>
>
> **[W2.3] "Why the disturbances between time steps are uncorrelated for the same policy?":** From the application perspective, we have used the widely adopted uncorrelated truncated lognormal distributions of the stock price model [23,27] in the stock market example to justify the temporal uncorrelated disturbances for the same policy. From the theoretical perspective, the performative disturbances are the data samples, which are assumed to be i.i.d. distributed according to a certain distribution map under a same policy in the existing framework of performative prediction [3,5,12,14,15,20,22,25]. In this work, we followed this convention.
>
> **[W3] "Lastly, I think the paper suffers from...":** We have gone through the paper thoroughly to make sure all the quantities are well defined. We have not claimed that "performative stability implies performative optimality". In fact, the performative stable solution $\mathbf{M}^{PS}$ and performative optimal solution $\mathbf{M}^{PO}$ converge to a same value $\mathbf{M}^{*}$ as the magnitudes of all the sensitivities {$\{\varepsilon_{t},0\leq t<T\}$} approach 0.
>
> **Questions:**
>
> **Q1. Could the authors motivate why they assume that A_t is performative but B is time invariant?**
>
> A1. The motivations have been clarified in Response to **[W2.1]**.
>
> **Q2. Why are the performative disturbances for A_t assumed to all be zero mean?**
>
> A2. The justifications have been clarified in Response to **[W2.2]**.
>
> **Q3. In L171, what is \tilde{A}? Is this just A? Or A-BK?**
>
> A3. In L171, $\mathbf{\widetilde{A}}$ is $\mathbf{A}-\mathbf{B}\mathbf{K}$,which has been defined in Definition 2 in L144.
>
> **Q4. In Lemma 1, is K assumed to be strongly stable? If so, could you please make that explicit?**
>
> A4. In Lemma 1, $\mathbf{K}$ is a strongly stabilizing linear controller.
>
> **Q5. In L182: the authors claim...**
>
> A5. The justifications have been clarified in Response to **[W3]**.
>
> **Q6. L205: there is a mistaken...**
>
> A6. We have changed the reference citation from "A1-6" to "A 4-6" in L205.
>
> **Q7. Why do the authors refer..**
>
> A7. To avoid potential confusion, we have rephrased the wording "almost surely stable" in Proposition 1 to "strongly stable",
> which has been defined in Definition 2. The "strongly stable" condition is indeed stronger and places requirements on the operator norm of matrices. Specifically, we let $\mathbf{A}\_{t}^{\pi}$ be $\left(\kappa,\gamma-\kappa^{2}\xi_{t}\right)$-strongly stable for real numbers $\kappa\geq1,\gamma-\kappa^{2}\xi_{t}<1$, $\forall0\leq t<T.$ The technical results in Proposition 1 then follow directly by letting $\zeta$=max{$ 1-\gamma+\kappa^{2} \xi\_{t},\forall 0\leq t<T$}.
>
> **Q8. Theorem 1 is not very readable...**
>
> A8.  Theorem 1 has a complex form because of its strong capability to accommodate general step size rules. Nevertheless, we respect and follow the reviewer's suggestion to plug in the specific diminishing step size and stated formally the associated final convergence rate. The technical results on the general step size have been moved into
> appendix.
>
> **Q9. What is reference 13 it doesn’t seem to have an author? Reference 18 was published in a conference.**
>
> A9. We have revised reference 13 and 18.

---

> > ### Author Response · Authors · 2024-08-11
> > **Seeking Your Feedback**
> >
> > Dear Reviewer nkz4,
> >
> > We appreciate your time and effort in reviewing our submission. We have now provided further clarification, explanation, and discussion to address your concerns. We hope you've had a chance to review them, consider whether our response addresses your concerns, and reevaluate your score accordingly. Please let us know if you have any further questions or need clarification. Thank you very much.
> >
> > Warmest regards,
> >
> > Authors

---

> > > ### Comment · Reviewer_nkz4 · 2024-08-13
> > >
> > > Hi,
> > >
> > > Sorry for the delay. Thank you for taking the time to answer my questions. Very helpful.
> > >
> > > I appreciate how making the B matrix performative complicates the analysis since it not as straightforward to use tools like the disturbance action control policies. It would be good for the authors to clarify in the manuscript that this assumption comes out of a technical necessity rather than an explicit modeling decision.
> > >
> > > "Why all policy induce disturbances that are zero-mean?": I see how this might be the case if the disturbances have non-zero mean, but this mean is constant throughout all time steps. Is this still the case if the mean of the disturbances varies by time step? (the system is time invariant here no?)
> > >
> > >
> > >
> > > W3. In L182, you claim that stable controllers "approximately solves the POC (6)". I read this as a claim that stability implies optimality which as far as I can see, is not shown in this model.
> > >
> > > I like this paper because I think it takes a step in the right direction in establishing technical transfers of knowledge between performative prediction and other fields which study with learning dynamical systems. On this note, one of the core problems in performative prediction is how to confront the fact that the observed data distributions $D(\theta_t)$ is just not just a function of $\theta_t$ but are stateful and depend on the the history of previously deployed models.
> > >
> > > I think this paper would be incredibly impactful if it could show and argue why the linear dynamical system framing can connect with learning under stateful distribution maps. More concretely, the most general setting for stateful performative maps, is to let
> > >
> > > $D_{t} = f(D_{t=1}, \theta_t)$ where $D_t$ is the distribution over states (data) at time $t$. Do the authors have a way of showing that by appropriately considering some Jacobian linearization of this map f, one could arrive at their linear dynamical system model?
> > >
> > > Illustrating some connection (even if not completely precise and exact) between LDS's and stateful performative prediction (as initiated by Brown et al AISTATS 2022) would really go a long way in terms of motivating the author's contribution.
> > >
> > > Sorry for asking this so late in the window, but any insight into this last question would be very helpful in my review.

---

> ### Author Response · Authors · 2024-08-13
>
> Thank you very much for your precious time to participate in the discussion. And thanks a lot for the many inspiring comments with deep insights.
>
> **1. "I appreciate how making the B matrix performative complicates..rather than an explicit modeling decision.":** We will follow this comment to clarify the assumption on $\mathbf{B}$ in the revised manuscript.
>
> **2. "Why all policy induce disturbances that are zero-mean?... Is this still the case if the mean of the disturbances
> varies by time step? (the system is time invariant here no?)":** Yes, this is still the case if the mean of the disturbances varies
> by time step. Specifically, suppose the mean of the disturbance is $\mathbb{E}[\mathbf{\Delta}\_{t}]=\Theta_{t},$ which is
> time-varying. Same as the constant mean case, this is equivalent a new $\mathbf{A}\_{t}^{\prime}=\mathbf{A}+\Theta\_{t}$ with zero mean disturbances. We only need to choose a new linear controller $\mathbf{K}\_{t}^{\prime}$ such that $\mathbf{A}\_{t}^{\prime}-\mathbf{B}\mathbf{K}\_{t}^{\prime}$ is $\left(\kappa\_{t},\gamma\_{t}\right)$-strongly stabilizing. We
> then let $\kappa=$max{$\kappa\_{t},0\leq t<T$} and $\gamma=$max{$\gamma\_{t},0\leq t<T$}, and all of our technical results still hold.
>
> **3. "W3. In L182, you claim that stable controllers 'approximately solves the POC (6)'. I read this as a claim that stability
> implies optimality which as far as I can see, is not shown in this model.":** To avoid any potential confusion, we will revise the phrase "approximately solves the POC" to a more specific statement that "the performative stable solution $\mathbf{M}^{PS}$ and performative optimal solution $\mathbf{M}^{PO}$ both converge to a common value $\mathbf{M}^{*}$ as the magnitudes of the sensitivities {$\varepsilon\_{t},0\leq t<T$} approach 0."in the revised manuscript.
>
> **4. "I like this paper because I think it takes a step in the
> right direction in establishing technical transfers of knowledge between
> performative prediction and other fields which study with learning
> dynamical systems. On this note, one of the core problems in performative
> prediction is how to confront the fact that the observed data distributions
> $D\left(\theta_{t}\right)$ is just not just a function of $\theta\_{t}$
> but are stateful and depend on the the history of previously deployed
> models.":** It is our great pleasure that the reviewer liked our paper. We firmly
> agree with the reviewer's point of view that one of the core problems
> in PP is how to deal with the stateful data distributions. The stateful
> distributional map setup in our paper also distinguishes our work
> from the existing PP works, including the original PP work [22].

---

> > ### Comment · Reviewer_nkz4 · 2024-08-13
> >
> > With regards to point 4, could you provide a more detailed answer illustrating a precise connection between stateful performative prediction models and the linear dynamical system setting you consider?

---

> > > ### Author Response · Authors · 2024-08-13
> > >
> > > Dear Reviewer nkz4,
> > >
> > > Due to writing space constraints, we're now typing the answers for the questions
> > >
> > > **5. I think this paper would be incredibly impactful if it
> > > could show and argue why the linear dynamical system framing can connect
> > > with learning under stateful distribution maps. More concretely, the
> > > most general setting for stateful performative maps, is to let $D_{t}=f\left(D_{t=1},\theta_{t}\right),$
> > > where $D_{t}$ is the distribution over states (data) at time . Do
> > > the authors have a way of showing that by appropriately considering
> > > some Jacobin linearization of this map f, one could arrive at their
> > > linear dynamical system model?**
> > >
> > > **6. Illustrating some connection (even if not completely precise
> > > and exact) between LDS's and stateful performative prediction (as
> > > initiated by Brown et al AISTATS 2022) would really go a long way
> > > in terms of motivating the author's contribution.**
> > >
> > > in the next official comment. Could you please wait several minutes? Thank you so much.
> > >
> > > Warmest regards,
> > > Authors.

---

> > > > ### Author Response · Authors · 2024-08-13
> > > >
> > > > **5. "I think this paper would be incredibly impactful if it
> > > > could show and argue why the linear dynamical system framing can connect
> > > > with learning under stateful distribution maps. More concretely, the
> > > > most general setting for stateful performative maps, is to let $D_{t}=f\left(D_{t=1},\theta_{t}\right),$
> > > > where $D_{t}$ is the distribution over states (data) at time . Do
> > > > the authors have a way of showing that by appropriately considering
> > > > some Jacobin linearization of this map f, one could arrive at their
> > > > linear dynamical system model?":**
> > > >
> > > > Great and inspiring comments! Let us consider a general setting for
> > > > stateful distribution maps $D\_{t}=f\left(D\_{t-1},\theta\_{t}\right),\forall t\geq1.$
> > > > To facilitate the analysis, we use a random variable $d\_{t}\sim D\_{t}$
> > > > to equivalently characterize the distribution $D\_{t}.$ If the properties
> > > > of $f\left(\cdot\right)$ are "nice“ enough, then from the perspective
> > > > of random variables, there will exist a corresponding mapping $\widetilde{f}\left(\cdot\right)$
> > > > such that
> > > > \begin{align*}
> > > >  & d_{t}\overset{\mathrm{d}}{=}\widetilde{f}\left(d_{t-1},\theta_{t}\right),d_{t}\sim D_{t},d_{t-1}\sim D_{t-1},\forall t\geq1,
> > > > \end{align*}
> > > > where $\overset{\mathrm{d}}{=}$ denotes equal in distribution. Linearizing
> > > > $\widetilde{f}\left(\cdot\right)$ at some equilibrium point $\left(\overline{d},\overline{\theta}\right)$
> > > > with $\overline{d}\sim\overline{D}$ and ignoring the higher order
> > > > terms will lead to
> > > > \begin{align}
> > > > d_{t} & \overset{\mathrm{d}}{=}\widetilde{f}\left(\overline{d},\overline{\theta}\right)+\left[\frac{\partial\widetilde{f}}{\partial d}\right]\_{\left(\overline{d},\overline{\theta}\right)}\left(d\_{t-1}-\overline{d}\right)+\left[\frac{\partial\widetilde{f}}{\partial\theta}\right]\_{\left(\overline{d},\overline{\theta}\right)}\left(\theta_{t}-\overline{\theta}\right)\\\\
> > > >  & \overset{\mathrm{d}}{=}\mathbf{A}d\_{t-1}+\mathbf{B}\theta\_{t}+\mathbf{w}, \quad (1)
> > > > \end{align}
> > > > where $\mathbf{A}=\left[\frac{\partial\widetilde{f}}{\partial d}\right]\_{\left(\overline{d},\overline{\theta}\right)}$
> > > > and $\mathbf{B}=\left[\frac{\partial\widetilde{f}}{\partial\theta}\right]\_{\left(\overline{d},\overline{\theta}\right)}$
> > > > are the Jacobin matrices at the equilibrium point $\left(\overline{d},\overline{\theta}\right)$
> > > > and $\mathbf{w}=\widetilde{f}\left(\overline{d},\overline{\theta}\right)-\mathbf{A}\overline{d}-\mathbf{B}\overline{\theta}.$
> > > >
> > > > We can view $d\_{t}$ in (1) as the random state
> > > > variable $\mathbf{x}\_{t}$ of the linear dynamical systems in our
> > > > paper, $\theta\_{t}$ as the input policy (i.e., $\mathbf{M}\_{t}$
> > > > in our case) and $\mathbf{w}$ as the additive system noise (i.e.,
> > > > $\mathbf{w}\_{t}\sim\mathbf{w}$ in our case). In this way, as suggested
> > > > by the reviewer, after Jacobin linearization of $\widetilde{f}\left(\cdot\right)$
> > > > (corresponding to $f\left(\cdot\right)$ from the perspective of distribution),
> > > > one can arrive at our linear dynamical model with $\mathbf{A}=\left[\frac{\partial\widetilde{f}}{\partial d}\right]\_{\left(\overline{d},\overline{\theta}\right)}$
> > > > and $\mathbf{B}=\left[\frac{\partial\widetilde{f}}{\partial\theta}\right]\_{\left(\overline{d},\overline{\theta}\right)}.$
> > > > In the most general case, $\mathbf{A},\mathbf{B}$ are random matrices,
> > > > and $\mathbf{A},\mathbf{B}$ will be static if the equilibrium point
> > > > $\left(\overline{d},\overline{\theta}\right)$ is chosen as constant.

---

> ### Author Response · Authors · 2024-08-13
>
> **6. "Illustrating some connection (even if not completely precise
> and exact) between LDS's and stateful performative prediction (as
> initiated by Brown et al AISTATS 2022) would really go a long way
> in terms of motivating the author's contribution":**
>
> Thank you very much for your suggestion. We illustrate some connections
> between our LDS and the seminal stateful PP work by Brown et al AISTATS
> 2022 [3] as follows.
>
> **$\bullet$ Performative Prediction in a Stateful World by Brown et al AISTATS
> 2022 [3]**: This seminal work introduces a transition map $f\left(\cdot\right)$
> to model the stateful distribution as
> \begin{align}
>  & D\_{t}=f\left(D\_{t-1},\theta\_{t}\right),\forall t\geq1. \quad  (2)
> \end{align}
> Similar as our response to Comment 5, after linearization and ignoring
> the higher order terms, equation (2) can be represented as
> \begin{align}
> d\_{t} & \overset{\mathrm{d}}{=}\mathbf{A}d\_{t-1}+\mathbf{B}\theta\_{t}+\mathbf{w}, \quad (3)
> \end{align}
> with $\mathbf{A}=\left[\frac{\partial\widetilde{f}}{\partial d}\right]\_{\left(\overline{d},\overline{\theta}\right)},$
> $\mathbf{B}=\left[\frac{\partial\widetilde{f}}{\partial\theta}\right]\_{\left(\overline{d},\overline{\theta}\right)}$
> and $\mathbf{w}=\widetilde{f}\left(\overline{d},\overline{\theta}\right)-\mathbf{A}\overline{d}-\mathbf{B}\overline{\theta}.$
>
> **$\bullet$ Our LDS**: View the state $\mathbf{x}\_{t}$ of LDS as a random variable,
> we have
> \begin{align}
>  & \mathbf{x}\_{t}\overset{\mathrm{d}}{=}\mathbf{A}\_{t}\left(\mathbf{M}\right)\mathbf{x}\_{t-1}+\mathbf{B}\mathbf{u}^{DAP}\left(\mathbf{M}\_{t}\right)+\mathbf{w}\_{t},\quad (4)
> \end{align}
> where
> \begin{align}
>  & \mathbf{A}\_{t}\left(\mathbf{M}\right)=\mathbf{A}-\mathbf{B}\mathbf{K}+\mathbf{\Delta}\_{t},\mathbf{\Delta}\_{t}\sim\mathcal{D}\_{t}\left(\mathbf{M}\_{t}\right).\quad (5)
> \end{align}
>
> **$\bullet$ Connections between our LDS and the stateful PP by Brown et al AISTATS 2022 [3]**:
>
> **$\quad 1. $  If $\mathbf{\Delta}\_{t}=\mathbf{0}$**, i.e., there is no performative
> disturbance to LDS state transition matrix $\mathbf{A}$: The performative
> distribution of $\mathbf{x}\_{t}$ will depends on $\mathbf{M}\_{t}$,
> but the randomness comes solely from the system noise $\mathbf{w}\_{t}$.
> In this case, if we let $\mathbf{A}$ and $\mathbf{B}$ in our LDS
> be random matrices and satisfy
> \begin{align*}
>  & \mathbf{A}-\mathbf{B}\mathbf{K}=\left[\frac{\partial\widetilde{f}}{\partial d}\right]\_{\left(\overline{d},\overline{\theta}\right)},\mathbf{B}=\left[\frac{\partial\widetilde{f}}{\partial\theta}\right]\_{\left(\overline{d},\overline{\theta}\right)},
> \end{align*}
> our LDS model in (4) matches the linearized stateful distribution map model in Brown et al AISTATS 2022 [3], i.e., Equation
> (3).
>
> **$\quad 2. $ If $\mathbf{\Delta}\_{t}\neq\mathbf{0}$**: Our LDS model in (4) will correspond to a linearized stateful distribution map with performative
> functional $\widetilde{f}\_{\mathbf{M}\_{t}}$ (rather than a fixed
> $\widetilde{f}$ or a fixed transition map $f\left(\cdot\right)$
> in Brown et al AISTATS 2022 [3]), where
> \begin{align*}
>  & \mathbf{A}\_{t}\left(\mathbf{M}\_{t}\right)=\left[\frac{\partial\widetilde{f}\_{\mathbf{M}\_{t}}}{\partial d}\right]\_{\left(\overline{d},\overline{\theta}\right)}.
> \end{align*}
> This means that our LDS model can also correspond to the linearized
> case of
> \begin{align*}
>  & D_{t}=f_{\theta_{t}}\left(D_{t-1},\theta_{t}\right).
> \end{align*}
> This generalizes the static transition map $f\left(\cdot\right)$
> in Brown et al AISTATS 2022 [3] to the more general case of performative
> transition map, i.e., the form of transition map  $f\_{\theta\_{t}}$ will depend on the deployed policy $\theta\_{t}.$
>
> **7. “Sorry for asking this so late in the window, but any insight into this last question would be very helpful in my review.”**
>
> Thank you so much for the great and insightful question! We have provided brief discussions on: (1) connection between general stateful distribution maps $D\_{t}=f\left(D\_{t-1},\theta\_{t}\right)$ and our linear dynamic
> model; and (2) the connection between LDS and the seminal work by
> Brown et al AISTATS 2022 [3]. We hope that these discussions will be satisfactory and address the reviewer's concerns.

---

> ### Comment · Reviewer_nkz4 · 2024-08-13
>
> Amazing. This I think would be a much much better motivation for the linear dynamical system model than the stock market example. I would encourage the authors to make this connection between stateful distribution maps and linear dynamical systems front and center of their paper. I have updated my score. Please pay attention to all my comments in the revision and take special care to improving the clarity of the technical writing.

---

> ### Author Response · Authors · 2024-08-14
>
> Dear Reviewer nkz4,
>
> We are very glad that you liked our paper. Thank you for the many valuable and insightful comments and discussions in the rebuttal period, which help us to motivate our contributions much better. Many thanks for raising the score.
>
> In the revision, we will pay special attention to all your comments and do our best to improve the clarity of the technical writing.
>
> Warmest regards,
>
> Authors.

---

### Official Review · Reviewer_JGbs · 2024-07-13

**Soundness:** 3
**Presentation:** 3
**Contribution:** 2
**Rating:** 7
**Confidence:** 4

**Summary:**

The work presents a new approach for linear dynamical systems and highlights how control policies can directly affect the system’s dynamics thereby resulting in policy-dependent changes in system states. It also outlines specific conditions under which a stable control strategy can be achieved and introduces a method based on repeated stochastic gradient descent to reach this solution. The authors support their theoretical claims with relevant empirical results.

**Strengths:**

- Work addresses gap in traditional control theory via introduction of performative control
- Provides theoretical foundation with sufficient conditions for performative stable control solutions
- Paper provides application of the framework via numerical examples
- Provides insights into system stability under different conditions
- Development of custom stochastic gradient descent (rsgd) tailored to this new pscs control framework

**Weaknesses:**

- The concepts and algorithms introduced may be complex for practical implementation without extensive customization
- Reliance on numerical examples vs extensive empirical data might limit how the framework will perform on real-world conditions

**Questions:**

- How is policy-depended temporal correlation in this work different from existing performative prediction works? this needs to be covered to highlight novelty
- As per A4, A5 and A6, the cost function is assumed to be strongly convex, smooth and have bounded gradients respectively. These are common assumptions in optimization, can we relax this for cost functions in this control setting?
- The RSGD Algorithm 1 has a projection step Proj_M - what metric is used for this projection? How is $M_n$ updated if the calculated M_n+1 falls outside of M? Also, is it reasonable to expect access to the true $A_t$, $w_t$ matrices in practice to compute the gradient eqn (17)?
- The projection step assumes projecting into feasible set M is computionally tractable. How is it justified based on M structure?
- RSGD is relatively straightforward, why it over other potential optimization methods for finding the psc solution?
- the RSGD convergence rate is analyzed but the work will benefit from an emperical results comparing the actual convergence speed & efficiency of rsgd over other candidate algo.
- Algo 1 assumes access to true gradients of the cost function, in practice they need to be estimated. So, impact of noisy gradient estimates on the convergance properties should be discussed

**Limitations:**

- Assumes the per stage cost function is strongly convex, smooth, and bounded. These assumptions may not hold for all types of control costs, limiting the applicability of the results.
- The implementation of the proposed repeated stochastic gradient descent and other algorithmic strategies may involve high computational costs, particularly as system dimensions and data volumes grow.
- The validation of the framework heavily relies on numerical simulations rather than empirical testing with real-world data.

---

> ### Author Rebuttal · Authors · 2024-08-07
>
> Thanks for your insightful comments and kind suggestions!
>
> **Weaknesses:**
>
> **[W1] "The concepts and algorithms introduced...":** Our primary focus is the thorough theoretical investigation
> of performative dynamical systems. The low complexity customized implementations
> can be exciting future research topics.
>
> **[W2] "Reliance on numerical examples...":** We fully agree with the reviewer's suggestion. In the revised paper,
> we will try to verify our developed theory using the stock market
> example in the appendix based on the real-world data of stock prices.
>
> **Questions:**
>
> **Q1. How is policy-depended temporal correlation...this needs to be covered to highlight novelty.**
>
> A1. In existing performative prediction works [3,5,12,14,15,20,22,25], the data input
> to the learning algorithms are independently generated based on the
> decision-dependent distributions. No temporal correlation is considered
> or exploited between two successive sets of input data. However, our
> work considers a linear dynamical systems, where the policy-dependent
> temporal correlation is introduced via the performative state transition
> matrices.
>
> **Q2. As per A4, A5 and A6, the cost function is assumed to be strongly convex... can we relax this for cost functions in this control setting?**
>
> A2. If we intend to find the performative stable control solution, we
> cannot relax these assumptions for cost functions. However, if we
> only need to find a stationary control policy, we can relax these
> assumptions, such as allowing the per stage cost to be non-convex,etc.
>
> **Q3. The RSGD Algorithm 1 has a projection step $\text{Proj}\_M$ - what metric is used for this projection? How is
> updated if the calculated $\mathbf{M}\_{n+1}$ falls outside of $\mathbf{M}$? Also, is it reasonable to expect access to the true matrices in practice to compute the gradient eqn (17)?**
>
> A3. The metric of the projection in Algorithm 1 is the matrix Frobenius
> norm. If $\mathbf{M}\_{n+1}$ falls outside of $\mathcal{\mathbb{M}},$
> we update the projection as $\mathrm{Proj}\_{\mathcal{\mathbb{M}}}${ $\mathbf{M}\_{n+1}$}=$\arg\min\_{\mathbf{M}\in\mathcal{\mathbb{M}}}||\mathbf{M}\_{n+1}-\mathbf{M}||\_{F}^{2}$.
> For the computation of the gradient in (17), we can first obtain $\mathbf{A}\_{t}=\mathbf{A}+\mathbf{\Delta}\_{t}$
> based on the known disturbance sample $\mathbf{\Delta}\_{t}$ associated
> with $\mathbf{M}\_{n}$. We next follow the conventions in non-stochastic
> control [1,8,10,11] to compute noise as $\mathbf{w}\_{t}=\mathbf{x}\_{t+1}-\mathbf{A}\_{t}\mathbf{x}\_{t}-\mathbf{B}\mathbf{u}\_{t}$
> based on $\mathbf{A}\_{t},$ current state $\mathbf{x}\_{t}$ and control
> input $\mathbf{u}\_{t}$.
>
> **Q4. The projection step assumes projecting into feasible set M is computationally tractable. How is it justified based on $\mathbf{M}$ structure?**
>
> A4. We have assume that the feasible set $\mathcal{\mathbb{M}}$ is a
> bounded convex set. Therefore, the projection is computationally tractable
> as a Frobenius norm square minimization problem with a convex constraint.
>
> **Q5. RSGD is relatively straightforward, why it over other potential optimization methods for finding the psc solution?**
>
> A5. We use the RSGD because it is a simple and commonly used algorithm
> in performative prediction works [3,5,12,14,15,20,22,25]. We agree with the reviewer
> that there may be other potential optimization methods such as the
> low complexity zeroth-order stochastic projected gradient descent [R1,R2].
>
> **Q6. The RSGD convergence rate is analyzed but the work will benefit from an emperical results comparing the actual convergence speed \& efficiency of rsgd over other candidate algo.**
>
> A6. We fully agree with the reviewer's suggestion. We will compare the
> actual convergence speed and efficiency of the proposed RSGD over
> other candidate algorithms, e.g., the low complexity zeroth-order
> optimization methods [R1,R2].
>
> **Q7. Algo 1 assumes access to true gradients of the cost function, in practice they need to be estimated. So, impact of noisy gradient estimates on the convergence properties should be discussed.**
>
> A7. In fact, Algorithm 1 uses the stochastic gradients of the cost function.
> The impact of the variance of the stochastic gradients has been characterized
> in the convergence speed in Theorem 1. Specifically, the second term
> in R.H.S. of (12) depends on the variance of the stochastic gradient
> $\nabla J_{T}(\mathbf{M};\mathbf{M})$ $\mathcal{O}(T\sum_{t=1}^{T}\vartheta_{t}^{2}),$
> which decays at the rate as $\mathcal{O}(\eta_{N-1})$.
>
> **Limitation:**
>
> **"Assumes the per stage cost function is strongly convex...":** We only assume the per stage cost function is strongly convex and smooth w.r.t. the state-action pair $(\mathbf{x},\mathbf{u})$. However, the justification of the strongly convex and smooth w.r.t. our optimization variable $\mathbf{M}$ (i.e., the disturbance-action policy) needs further problem-specific theoretical derivations. Besides, if we only need to find a stationary control policy, we can relax these assumptions, such as allowing the per stage cost to be non-convex, etc.
>
> **"The implementation of the proposed repeated stochastic gradient...":** This paper's primary focus is the theoretical investigation of performative dynamical systems. The development of low complexity implementation
> algorithms, e.g., the zeroth-order methods [R1,R2], can serve as interesting future research topics.
>
> **"The validation of the framework heavily ...":** We will try to verify our developed theory using the stock market
> example in the appendix based on the real-world data of stock prices
> in the revised paper.
>
> [R1]. Golovin D, Karro J, Kochanski G, et al. Gradientless descent: High-dimensional zeroth-order optimization[J]. arXiv preprint arXiv:1911.06317, 2019.
>
> [R2]. Liu S, Li X, Chen P Y, et al. Zeroth-order stochastic projected gradient descent for nonconvex optimization[C]//2018 IEEE Global Conference on Signal and Information Processing (GlobalSIP). IEEE, 2018: 1179-1183.

---

> > ### Comment · Reviewer_JGbs · 2024-08-14
> >
> > Thank you for answering all my comments. The responses on Q5 and Q6 align with my initial comments and addresses that. I went through all the answers and I have no further questions at this point. I believe the weakness highlighted in W1, W2 and empirical validation still remains critical for this work. I hope the authors can leverage context in the rebuttal answers and verify the developed theory on more examples.

---

> > > ### Author Response · Authors · 2024-08-14
> > >
> > > Dear Reviewer JGbs,
> > >
> > > Thank you very much for your valuable comments and suggestions in the rebuttal period.
> > >
> > > We will follow your comments to validate our developed theory on more examples using real-world data.
> > >
> > > Warmest regards,
> > >
> > > Authors

---

### Official Review · Reviewer_wfwc · 2024-07-15

**Soundness:** 3
**Presentation:** 3
**Contribution:** 2
**Rating:** 4
**Confidence:** 2

**Summary:**

The paper introduces the framework of performative control in the context of linear dynamical systems, where the control policy chosen by the controller impacts the underlying system dynamics. This interaction results in policy-dependent system states with temporal correlations. Inspired by the concept of performative prediction, the authors define a performatively stable control (PSC) solution and provide conditions for its existence and uniqueness. Specifically, they analyze how system stability affects the existence of the PSC solution, showing that for almost surely stable dynamics, the PSC solution exists if the sum of distributional sensitivities is sufficiently small. Conversely, for almost surely unstable dynamics, the existence of the PSC solution requires a backward decay of these sensitivities. The paper also introduces a repeated stochastic gradient descent (RSGD) algorithm that converges to the PSC solution, with theoretical analysis of its non-asymptotic convergence rate. Numerical experiments validate the theoretical findings, demonstrating the practical applicability of the proposed framework in scenarios such as stock investment risk minimization.

**Strengths:**

- The paper offers a thorough theoretical examination of the conditions required for the existence and uniqueness of performatively stable control solutions in the linear dynamical system.

**Weaknesses:**

I feel I does not get the punchline of the paper. Many of the convergence results appear standard: if your problem is convex and sufficiently smooth ($\epsilon$-sensitivity), gradient descent is the obvious approach. This observation is well exploited since the original performative prediction paper.

Moreover, despite the existing MDP literature typically considering finite action and state spaces, it is not clear to me that the linear model presents a serious challenge.

Finally, I would like to add I believe the model presented in the paper has potential to enhance our understanding of general performative prediction. Albeit this is not the primary intention of the paper.
In the most abstract sense, the distribution map $\mathcal{D}$ depends on the controller's action $u$ (or policy $M$). I wonder if we can interpret this paper's model as having a distribution map that includes a linear term $Bu_t$ along with a small perturbation $A_t(M)x_t$. In this context, we might say this paper generalizes the original performative prediction framework, which requires the distribution map $\mathcal{D}$ to be insensitive to the controller's actions, by allowing for any linear shift represented by $B$.

**Questions:**

Can you highlight the challenge the is unique to linear dynamical system compared to MDP with finite numbers of states and actions?

**Limitations:**

There is not particular limitations for the paper to address.

---

> ### Author Rebuttal · Authors · 2024-08-07
>
> Thank you for the comments which inspired us to explore our contributions more.
>
> **Weaknesses:**
>
> **[W1] "I feel I does not get the punchline...":** We want to clarify that the reviewer has a significant misunderstanding of our work compared to the original performative prediction (PP).
>
> **1. Convexity**: **We only assume the per stage cost $c_{t}(\mathbf{x},\mathbf{u})$ is $\mu$-strongly convex for the state-action pair $(\mathbf{x},\mathbf{u})$ in Assumption A 4. This does not by itself imply that expected per stage cost $\mathbb{E}[c_{t}(\mathbf{x},\mathbf{u})]$ is strongly convex w.r.t. our optimization variable $\mathbf{M}$** (i.e., the disturbance-action policy). We have highlighted this key point after Lemma 2.
>
> Even if $c_{t}(\mathbf{x},\mathbf{u})$ is $\mu$-strongly convex w.r.t. $(\mathbf{x},\mathbf{u})$, $\mathbb{E}[c_{t}(\mathbf{x},\mathbf{u})]$ can be non-strongly convex w.r.t. $\mathbf{M}$ when $t<H,$ where $H$ is the time horizon of $\mathbf{M}$ defined in Definition 1.
>
> Even if in the cases that $\mathbb{E}[c_{t}(\mathbf{x},\mathbf{u})]$ is indeed strongly convex w.r.t. $\mathbf{M}$, the strongly convex constant is **not $\mu$, but min{$\frac{\mu\sigma^{2}}{2},\frac{\mu\sigma^{2}\gamma^{2}}{64\kappa^{10}}$}** as we have established in Lemma B.2, where $\kappa$ and $\gamma$ are related to the dynamical system parameters $\mathbf{A},\mathbf{B}$ and $\mathbf{K}$. **This is totally different from the original PP paper, where one can directly assume $\mathbb{E}[c_{t}(\mathbf{x},\mathbf{u})]$ is $\mu$-strongly convex w.r.t. $\mathbf{M}$ without any problem-specific theoretical analysis.**
>
>
> **2. Gradient Descent**: To guarantee convergence of our gradient descent algorithm, the designing of the step sizes relies heavily on the problem-specific strongly convex constant min{$\frac{\mu\sigma^{2}}{2},\frac{\mu\sigma^{2}\gamma^{2}}{64\kappa^{10}}$} and the distributional sensitivity propagation and aggregation $\mathbf{M}^{PS}$ existence condition. These are unique to our performative dynamical systems, and have never been exploited in the original PP paper.
>
> **[W2] "Moreover, despite the existing MDP...":** A unique and serious challenge posed by the linear dynamic model is the existence of the optimal solution to the MDP. In particular, for MDPs with finite state space, it is known that optimal strategies exist for a wide range of costs. In contrast, for dynamical systems with infinite state space, optimal solution may not exist [Line 2-5, Abstract, R1]. This unique challenge has been discussed in detail in the Response to **Question** below.
>
> It is worth noting that there are methods for embedding a linear dynamical model into an MDP with finite action and state spaces, e.g. [R2], where the state transition matrix will have certain restrictions to keep the states in the finite set. However, the linear model-based MDPs with infinite action and state spaces will have more general state transition matrices.
>
> **[W3] "Finally, ... by allowing for any linear shift represented by $\mathbf{B}$":** Great and inspiring comments! We completely agree with the reviewer's opinion that our work has the potential to improve the community's understanding of general performative prediction. Inspired by the reviewer, our work's generalizations to the original PP framework include:
>
> • Distribution map with small perturbations. When the perturbation $\mathbf{A}\_{t}(\mathbf{M})\mathbf{x}\_{t}$ is small, a linear shift of the policy represented by $\mathbf{B}$ will increase the insensitivity of the distribution map to the perturbations.
>
> • Distribution map with temporal correlations. If the perturbation $\mathbf{A}\_{t}(\mathbf{M})\mathbf{x}\_{t}$ is large, we will have a sequence of temporal correlated distribution maps introduced by the dynamical model. We have explored the unique sensitivity temporal propagation and aggregation structure for establishing the existence condition of a performative stable solution.
>
> We believe that the above generalizations are part of the strengths, novelties, and contributions of our work, rather than a weakness.
>
> **Answer to the Question**: We highlight a unique challenge on the existence of optimal solution to the MDP [Line 2-5, Abstract, R1].
>
> Consider an illustrative example of standard 1-dimensional linear quadratic optimal control problem (P1):
> \begin{align}
> \min\_{u\_{t}}  \limsup_{T\rightarrow\infty}\frac{1}{T}\sum\_{t=0}^{T-1}\mathbb{E}[x\_{t}^{2}+u\_{t}^{2}]\\
> s.t. x\_{t+1}=A\_{t}x\_{t}+B\_{t}u\_{t}+w\_{t},\forall t\geq0. (P1)
> \end{align}
>
> From the linear dynamical system (LDS) perspective, the state and action spaces are infinite, i.e., $x\_{t},u\_{t}\in\mathbb{R},$ $w_{t}\sim\mathcal{N}(0,1)$ is the Gaussian additive noise, the state-action transition kernel is given by the constraint in (P1). It is well-known that the optimal control action $u_{t}^{*}$ exists if and only if the following Riccati recursion converges as $t\rightarrow\infty$:
> \begin{align}
> P_{t+1} & =A_{t}(B_{t}B_{t}+P_{t}^{-1})^{-1}A_{t}+1,P_{0}=1,\forall t\geq0.
> \end{align}
> If $A_{t}$ and $B_{t}$ keep time varying, the limit of $P_{t}$ may not exist, which results in the non-existence of optimal solution
> to the MDP.
>
> For MDP with finite state and action spaces, where $x_{0}=1$, $x\_{t}\in$ {1,2}, $u\_{t}\in${$1,10^{6}$}, $w_{t}=0, A_{t}=1,B_{t},$ fluctuates based on the pattern of 1, 0, -1,  $\forall t\geq0,$ the optimal solution exists and is given by $u_{t}^{*}=1$.
>
> For scenario considered in this paper, where $A_{t}$ is performative, the situation will be even more complicated and challenging.
>
> [R1]. Kiefer S, Mayr R, Shirmohammadi M, et al. How to play in infinite MDPs, 47th International Colloquium on Automata, Languages and Programming. Schloss Dagstuhl-Leibniz-Zentrum fuer Informatik, Germany, 2020: 1-18.
>
> [R2]. Relund Nielsen L, Jørgensen E, Højsgaard S. Embedding a state space model into a Markov decision process. Annals of Operations Research, 2011, 190: 289-309.

---

> > ### Author Response · Authors · 2024-08-11
> > **Seeking Your Feedback**
> >
> > Dear wfwc,
> >
> > We appreciate your time and effort in reviewing our submission. We have now provided further clarification, explanation, and discussion to address your concerns. We hope you've had a chance to review them, consider whether our response addresses your concerns, and reevaluate your score accordingly. Please let us know if you have any further questions or need clarification. Thank you very much.
> >
> > Warmest regards,
> >
> > Authors

---

> > ### Comment · Reviewer_wfwc · 2024-08-12
> >
> > **Convexity:**  The *convexity* of $c_t$ in $M$ follows quite straightforwardly from the convexity of $c_t$ in $x_t$.  While the author seems to emphasize that the *strong convexity* aspect is the more challenging part, the reviewer acknowledges this but does not find it particularly compelling.
> >
> > **Difference from existing approach**  The reviewer acknowledges that a linear dynamic model differs from an MDP with finite states.   Maybe a more accurate question is **Can you highlight the challenge the is unique your problem on PP on linear dynamical system compared to existing works (the original paper on PP and PP on MDP)?**  Upon closer examination, the proof structure appears similar to the original PP framework [22]. Could you elaborate on this beyond what was addressed in your response to Reviewer e5pZ?
> >
> >
> > **Additional Question** The reviewer wants to understand when the system responds to the policy $M$ instead of action $u_t$.  In particular, the system does not have direct access to $M$, but only $u_t$ in each round.  Can the author provide examples to illustrate this? Or explain Equation (15) and (16) in example 1?

---

> > > ### Author Response · Authors · 2024-08-14
> > >
> > > Dear Reviewer wfwc,
> > >
> > > Thank you for participating in the discussion and your thoughtful comments. We hope we have addressed your concerns, and we would be happy to provide additional explanations and insights as needed.
> > >
> > > Warmest regards,
> > >
> > > Authors

---

> ### Author Response · Authors · 2024-08-12
>
> Thank you very much for participating in the discussion and providing us with plenty of valuable comments.
>
> **1. "While the author seems to emphasize that the strong convexity aspect is the more challenging part, the reviewer acknowledges this but does not find it particularly compelling.":**
>
> We thank the reviewer for acknowledging that we are indeed emphasizing the **linear-system-dynamics-dependent strong convexity analysis** of $\mathbb{E}[c\_{t}]$ in $\mathbf{M}$.
>
> As discussed in our paper above Lemma 2, the strong convexity of $c_{t}(\mathbf{x},\mathbf{u})$
> over the state-action space $(\mathbf{x},\mathbf{u})$ does not by itself imply the strong convexity of $\mathbb{E}[c\_{t}]$
> over the space of policies $\mathbf{M}$. This is because the policy
> $\mathbf{M}$, which maps from a space of dimensionality $H\times d\_{x}\times d\_{u}$
> to that of $d\_{x}+d\_{u}$, is not necessarily full column-rank.
>
> Under the conditions: (1) general $c\_{t}$ that is strongly convex in $(\mathbf{x},\mathbf{u})$; (2) general linear dynamics (i.e., $\mathbf{A},\mathbf{B}$ and $\mathbf{K}$) in Definition 2; and (3) general performative disturbances {$\mathbf{\Delta}\_i$}$\_{0\leq i<T}$ with general distributional maps {$\mathcal{D}\_i(\mathbf{M})$}$\_{0\leq i<T}$ in Assumption A2 and A3, identifying the strong convexity constant min{$\frac{\mu\sigma^{2}}{2},\frac{\mu\sigma^{2}\gamma^{2}}{64\kappa^{10}}$} of $\mathbb{E}[c\_{t}]$ in $\mathbf{M}$, where $\kappa$ and $\gamma$ are linear system dynamics-dependent, requires delicate linear-system-dynamics-dependent strong convexity analysis that is unique to performative linear dynamical systems.
>
> **2. "Difference from existing approach... ":**
>
> We thank the reviewer for giving us the opportunity to highlight the unique challenge in our PP for linear dynamical systems.
>
> **Original PP**: We first point out that the original PP is **non-stateful** because all the data samples follow a static distribution $Z\sim\mathcal{D}(\theta).$
>
> **PP on MDP with finite states [R1]**: The authors first introduced an occupancy measure $d(s,a)$ for each state-action pair
> $(s,a)\in\mathcal{S}\times\mathcal{A}.$ Since the number of states and actions is finite, the Markovian temporal correlation
> of the performative states is then modeled as a linear function of the concatenated occupancy measure $d=[d(s,a)]\_{(s,a)\in\mathcal{S}\times\mathcal{A}.}$ The authors further added a regulation term in the optimization objective function to guarantee the strong convexity in $d$ and then obtained the performative stable solution $d^{PS}.$ However, this method is not applicable to our case because:
>
> **(1)** $d(s,a)$ and $d$ cannot be constructed for MDP with infinite states;
>
> **(2)** We cannot artificially add a regulation term to make the optimization objective function strongly convex.
>
> **PP on linear dynamical system (our paper)**: In addition to the **linear-system-dynamics-dependent strong convexity analysis**, the linear dynamic system will introduce **a unique challenge of stateful performative transition maps of performative distributions**, which cannot be covered by the above original PP and PP on MDP with finite states. Specifically,
>
> **$\bullet$** When there is no performative disturbance to the state transition matrix $\mathbf{A}$, our **PP via linear dynamical system modeling** can be reduced to the **linearized general stateful distribution map** in the seminal work of **stateful PP [R2]**.
>
> **$\bullet$** If there exist performative disturbances to $\mathbf{A}$, our **PP via linear dynamical system modeling** will **uniquely introduce stateful performative transition maps of the performative distributions**. This **extends** the technical results of **fixed performative distribution transition map** in the stateful PP work [R2].
>
> Therefore, compared to existing works (the original paper on PP, PP on MDP with finite states and stateful PP), our PP on linear dynamical system **has the unique potential** to enhance the understanding of general performative prediction.
>
>
> [R1]. Mandal D, Triantafyllou S, Radanovic G. Performative reinforcement learning, International Conference on Machine Learning. pages 23642–23680. PMLR, 2023.
>
> [R2]. Gavin Brown, Shlomi Hod, and Iden Kalemaj. Performative prediction in a stateful world. In International conference on artificial intelligence and statistics, pages 6045–6061. PMLR, 2022.

---

> ### Author Response · Authors · 2024-08-12
>
> **3. Additional Question.**
>
> Thanks for your additional question. In a high level sense, consider
> a class of application scenarios involving a group of linear stochastic
> dynamic sub-systems, where each sub-system has a random state transition
> matrix. If we construct a new system state by linearly mixing the
> states of all the sub-systems using a weight matrix $\mathbf{M}$ (policy),
> then the performative state transition matrix of this new system will
> respond to the policy $\mathbf{M}$ instead of the action $\mathbf{u}_{t}.$
> Our Example 1 is also constructed based on this intuition.
>
> **Detailed explanation for Equation (15) and (16).** In Example 1, we can view each stock as a sub-system with a random state transition, where the evolution of price of $i$-th stock follows the stochastic volatility model [R3, R4] of
> \begin{align*}
>  & \log s_{t+1}^{\left(i\right)}=\log s_{t}^{\left(i\right)}+\left(\frac{r-\frac{1}{2}\left(v_{t}^{\left(i\right)}\right)^{2}}{T}+\frac{v_{t}^{\left(i\right)}}{\sqrt{T}}\right),\ s_{1}^{\left(i\right)}>0,\forall1\leq t<T,
> \end{align*}
> or equivalently
> \begin{align*}
>  & s_{t+1}^{\left(i\right)}=e^{\left(\frac{r-\frac{1}{2}\left(v_{t}^{\left(i\right)}\right)^{2}}{T}+\frac{v_{t}^{\left(i\right)}}{\sqrt{T}}\right)}s_{t}^{\left(i\right)},\ s_{1}^{\left(i\right)}>0,\forall1\leq t<T.
> \end{align*}
>
> We next construct a new state, i.e., the return for the $j$-th portfolio
> $q_{t}^{\left(j\right)},$ using a couple of linear mixing weights $\left[m^{j,1},\cdots,m^{j,N}\right],$
> i.e., the investment strategy for the $N$ stocks.
>
> The concatenated return vector $\mathbf{q}\_{t}$ for all portfolios under investment policy $\mathbf{M}$ is given by
> \begin{align}
>  \mathbf{q}\_{t+1}=\mathbf{V}\_{t}^{(\mathbf{M})}\mathbf{q}\_{t}, (1)
> \end{align}
> where performative state transition matrix $\mathbf{V}\_{t}^{\left(\mathbf{M}\right)}$ is a diagonal matrix with the $j$-th diagonal element given by
> \begin{align*}
>  & \left(\mathbf{V}\_{t}^{\left(\mathbf{M}\right)}\right)\_{j,j}=\sum\_{i=1}^{N}m^{j,i}e^{\left(\frac{r-\frac{1}{2}\left(v_{t}^{\left(i\right)}\right)^{2}}{T}+\frac{v_{t}^{\left(i\right)}}{\sqrt{T}}\right)}
> \end{align*}
>
> Equations (15) and (16) in Example 1 are used to illustrate the performative
> state transition matrix for the concatenated state $\mathbf{x}\_{t}=\left[\begin{array}{c}
> \mathbf{r}\_{t}\\\\
> \mathbf{q}\_{t}
> \end{array}\right]$, where $\mathbf{r}\_{t}=\mathbf{q}\_{t}-\mathbb{E}\left[\mathbf{q}\_{t}\right]$
> is the mean-shifted return. Based on (1), it follows immediately that the performative state transition matrix associated with
> $\mathbf{x}\_{t}$ is
>
> $$ \mathbf{A}\_{t}\left(\mathbf{M}\right)=\left[\begin{array}{cc}
> \mathbb{E}\left[\mathbf{V}\_{t}^{\left(\mathbf{M}\right)}\right] & \mathbf{V}\_{t}^{\left(\mathbf{M}\right)}-\mathbb{E}\left[\mathbf{V}\_{t}^{\left(\mathbf{M}\right)}\right]\\\\
> \mathbf{0}\_{N\times N} & \mathbf{V}\_{t}^{\left(\mathbf{M}\right)}
> \end{array}\right].
> $$
>
> Equations (15) and (16) then hold by choosing ${\mathbf{A}}=\mathbf{I}\_{2N\times2N}$
> and noting that $\mathbf{\Delta}\_{t}=\mathbf{A}\_{t}\left(\mathbf{M}\right)-\mathbf{A}.$
>
> [R3]. Huyên Pham, Marco Corsi, and Wolfgang J Runggaldier. Numerical approximation by quanti- zation of control problems in finance under partial observations. In Handbook of Numerical Analysis, volume 15, pages 325–360. Elsevier, 2009.
>
> [R4]. Gallant A R, Hsieh D, Tauchen G. Estimation of stochastic volatility models with diagnostics[J]. Journal of econometrics, 1997, 81(1): 159-192.

---

### Author Rebuttal · Authors · 2024-08-07

We thank all reviewers for recognizing that our paper provides a thorough and mathematically rigorous theoretical investigation of performative linear control, where control policies can directly affect the dynamics of the system (reviewers wfwc, JGbs, nkz4, e5pZ), which fills a gap in traditional control theory (reviewer JGbs), is very exciting (reviewer nkz4), and is a good step towards improving the understanding of general performative prediction (reviewer wfwc). We would now like to address frequently raised comments, which we believe will clarify any previous concerns:

**1.Assumption of strong convexity.** For the existing performative prediction works [3,5,12,14,15,20,22,25], **one can directly assume the objective function $\mathbb{E}\left[c_{t}\left(\mathbf{x},\mathbf{u}\right)\right]$ is $\mu$-strongly convex w.r.t. optimization variable $\mathbf{M}$ (i.e., the disturbance-action policy)** without any problem-specific theoretical convexity analysis. However, in our work, **we only have the freedom to assume the per stage cost $c_{t}\left(\mathbf{x},\mathbf{u}\right)$ is $\mu$-strongly convex for the state-action pair $\left(\mathbf{x},\mathbf{u}\right)$**, which does not by itself imply that expected per stage cost $\mathbb{E}\left[c_{t}\left(\mathbf{x},\mathbf{u}\right)\right]$ is strongly convex w.r.t. our optimization variable $\mathbf{M}$. We provided rigorous theoretical analysis that $\mathbb{E}\left[c_{t}\left(\mathbf{x},\mathbf{u}\right)\right]$
is strongly convex w.r.t. $\mathbf{M}$ and the strongly convex constant
is **not $\mu$, but min{$\frac{\mu\sigma^{2}}{2},\frac{\mu\sigma^{2}\gamma^{2}}{64\kappa^{10}}$}**, where $\kappa$ and $\gamma$ are problem-specific parameters related to system dynamics.

**2. Existence and uniqueness of performative stable solution.** Our sufficient
condition for existence and uniqueness of performative stable solution
has a **distributional sensitivity temporal propagation and aggregation
structure, which is unique to the performative dynamical systems**.

**3. Convergence analysis.** To guarantee the convergence of our proposed
algorithm to the performative stable solution, the designing of the
step sizes relies heavily on the strongly convex constant min{$\frac{\mu\sigma^{2}}{2},\frac{\mu\sigma^{2}\gamma^{2}}{64\kappa^{10}}$}, and the distributional sensitivity temporal propagation and aggregation
condition. **These are unique in our performative dynamical systems,
and have never been exploited in existing performative prediction works** [3,5,12,14,15,20,22,25].

**4. Generalization to the existing performative prediction framework.**
Our proposed framework can effectively accommodate distribution maps with perturbations and temporal correlations, which is a meaningful generalization of the standard distribution maps considered in existing performative prediction works [3,5,12,14,15,20,22,25].

---

> ### Author Response · Authors · 2024-08-12
> **Summary of key differences between our work and existing PP works (original PP, PP on MDP with finite states and stateful PP)**
>
> As suggested by Reviewer wfwc and Reviewer nkz4, we here recap the key differences between our work and the existing PP works.
>
> **Original PP [22]**: We first point out that the original PP is **non-stateful**, where all the data samples follow a static distribution $Z\sim\mathcal{D}(\theta).$ Besides, **strong convexity is directly assumed without any problem-specific analysis**.
>
> **PP on MDP with finite states [R1]**: The authors first introduced an occupancy measure $d(s,a)$ for each state-action pair
> $(s,a)\in\mathcal{S}\times\mathcal{A}.$ Since the number of states and actions is finite, the Markovian temporal correlation
> of the performative states is then modeled as a linear function of the concatenated occupancy measure $d=[d(s,a)]\_{(s,a)\in\mathcal{S}\times\mathcal{A}.}$ **The strong convexity is artificial**. The authors further added a regulation term in the optimization objective function to guarantee the strong convexity in $d$ and then obtained the performative stable solution $d^{PS}.$ However, this method is not applicable to our case because:
>
> **(1)** $d(s,a)$ and $d$ cannot be constructed for MDP with infinite states;
>
> **(2)** We cannot artificially add a regulation term to make the optimization objective function strongly convex.
>
> **Stateful PP [R2]**: This seminal work considers the **stateful distribution map**, where the observed data distribution is stateful and depends on the the history of previously deployed polices. But the **strong convexity is also directly assumed** as in original PP [22] without any problem analysis.
>
> **PP on linear dynamical system (our paper)**: We provide problem-specific **linear-system-dynamics-dependent strong convexity analysis**. Our work enables the **stateful performative transition maps of performative distributions** and **extends** the technical results of the **fixed performative distribution transition map** in the stateful PP work [R2].
>
> The **stateful performative transition map modeling via linear dynamical systems** in our paper has potential to enhance the understanding of general performative prediction.
>
> [R1]. Mandal D, Triantafyllou S, Radanovic G. Performative reinforcement learning, International Conference on Machine Learning. pages 23642–23680. PMLR, 2023.
>
> [R2]. Gavin Brown, Shlomi Hod, and Iden Kalemaj. Performative prediction in a stateful world. In International conference on artificial intelligence and statistics, pages 6045–6061. PMLR, 2022.

---

### Author Response · Authors · 2024-08-10
**seeking feedback from reviewers**

Dear AC,

We thank all reviewers for their valuable time in reviewing our paper. The discussion period will end in a few days, but we have not received any feedback from the reviewers regarding our rebuttal. We have addressed all concerns of the reviewers and hope to discuss with them and know if there are any further clarifications to be made. Could you please kindly remind them to respond to our rebuttal? Thank you very much.

Best regards,

Authors

---

### Decision · Program_Chairs · 2024-09-25

**Decision:**

Accept (poster)

**Comment:**

The paper establishes an interesting and novel connection between performative prediction and linear control. The paper also provides theoretical analysis for stable control solutions in this framework, propose an algorithm, and illustrate the approach and results with a numerical example. Although there has been some divergence on the appreciation of the technical contributions, all reviewers appreciate the novelty of the framework. I believe this paper can inspire further research. Indeed, during the review process, the authors have shown their work also generalizes recent seminal work on this area.

The reviewers highlighted some improvements that must be addressed for the camera ready version.